# DINOv3

**Oriane Siméoni**[*]   **Huy V. Vo**[*]   **Maximilian Seitzer**[*]   **Federico Baldassarre**[*]   **Maxime Oquab**[*]

**Cijo Jose**   **Vasil Khalidov**   **Marc Szafraniec**   **Seungeun Yi**   **Michaël Ramamonjisoa**

**Francisco Massa**   **Daniel Haziza**   **Luca Wehrstedt**   **Jianyuan Wang**

**Timothée Darcet**   **Théo Moutakanni**   **Leonel Sentana**   **Claire Roberts**

**Andrea Vedaldi**   **Jamie Tolan**   **John Brandt**[1]   **Camille Couprie**

**Julien Mairal**[2]   **Hervé Jégou**   **Patrick Labatut**   **Piotr Bojanowski**

*Meta AI Research*       [1] *WRI*       [2] *Inria*

[*] *corresponding authors: {osimeoni,huyvvo,seitzer,baldassarre,qas}@meta.com*
**Reviewed on OpenReview:** *https://openreview.net/forum?id=2NlGyqNjns*

## Abstract

Self-supervised learning holds the promise of eliminating the need for manual data annotation, enabling models to scale effortlessly to massive datasets and larger architectures. By not being tailored to specific tasks or domains, this training paradigm has the potential to learn visual representations from diverse sources, ranging from natural to aerial images—using a single algorithm. This technical report introduces DINOv3, a major milestone toward realizing this vision by leveraging simple yet effective strategies. First, we leverage the benefit of scaling both dataset and model size by careful data preparation, design, and optimization. Second, we introduce a new method called Gram anchoring, which effectively addresses the known yet unsolved issue of dense feature maps degrading during long training schedules. Finally, we apply post-hoc strategies that further enhance our models' flexibility with respect to resolution, model size, and alignment with text. As a result, we present a versatile vision foundation model that outperforms the specialized state of the art across a broad range of settings, without fine-tuning. DINOv3 produces high-quality dense features that achieve outstanding performance on various vision tasks, significantly surpassing previous self- and weakly-supervised foundation models. We also share the DINOv3 suite of vision models, designed to advance the state of the art on a wide spectrum of tasks and data by providing scalable solutions for diverse resource constraints and deployment scenarios.

## 1 Introduction

Foundation models have become a central building block in modern computer vision, enabling broad generalization across tasks and domains through a single, reusable model. Self-supervised learning (SSL) is a powerful approach for training such models, by learning directly from raw pixel data and leveraging the natural co-occurrences of patterns in images. Unlike weakly and fully supervised pretraining methods (Radford et al., 2021; Dehghani et al., 2023; Bolya et al., 2025) which require images paired with high-quality metadata, SSL unlocks training on massive, raw image collections. This is particularly effective for training large-scale visual encoders thanks to the availability of virtually unlimited training data. DINOv2 (Oquab et al., 2024) exemplifies these strengths, achieving impressive results in image understanding tasks (Wang et al., 2025) and enabling pre-training for complex domains such as histopathology (Chen et al., 2024). Models trained with SSL exhibit additional desirable properties: they are robust to input distribution shifts, provide strong global and local features, and generate rich embeddings that facilitate physical scene understanding. Since SSL models are not trained for any specific downstream task, they produce versatile and robust generalist features. For instance, DINOv2 models deliver strong performance across diverse tasks and domains without requiring task-specific finetuning, allowing a single frozen backbone to serve multiple purposes. Importantly,

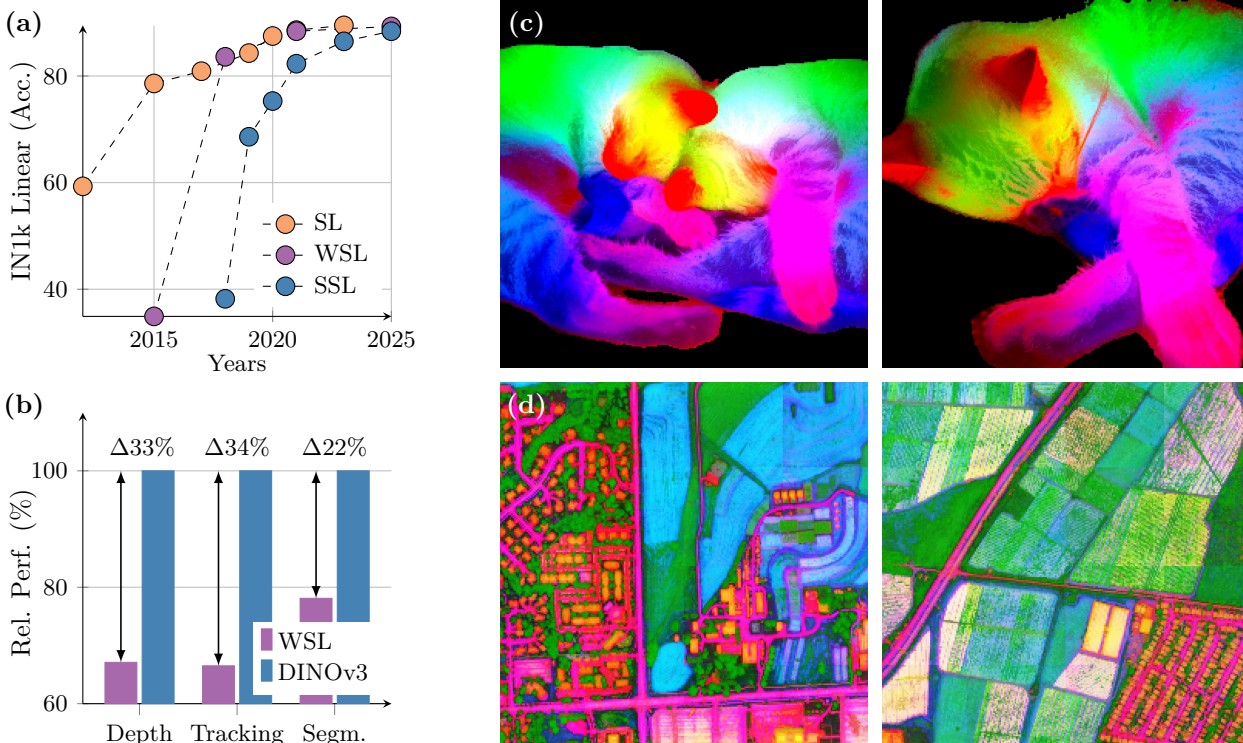

Figure 1: (a) Evolution of linear probing results on ImageNet1k (IN1k) over the years, comparing fully-(SL), weakly- (WSL) and self-supervised learning (SSL) methods. Despite coming into the picture later, SSL has quickly progressed and now reached the Imagenet accuracy plateau of recent years. On the other hand, we demonstrate that SSL offers the unique promise of high-quality dense features. With DINOv3, we markedly improve over weakly-supervised models on dense tasks, as shown by the relative performance of the best-in-class WSL models to DINOv3 (b). We also produce PCA maps of features obtained from high resolution images with DINOv3 trained on natural (c) and aerial images (d).

self-supervised learning is especially suitable to train on the vast amount of available observational data in domains like histopathology (Vorontsov et al., 2024), biology (Kim et al., 2025), medical imaging (Pérez-García et al., 2025), remote sensing (Cong et al., 2022; Tolan et al., 2024), astronomy (Parker et al., 2024), or high-energy particle physics (Dillon et al., 2022). These domain often lack metadata and have already been shown to benefit from foundation models like DINOv2. Finally, SSL, requiring no human intervention, is well-suited for lifelong learning amid the growing volume of web data.

In practice, the promise of SSL, namely producing arbitrarily large and powerful models by leveraging large amounts of unconstrained data, remains challenging at scale. While model instabilities and collapse are mitigated by the heuristics proposed by Oquab et al. (2024), more problems emerge from scaling further. First, it is unclear how to collect useful data from unlabeled collections. Second, in usual training practice, employing cosine schedules implies knowing the optimization horizon a priori, which is difficult when training on large image corpora. Third, the performance of the features gradually decreases after early training, confirmed by visual inspection of the patch similarity maps. This phenomenon appears in longer training runs with models above ViT-Large size (300M parameters), reducing the usefulness of scaling DINOv2.

Addressing the problems above leads to this work, *DINOv3*, which advances SSL training at scale. We demonstrate that a *single frozen SSL backbone* can serve as a universal visual encoder that achieves state-of-the-art performance on challenging downstream tasks, outperforming supervised and metadata-reliant pre-training strategies. Our research is guided by the following objectives: (1) training a foundational model versatile across tasks and domains, (2) improving the shortcomings of existing SSL models on dense features, (3) disseminating a family of models that can be used off-the-shelf. We discuss the three aims in the following.

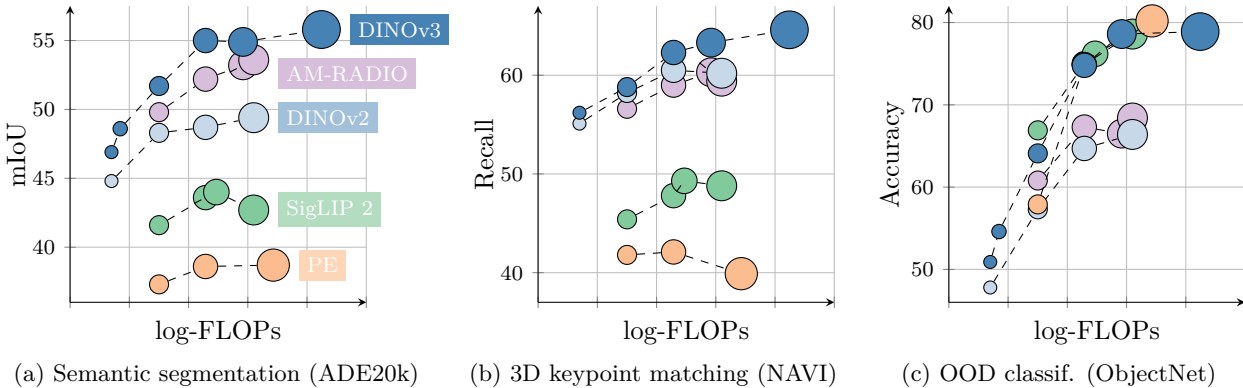

(a) Semantic segmentation (ADE20k)    (b) 3D keypoint matching (NAVI)    (c) OOD classif. (ObjectNet)

Figure 2: Performance of the DINOv3 family of models, compared to other families of self- or weakly-supervised models, on different benchmarks. DINOv3 significantly surpasses others on dense benchmarks, including models that leverage mask annotation priors such as AM-RADIO (Heinrich et al., 2025).

**Strong & Versatile Foundational Models**  DINOv3 aims to offer a high level of versatility along two axes, which is enabled by the scaling of the model size and training data. First, a key desirable property for SSL models is to achieve excellent performance while being kept frozen, ideally reaching similar state-of-the-art results as specialized models. In that case, a single forward pass can deliver cutting-edge results across multiple tasks, leading to substantial computational savings—an essential advantage for practical applications, particularly on edge devices. We show the wide breadth of tasks that DINOv3 can successfully be applied to in Sec. 6. Second, a scalable SSL training pipeline that does not depend on metadata unlocks numerous scientific applications. By pre-training on a diverse set of images, whether web images or observational data, SSL models generalize across a large set of domains and tasks. As illustrated in Fig. 1(d), the PCA of DINOv3 features extracted from a high-resolution aerial image clearly allows to separates roads, houses, and greenery, highlighting the model's feature quality.

**Superior Feature Maps Through Gram Anchoring**  Another key feature of DINOv3 is a significant improvement of its dense feature maps. The DINOv3 SSL training strategy aims at producing models excelling at high-level semantic tasks while producing excellent feature maps amenable to solving geometric tasks such as depth estimation, or 3D matching. In particular, the models should produce dense features that can be used off-the-shelf or with little post-processing. The compromise between dense and global representation is especially difficult to optimize when training with vast amounts of images, since the objective of high-level understanding can conflict with the quality of the dense feature maps. These contradictory objectives lead to a collapse of dense features with large models and long training schedules. Our new Gram anchoring strategy effectively mitigates this collapse (see Sec. 4). As a result, DINOv3 obtains significantly better dense feature maps than DINOv2, staying clean even at high resolutions (see Fig. 3).

**The DINOv3 Family of Models**  Solving the degradation of dense feature map with Gram anchoring unlocks the power of scaling. As a consequence, training a much larger model with SSL leads to significant performance improvements. In this work, we successfully train a DINO model with 7B parameters. Since such a large model requires significant resources to run, we apply distillation to compress its knowledge into smaller variants. As a result, we present the *DINOv3 family of vision models*, a comprehensive suite designed to address a wide spectrum of computer vision challenges. This model family aims to advance the state of the art by offering scalable solutions adaptable to diverse resource constraints and deployment scenarios. The distillation process produces model variants at multiple scales, including Vision Transformer (ViT) Small, Base, and Large, as well as ConvNeXt-based architectures. Notably, the efficient and widely adopted ViT-L model achieves performance close to that of the original 7B teacher across a variety of tasks. Overall, the DINOv3 family demonstrates strong performance on a broad range of benchmarks, matching or exceeding the accuracy of competing models on global tasks, while significantly outperforming them on dense prediction tasks, as visible in Fig. 2.

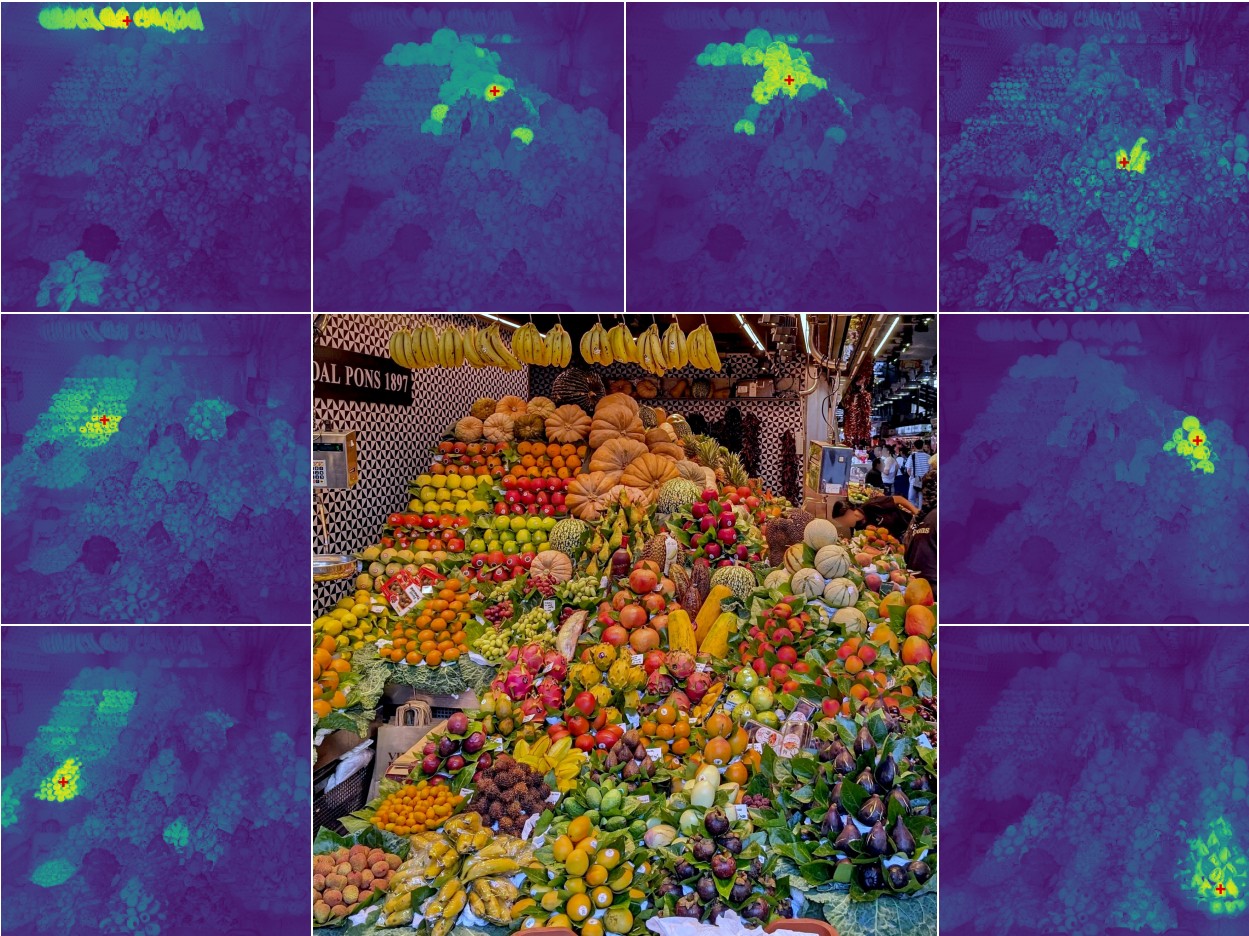

Figure 3: High-resolution dense features. We visualize the cosine similarity maps obtained with DINOv3 output features between the patches marked with a red cross and all other patches. Input image at 4096×4096. *Please zoom in*, do you agree with DINOv3?

**Overview of Contributions**   In this work, we introduce multiple contributions to address the challenge of scaling SSL towards a large frontier model. We build upon recent advances in automatic data curation (Vo et al., 2024) to obtain a large "background" training dataset that we carefully mix with a bit of specialized data (ImageNet-1k). This allows leveraging large amounts of unconstrained data to improve the model performance. This contribution **(i)** around data scaling will be described in Sec. 3.1.

We increase our main model size to 7B parameters by defining a custom variant of the ViT architecture. We include modern position embeddings (axial RoPE) and develop a regularization technique to avoid positional artifacts. Departing from the multiple cosine schedules in DINOv2, we train with constant hyperparameter schedules for 1M iterations. This allows producing models with stronger performance. This contribution **(ii)** on model architecture and training will be described in Sec. 3.2.

With the above techniques, we are able to train a model following the DINOv2 algorithm at scale. However, as mentioned previously, scale leads to a degradation of dense features. To address this, we propose a core improvement of the pipeline with a Gram anchoring training phase. This cleans the noise in the feature maps, leading to impressive similarity maps, and drastically improving the performance on both parametric and non-parametric dense tasks. This contribution **(iii)** on Gram training will be described in Sec. 4.

Following previous practice, the last steps of our pipeline consist of a high-resolution post-training phase and distillation into a series of high-performance models of various sizes. For the latter, we develop a novel and

efficient single-teacher multiple-students distillation procedure. This contribution **(iv)** transfers the power of our 7B frontier model to a family of smaller practical models for common usage, that we describe in Sec. 5.2.

As measured in our thorough benchmarking, results in Sec. 6 show that our approach defines a new standard in dense tasks and performs comparably to CLIP derivatives on global tasks. In particular, *with a frozen vision backbone*, we achieve state-of-the-art performance on longstanding computer vision problems such as object detection (COCO detection, mAP 66.1) and image segmentation (ADE20k, mIoU 63.0), outperforming specialized fine-tuned pipelines. Moreover, we provide evidence of the generality of our approach across domains by applying the DINOv3 algorithm to satellite imagery, in Sec. 8, surpassing all prior approaches.

## 2 Related Work

**Self-Supervised Learning** Learning without annotations requires an artificial learning task that provides supervision in lieu for training. The art and challenge of SSL lies in carefully designing these so-called pre-text tasks in order to learn powerful representations for downstream tasks. The language domain, by its discrete nature, offers straightforward ways to set up such tasks, which led to many successful unsupervised pre-training approaches for text data. Examples include word embeddings (Mikolov et al., 2013; Bojanowski et al., 2017), sentence representations (Devlin et al., 2018; Liu et al., 2019), and plain language models (Mikolov et al., 2010; Zaremba et al., 2014). In contrast, computer vision presents greater challenges due to the continuous nature of the signal. Early attempts mimicking language approaches extracted supervisory signals from parts of an image to predict other parts, *e.g.* by predicting relative patch position (Doersch et al., 2015), patch re-ordering (Noroozi & Favaro, 2016; Misra & Maaten, 2020), or inpainting (Pathak et al., 2016). Other tasks involve re-colorizing images (Zhang et al., 2016) or predicting image transformations (Gidaris et al., 2018).

Among these tasks, *inpainting-based* approaches have gathered significant interest thanks to the flexibility of the patch-based ViT architecture (He et al., 2021; Bao et al., 2021; El-Nouby et al., 2021). The objective is to reconstruct corrupted regions of an image, which can be viewed as a form of denoising auto-encoding and is conceptually related to the masked token prediction task in BERT pretraining (Devlin et al., 2018). Notably, He et al. (2021) demonstrated that pixel-based masked auto-encoders (MAE) can be used as strong initializations for finetuning on downstream tasks. In the following, Baevski et al. (2022; 2023); Assran et al. (2023) showed that predicting a *learned latent space* instead of the pixel space leads to more powerful, higher-level features—a learning paradigm called JEPA: "Joint-Embedding Predictive Architecture" (LeCun, 2022). Recently, JEPAs have also been extended to video training (Bardes et al., 2024; Assran et al., 2025).

A second line of work, closer to ours, leverages *discriminative signals between images* to learn visual representations. This family of methods traces its origins to early deep learning research (Hadsell et al., 2006), but gained popularity with the introduction of instance classification techniques (Dosovitskiy et al., 2016; Bojanowski & Joulin, 2017; Wu et al., 2018). Subsequent advancements introduced contrastive objectives and information-theoretic criteria (Hénaff et al., 2019; He et al., 2020; Chen & He, 2020; Chen et al., 2020a; Grill et al., 2020; Bardes et al., 2021), as well as self clustering-based strategies (Caron et al., 2018; Asano et al., 2020; Caron et al., 2020; 2021). More recent approaches, such as iBOT (Zhou et al., 2021), combine these discriminative losses with masked reconstruction objectives. All of these methods show the ability to learn strong features and achieve high performance on standard benchmarks like ImageNet (Russakovsky et al., 2015). However, most face challenges scaling to larger model sizes (Chen et al., 2021).

**Vision Foundation Models** The deep learning revolution began with the AlexNet breakthrough (Krizhevsky et al., 2012), a deep convolutional neural network that outperformed all previous methods on the ImageNet challenge (Deng et al., 2009; Russakovsky et al., 2015). Already early on, features learned end-to-end on the large manually-labeled ImageNet dataset were found to be highly effective for a wide range of transfer learning tasks (Oquab et al., 2014). Early work on vision *foundation models* then focused on architecture development, including VGG (Simonyan & Zisserman, 2015), GoogleNet (Szegedy et al., 2015), and ResNets (He et al., 2016).

Given the effectiveness of *scaling*, subsequent works explored training larger models on big datasets. Sun et al. (2017) expanded supervised training data with the proprietary JFT dataset containing 300 million

labeled images, showing impressive results. JFT also enabled significant performance gains for Kolesnikov et al. (2020). In parallel, scaling was explored using a combination of supervised and unsupervised data. For instance, an ImageNet-supervised model can be used to produce pseudo-labels for unsupervised data, which then serve to train larger networks (Yalniz et al., 2019). Subsequently, the availability of large supervised datasets such as JFT also facilitated the adaptation of the transformer architecture to computer vision (Dosovitskiy et al., 2020). In particular, achieving performance comparable to that of the original vision transformer (ViT) without access to JFT requires substantial effort (Touvron et al., 2020; 2022). Due to the learning capacity of ViTs, scaling efforts were further extended by Zhai et al. (2022a), culminating in the very large ViT-22B encoder (Dehghani et al., 2023).

Given the complexity of manually labeling large datasets, *weakly-supervised training*—where annotations are derived from metadata associated with images—provides an effective alternative to supervised training. Early on, Joulin et al. (2016) demonstrated that a network can be pre-trained by simply predicting all words in the image caption as targets. This initial approach was further refined by leveraging sentence structures (Li et al., 2017), incorporating other types of metadata and involve curation (Mahajan et al., 2018), and scaling (Singh et al., 2022). However, weakly-supervised algorithms only reached their full potential with the introduction of contrastive losses and the joint-training of caption representations, as exemplified by Align (Jia et al., 2021) and CLIP (Radford et al., 2021).

This highly successful approach inspired numerous *open-source reproductions and scaling efforts.* Open-CLIP (Cherti et al., 2023) was the first open-source effort to replicate CLIP by training on the LAION dataset (Schuhmann et al., 2021); following works leverage pre-trained backbones by fine-tuning them in a CLIP-style manner (Sun et al., 2023; 2024). Recognizing that data collection is a critical factor in the success of CLIP training, MetaCLIP (Xu et al., 2024) precisely follows the original CLIP procedure to reproduce its results, whereas Fang et al. (2024a) use supervised datasets to curate pretraining data. Other works focus on improving the training loss, *e.g.* using a sigmoid loss in SigLIP (Zhai et al., 2023), or leveraging a pre-trained image encoder (Zhai et al., 2022b). Ultimately though, the most critical components for obtaining cutting-edge foundation models are abundant high-quality data and substantial compute resources. In this vein, SigLIP 2 (Tschannen et al., 2025) and Perception Encoder (PE) (Bolya et al., 2025) achieve impressive results after training on more than 40B image-text pairs. The largest PE model is trained on 86B billion samples with a global batch size of 131K. Finally, a range of more complex and natively multimodal approaches have been proposed; these include contrastive captioning (Yu et al., 2022), masked modeling in the latent space (Bao et al., 2021; Wang et al., 2022b; Fang et al., 2023; Wang et al., 2023a), and auto-regressive training (Fini et al., 2024).

In contrast, relatively little work has focused on *scaling unsupervised image pretraining.* Early efforts include Caron et al. (2019) and Goyal et al. (2019) utilizing the YFCC dataset (Thomee et al., 2016). Further progress has been achieved by focusing on larger datasets and models (Goyal et al., 2021; 2022a), as well as initial attempts at data curation for SSL (Tian et al., 2021). Careful tuning of the training algorithms, larger architectures, and more extensive training data lead to the impressive results of DINOv2 (Oquab et al., 2024); for the first time, an SSL model matched or surpassed open-source CLIP variants on a range of tasks. This direction has recently been further pushed by Fan et al. (2025) by scaling to larger models without data curation, or by Venkataramanan et al. (2025) using open datasets and improved training recipes.

**Dense Transformer Features**  A broad range of modern vision applications consume *dense features* of pre-trained transformers, including multi-modal models (Liu et al., 2023; Beyer et al., 2024), generative models (Yu et al., 2025; Yao et al., 2025), 3D understanding (Wang et al., 2025), video understanding (Lin et al., 2023a; Wang et al., 2024c), and robotics (Driess et al., 2023; Kim et al., 2024). On top of that, traditional vision tasks such as detection, segmentation, or depth estimation require accurate local descriptors. To enhance the quality of SSL-trained local descriptors, a substantial body of work focuses on developing *local SSL losses.* Examples include leveraging spatio-temporal consistency in videos, *e.g.* using point track loops as training signal (Jabri et al., 2020), exploiting the spatial alignment between different crops of the same image (Pinheiro et al., 2020; Bardes et al., 2022), or enforcing consistency between neighboring patches (Yun et al., 2022). Darcet et al. (2025) show that predicting clustered local patches leads to improved dense representations. DetCon (Hénaff et al., 2021) and ORL (Xie et al., 2021) perform contrastive learning on

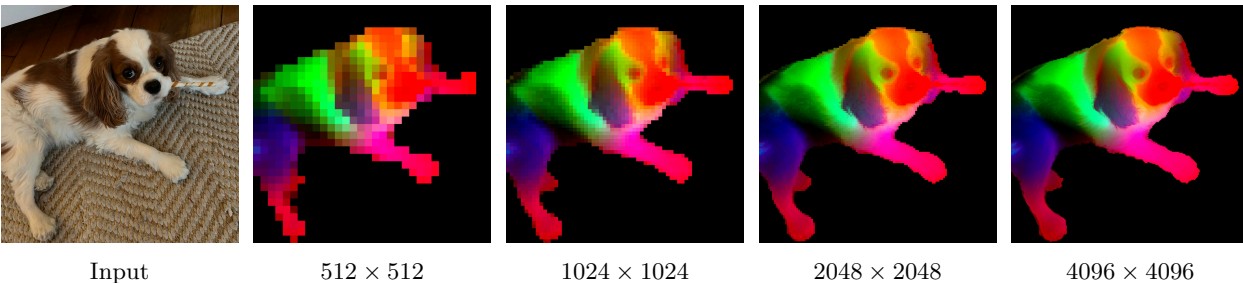

| Input | $512 \times 512$ | $1024 \times 1024$ | $2048 \times 2048$ | $4096 \times 4096$ |

Figure 4: DINOv3 at very high resolution. We visualize dense features of DINOv3 by mapping the first three components of a PCA computed over the feature space to RGB. To focus the PCA on the subject, we mask the feature maps via background subtraction. With increasing resolution, DINOv3 produces crisp features that stay semantically meaningful. We visualize more PCAs in Sec. 6.1.1.

region proposals but assume that such proposals exist *a priori*; this assumption is relaxed by approaches such as ODIN (Hénaff et al., 2022) and SlotCon (Wen et al., 2022). Without modifying the training objective, Darcet et al. (2024) demonstrate that introducing register tokens into the input sequence greatly enhances dense feature maps by mitigating high-norm artifacts. Subsequently, Wang et al. (2024b) propose a post-hoc fine-tuning regularization strategy that modify network parameters to address high-norm anomaly tokens, without requiring full retraining. More recent studies further show that such mitigation can be achieved without any model training (Jiang et al., 2025; Chen et al., 2025).

A recent trend are distillation-based, "*agglomerative*" methods that combine information from multiple image encoders with varying in global and local feature quality, trained using different levels of supervision (Ranzinger et al., 2024; Bolya et al., 2025): AM-RADIO (Ranzinger et al., 2024) combines the strengths of the fully-supervised SAM (Kirillov et al., 2023), the weakly-supervised CLIP, and the self-supervised DINOv2 into a unified backbone. The Perception Encoder (Bolya et al., 2025) similarly distills SAM(v2) into a specialized dense variant called PEspatial. They use an objective enforcing cosine similarity between student and teacher patches to be high, where their teacher is trained with mask annotations. Similar losses were shown to be effective in the context of style transfer, by reducing the inconsistency between the Gram matrices of feature dimensions (Gatys et al., 2016; Johnson et al., 2016; Yoo et al., 2024). In this work, we adopt a Gram objective to regularize cosine similarity between student and teacher patches, favoring them being close. In our case, we use earlier iterations of the SSL model itself as the teacher, demonstrating that early-stage SSL models effectively guides SSL training for both global and dense tasks.

Other works focus on post-hoc improvements to the local features of SSL-trained models. For example, Ziegler & Asano (2022) fine-tune a pre-trained model with a dense clustering objective; similarly, Salehi et al. (2023) fine-tune by aligning patch features temporally, in both cases enhance the quality of local features. Closer to us, Pariza et al. (2025) propose a patch-sorting based objective to encourage the student and teacher to produce features with consistent neighbor ordering. Without finetuning, STEGO (Hamilton et al., 2022) learns a non-linear projection on top of frozen SSL features to form compact clusters and amplify correlation patterns. Alternatively, Simoncini et al. (2024) augment self-supervised features by concatenating gradients from different self-supervised objectives to frozen SSL features. Recently, Wysoczańska et al. (2024) show that noisy feature maps are significantly improved through a weighted average of patches.

Related, but not specific to SSL, some recent works aim to generate high-resolution feature maps from ViT feature maps (Fu et al., 2024; Suri et al., 2024; Huang et al., 2025; Couairon et al., 2025; Wimmer et al., 2025), which are often low-resolution due to patchification of images. In contrast with this line of work, our models natively deliver high-quality dense feature maps that remain stable and consistent across resolutions, as shown in Fig. 4.

Table 1: Influence of training data on features quality shown via performance on downstream tasks. We compare datasets curated with *clustering* (Vo et al., 2024) and *retrieval* (Oquab et al., 2024) to *raw* data and to our data mixture. This ablation study is run for a shorter schedule of 200k iterations.

| Curation | Size | IN1k $k$-NN | IN1k Linear | ObjectNet | iNaturalist 2021 | Paris Retrieval |
|---|---|---|---|---|---|---|
| Random | 1.7B | 80.1 | 84.8 | 70.3 | 70.1 | 63.3 |
| Clustering | 1.7B | 79.4 | 85.4 | 72.3 | 81.3 | 85.2 |
| Retrieval | 139M | 84.0 | 86.7 | 70.7 | 86.0 | 82.7 |
| Ours | 1.7B | **84.6** | **87.2** | **72.8** | **87.0** | **85.9** |

## 3 Training at Scale Without Supervision

DINOv3 is a next-generation model designed to produce the most robust and flexible visual representations to date by pushing the boundaries of self-supervised learning. We draw inspiration from the success of large language models (LLMs), for which scaling-up the model capacity leads to outstanding *emerging properties*. By leveraging models and training datasets that are an order of magnitude larger, we seek to unlock the full potential of SSL and drive a similar paradigm shift for computer vision, unencumbered by the limitations inherent to traditional supervised or task-specific approaches. In particular, SSL produces rich, high-quality visual features that are not biased toward any specific supervision or task, thereby providing a versatile foundation for a wide range of downstream applications. While previous attempts at scaling SSL models have been hindered by issues of instability, this section describes how we harness the benefits of scaling with careful data preparation, design, and optimization. We first describe the dataset creation procedure (Sec. 3.1), then present the self-supervised SSL recipe used for this first training phase of DINOv3 (Sec. 3.2). This includes the choice of architecture, loss functions, and optimization techniques. The second training phase, focusing on dense features, will be described in Sec. 4.

### 3.1 Data Preparation

Data scaling is one of the driving factors behind the success of large foundation models (Touvron et al., 2023; Radford et al., 2021; Xu et al., 2024; Oquab et al., 2024). However, increasing naively the size of the training data does not necessarily translate into higher model quality and better performance on downstream benchmarks (Goyal et al., 2021; Oquab et al., 2024; Vo et al., 2024): Successful data scaling efforts typically involve careful data curation pipelines. These algorithms may have different objectives: either focusing on improving data *diversity* and *balance*, or data *usefulness*—its relevance to common practical applications. For the development of DINOv3, we combine two complementary approaches to improve both the generalizability and performance of the model, striking a balance between the two objectives.

**Data Collection and Curation**   We build our large-scale pre-training dataset by leveraging a large data pool of web images collected from public posts on Instagram. These images already went through platform-level content moderation to help prevent harmful contents and we obtain an initial data pool of approximately 17 billions of images. Using this raw data pool, we create two dataset *parts*. We construct the first part by applying the automatic curation method based on hierarchical $k$-means from Vo et al. (2024). We employ DINOv2 as image embeddings, and use 5 levels of clustering with the number of clusters from the lowest to highest levels being 200M, 8M, 800k, 100k, and 25k respectively. After building the hierarchy of clusters, we apply the balanced sampling algorithm proposed in Vo et al. (2024). This results in a curated subset of 1.7B images that guarantees a balanced coverage of all visual concepts appearing on the web. For the second part, we adopt a retrieval-based curation system similar to the procedure proposed by Oquab et al. (2024). We retrieve 28.5M images from the 17B data pool that are similar to those from selected seed datasets, creating a dataset that covers visual concepts relevant for downstream tasks. We provide the list of seed datasets and the corresponding number of retrieved images in App. D.2. Additionally, we integrate the raw publicly available computer vision datasets ImageNet1k (Deng et al., 2009), ImageNet22k (Russakovsky et al., 2015), and Mapillary Street-level Sequences (Warburg et al., 2020) in our final training dataset. We name our final dataset LVD-1689M.

Table 2: Comparison of the teacher architectures used in DINOv2 and DINOv3 models. We keep the model 40 blocks deep, and increase the embedding dimension to 4096. Importantly, we use a patch size of 16 pixels, changing the effective sequence length for a given resolution.

| Teacher model | DINOv2 | DINOv3 |
|---|---|---|
| Backbone | ViT-giant | ViT-7B |
| #Params | 1.1B | 6.7B |
| #Blocks | 40 | 40 |
| Patch Size | 14 | 16 |
| Pos. Embeddings | Learnable | RoPE |
| Registers | 4 | 4 |
| Embed. Dim. | 1536 | 4096 |
| FFN Type | SwiGLU | SwiGLU |
| FFN Hidden Dim. | 4096 | 8192 |
| Attn. Heads | 24 | 32 |
| Attn. Heads Dim. | 64 | 128 |
| DINO Head MLP | 4096-4096-256 | 8192-8192-512 |
| DINO Prototypes | 128k | 256k |
| iBOT Head MLP | 4096-4096-256 | 8192-8192-384 |
| iBOT Prototypes | 128k | 96k |

**Data Sampling**  During pre-training, we use a sampler to mix different data parts together. There are several different options for mixing the above data components. One is to train with *homogeneous* batches of data that come from a single, randomly selected component in each iteration. Alternatively, we can optimize the model on *heterogeneous* batches that are assembled by data from all components, selected using certain ratios. In each iteration, we randomly sample either a homogeneous batch from ImageNet1k alone or a heterogeneous batch mixing data from other parts (the curated parts from our data pool, ImageNet22k and Mapillary) following a multinomial distribution with weights specified in App. D.2. This is inspired by Charton & Kempe (2024), who observed it to be beneficial to use homogeneous batches consisting of very high quality data from a small dataset. In our training, homogeneous batches from ImageNet1k account for 10% of training. In App. C.2, we provide a comparison with using fully-heterogeneous or homogeneous batches, showing the benefit of our mixed setup.

**Comparison of Curation Methods**  In Tab. 1, we present a comparative analysis of several data curation strategies applied to our 17B data pool: random sampling, a clustering-based technique (Vo et al., 2024), a retrieval-based method (Oquab et al., 2024), and ours. For all methods except retrieval-based curation, we sample approximately 1.7B images; the retrieval-based pipeline of Oquab et al. (2024) yields a dataset of 139M images. We train a 7B-parameter model on each dataset and evaluate their performance on standard downstream tasks, using a shortened training schedule of 200k iterations for efficiency. Our results show that both clustering-based and retrieval-based strategies outperform random sampling, though neither consistently surpasses the other across all benchmarks. Notably, our final curation recipe achieves the best results on all considered benchmarks.

## 3.2 Large-Scale Training with Self-Supervision

While models trained with SSL have demonstrated interesting properties (Chen et al., 2020b; Caron et al., 2021), most SSL algorithms have not been scaled-up to larger models sizes. This is either due to issues with training stability (Darcet et al., 2025), or overly simplistic solutions that fail to capture the full complexity of the visual world. When trained at scale (Goyal et al., 2022a), models trained with SSL do not necessarily show impressive performance. One notable exception is DINOv2, a model with 1.1 billion parameters trained on curated data, matching the performance of weakly-supervised models like CLIP (Radford et al., 2021). A recent effort to scale DINOv2 to 7 billion parameters (Fan et al., 2025) demonstrates promising results on global tasks, but with disappointing results on dense prediction. Here, we aim to scale up the model and data, and obtain even more powerful visual representations with both improved global and local properties.

**Learning Objective**  We train the model with a discriminative self-supervised strategy which is a mix of several self-supervised objectives with both global and local loss terms. Following DINOv2 (Oquab et al., 2024), we use an image-level objective (Caron et al., 2021) $\mathcal{L}_{\text{DINO}}$, and balance it with a patch-level latent reconstruction objective (Zhou et al., 2021) $\mathcal{L}_{\text{iBOT}}$. We also replace the centering from DINO with the Sinkhorn-Knopp from SwAV (Caron et al., 2020) in both objectives. Each objective is computed using the output of a dedicated head on top of the backbone network, allowing for some specialization of features before the computation of the losses. Additionally, we use a dedicated layer normalization applied to the backbone outputs of the local and global crops. Empirically, we found this change to stabilize ImageNet kNN-classification late in training (+0.2 accuracy) and improve dense performance (*e.g.* +1 mIoU on ADE20k segmentation, -0.02 RMSE on NYUv2 depth estimation). In addition, a Koleo regularizer $\mathcal{L}_{\text{Koleo}}$ is added to encourage the features within a batch to spread uniformly in the space (Sablayrolles et al., 2018). We use a distributed implementation of Koleo in which the loss is applied in small batches of 16 samples—possibly across GPUs. Our initial training phase is carried by optimizing the following loss:

$$\mathcal{L}_{\text{Pre}} = \mathcal{L}_{\text{DINO}} + \mathcal{L}_{\text{iBOT}} + 0.1 * \mathcal{L}_{\text{DKoleo.}} \tag{1}$$

**Updated Model Architecture**  For the model scaling aspect of this work, we increase the size of the model to 7B parameters, and provide in Tab. 2 a comparison of the corresponding hyperparameters with the 1.1B parameter model trained in the DINOv2 work. We also employ a custom variant of RoPE: our base implementation assigns coordinates in a normalized $[-1, 1]$ box to each patch, then applies a bias in the multi-head attention operation depending on the relative position of two patches. In order to improve the robustness of the model to resolutions, scales and aspect ratios, we employ *RoPE box jittering*. For each sample and layer, the coordinate box $[-1, 1]$ is randomly scaled to $[-s, s]$, where $s \in [0.5, 2]$. We provide a comparison of standard RoPE and its jittering counterpart in App. C.3. Together, these changes enable DINOv3 to better learn detailed and robust visual features, improving its performance and scalability.

**Optimization**  Training large models on very large datasets represents a complicated experimental workflow. Because the interplay between model capacity and training data complexity is hard to assess *a priori*, it is impossible to guess the right optimization horizon. To overcome this, we get rid of all parameter scheduling, and train with constant learning rate, weight decay, and teacher EMA momentum. This has two main benefits. First, we can continue training as long as downstream performance continues to improve. Second, the number of optimization hyperparameters is reduced, making it easier to choose them properly. For the training to start properly, we still use a linear warmup for learning rate and teacher temperature. Following common practices, we use AdamW (Loshchilov & Hutter, 2017), and set the total batch size to 4096 images split across 256 GPUs. We train our models using the multi-crop strategy (Caron et al., 2020), taking 2 global crops and 8 local crops per image. We use square images with a side length of 256/112 pixels for global/local crops, which, along with the change in patch size, results in the same effective sequence length per image as in DINOv2 and a total sequence length of 3.7M tokens per batch. Additional hyperparameters can be found in App. D and in the code release; we ablate a few hyperparameters in App. C.4.

## 4 Gram Anchoring: A Regularization for Dense Features

To fully leverage the benefits of large-scale training, we aim to train the 7B model for an extended duration, with the notion that it could potentially train indefinitely. As expected, prolonged training leads to improvements on global benchmarks. However, as training progresses, the performance degrades on dense tasks (Figs. 5b and 5c). This phenomenon, which is due to the emergence of patch-level inconsistencies in feature representations, undermines the interest behind extended training.[1] In this section, we first analyze the loss of patch-level consistency, then propose a new objective to mitigate it, called *Gram anchoring*. We finally discuss the impact of our approach on both training stability and model performance.

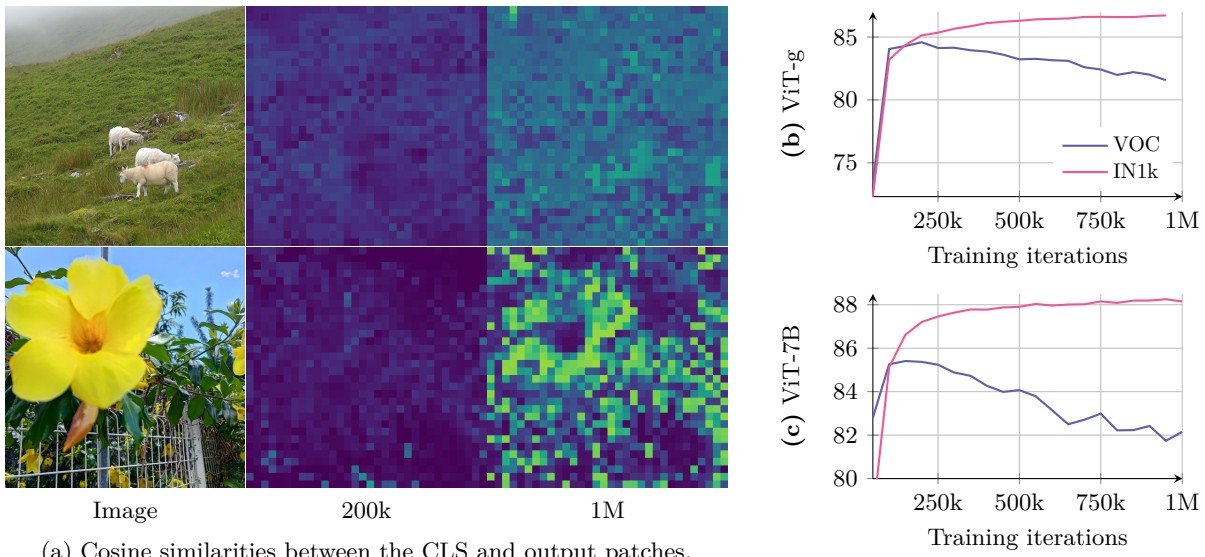

(a) Cosine similarities between the CLS and output patches.

Figure 5: Evolution of the cosine similarities (a) and of the accuracy on ImageNet1k linear (IN1k) and segmentation on VOC for ViT-g (b) and ViT-7B (c). We observe that the segmentation performance is maximal when the cosine similarities between the patch tokens and the class tokens are low. As training progresses, these similarities increase and the performance on dense tasks decreases.

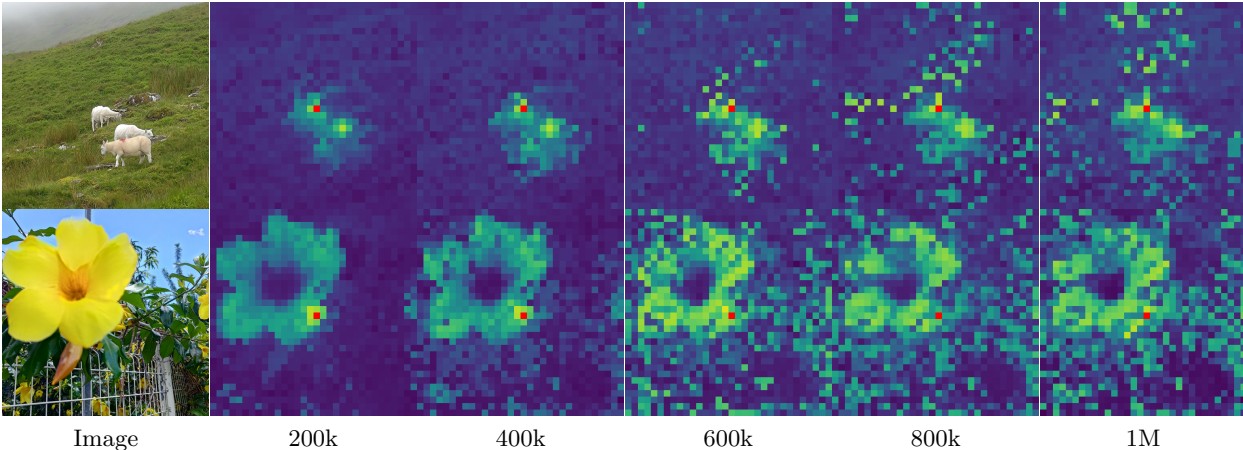

Figure 6: Evolution of the cosine similarity between the patch noted in red and all other patches. As training progresses, the features produced by the model become less localized and the similarity maps become noisier.

## 4.1 Loss of Patch-Level Consistency Over Training

During extended training, we observe consistent improvements in global metrics but a notable decline in performance on dense prediction tasks. This behavior was previously observed, to a lesser extent, during the training of DINOv2, and also discussed in the scaling effort of Fan et al. (2025). However, to the best of our knowledge, it remains unresolved to date. We illustrate the phenomenon in Figs. 5b and 5c, which present the performance of the model across iterations on both image classification and segmentation tasks. For classification, we train a linear classifier on ImageNet-1k using the CLS token and report top-1 accuracy. For segmentation, we train a linear layer on patch features extracted from Pascal VOC and report mean Intersection over Union (mIoU). We observe that both for the ViT-g and the ViT-7B, the classification

---

[1]We also observed different types of outliers appearing with continued training; we provide a discussion in App. B.

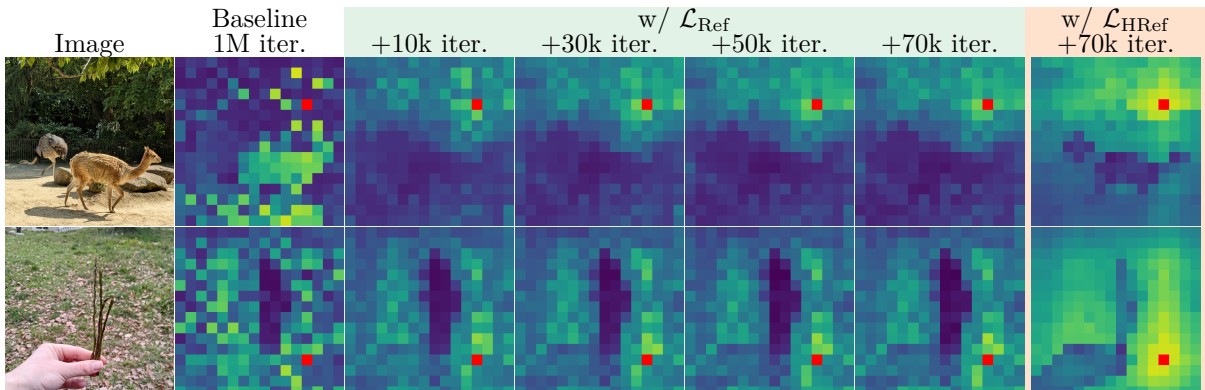

Figure 7: Effect of Gram anchoring (Sec. 4.2) on the cosine similarities of the patch features when using $\mathcal{L}_{\text{Ref}}$ after 1M iterations of the baseline model. The right column (orange) corresponds to using high-resolution features as presented in Sec. 4.3. In both cases, the Gram anchor model is the 200k checkpoint.

accuracy monotonically improves throughout training. However, segmentation performance declines in both cases after approximately 200k iterations, falling below its early levels in the case of the ViT-7B.

To better understand this degradation, we analyze the quality of patch features by visualizing cosine similarities between patches. Fig. 6 shows the cosine similarity maps between the backbone's output patch features and a reference patch (highlighted in red). At 200k iterations, the similarity maps are smooth and well-localized, indicating consistent patch-level representations. However, by 600k iterations and beyond, the maps degrade substantially, with an increasing number of irrelevant patches with high similarity to the reference patch. This loss of patch-level consistency correlates with the drop in dense task performance.

These patch-level irregularities differ from the high-norm patch outliers described in Darcet et al. (2024). Specifically, with the integration of register tokens, patch norms remain stable throughout training. However, we notice that the cosine similarity between the CLS token and the patch outputs gradually increases during training. This is expected, yet it means that the locality of the patch features diminishes. We visualize this phenomenon in Fig. 5a, which depicts the cosine maps at 200k and 1M iterations. In order to mitigate the drop on dense tasks, we propose a new objective specifically designed to regularize the patch features and ensure a good patch-level consistency, while preserving high global performance.

## 4.2 Gram Anchoring Objective

Throughout our experiments, we have identified a relative independence between learning strong discriminative features and maintaining local consistency, as observed in the lack of correlation between global and dense performance. While combining the global DINO loss with the local iBOT loss has begun to address this issue, we observe that the balance is unstable, with global representation dominating as training progresses. Building on this insight, we propose a novel solution that explicitly leverages this independence.

**The objective.** We introduce a new objective which mitigates the degradation of patch-level consistency by enforcing the quality of the patch-level consistency, without impacting the features themselves. This new loss function operates on the Gram matrix: the matrix of all pairwise dot products of patch features in an image. We want to push the Gram matrix of the student towards that of an earlier model, referred to as the *Gram teacher*. We select the Gram teacher by taking an early iteration of the teacher network, which exhibits superior dense properties. By operating on the Gram matrix rather than the feature themselves, the local features are free to move, provided the structure of similarities remains the same. Suppose that we have an image composed of $P$ patches, and a network that operates in dimension $d$. Let us denote by $\mathbf{X}_S$ (respectively $\mathbf{X}_G$) the $P \times d$ matrix of $\mathbf{L}_2$-normalized local features of the student (respectively the Gram teacher). We define the loss $\mathcal{L}_{\text{Gram}}$ as follows:

$$\mathcal{L}_{\text{Gram}} = \left\| \mathbf{X}_S \cdot \mathbf{X}_S^\top - \mathbf{X}_G \cdot \mathbf{X}_G^\top \right\|_{\text{F}}^2. \tag{2}$$

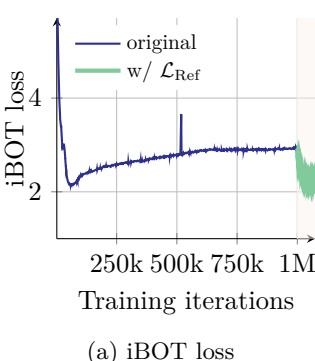 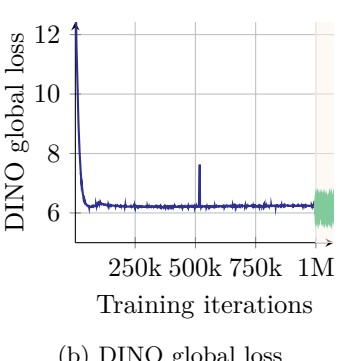 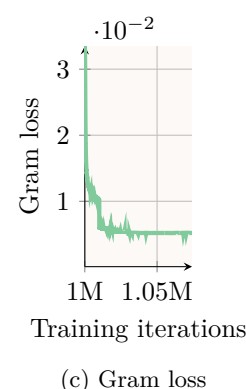

(a) iBOT loss        (b) DINO global loss        (c) Gram loss

Figure 8: Evolution trough the training iterations of the patch-level iBOT loss, the global loss DINO (applied to the global crops) and the newly introduced Gram loss. We highlight the iterations of the refinement step $\mathcal{L}_{\text{Ref}}$ which uses the Gram objective.

We only compute this loss on the global crops. We call this second step of training the *refinement step*, which optimizes the objective $\mathcal{L}_{\text{Ref}}$, with

$$\mathcal{L}_{\text{Ref}} = w_{\text{D}}\mathcal{L}_{\text{DINO}} + \mathcal{L}_{\text{iBOT}} + w_{\text{DK}}\mathcal{L}_{\text{DKoleo}} + w_{\text{Gram}}\mathcal{L}_{\text{Gram}}. \tag{3}$$

The Gram objective requires an additional forward pass through the Gram anchor model, *i.e.*, a third forward pass overall, which introduces some computational overhead. For efficiency, we therefore apply $\mathcal{L}_{\text{Gram}}$ as late as possible during training. Concretely, we observe that enabling the loss at 1M iterations—a point at which patch cosine similarities are already highly degraded (Fig. 6)—is sufficient to successfully "repair" the local features and restore well-structured patch similarities. However, we found that the loss can also be applied at an early training stage (*e.g.* after 200k iterations), in which case it stabilizes the patch features and prevents them from degrading in the first place. Finally, to further improve performance, we refresh the Gram teacher every 10k iterations, at which point the Gram teacher is reset to be identical to the main EMA teacher.

**Effect on patch features.** We include qualitative visualizations of the patch features before and after enabling the loss in Fig. 7. Adding the Gram objective has an immediate impact: only 10k iterations are enough to recover the desired similarity structure and restore patch-level consistency. More training iterations stabilize these rapid changes, further smoothing the patch similarities.

**Effect on losses.** We visualize the evolution of different losses in Fig. 8 and observe that applying the Gram objective significantly influences the iBOT loss, causing it to decrease more rapidly. This suggests that the stability introduced by the stable Gram teacher positively impacts the iBOT objective. In contrast, the Gram objective does not have a significant effect on the DINO losses. This observation implies that Gram and iBOT objectives impact the features in a similar way, whereas DINO losses affect them differently.

**Effect on performance.** Regarding performance, we find that the effect of the new loss on dense prediction tasks is almost immediate, consistent with the rapid improvement in patch cosine similarities shown in Fig. 7. As shown in Fig. 9, incorporating Gram anchoring leads to significant improvements on dense tasks within the first 10k iterations. We also see notable gains on the ADE20k benchmark following the Gram teacher updates. We observe similar gains on the dense depth task NYU in Tab. 3 (second row). Beyond early gains, longer training further improves performance on ObjectNet, while the impact on other global benchmarks remains comparatively modest.

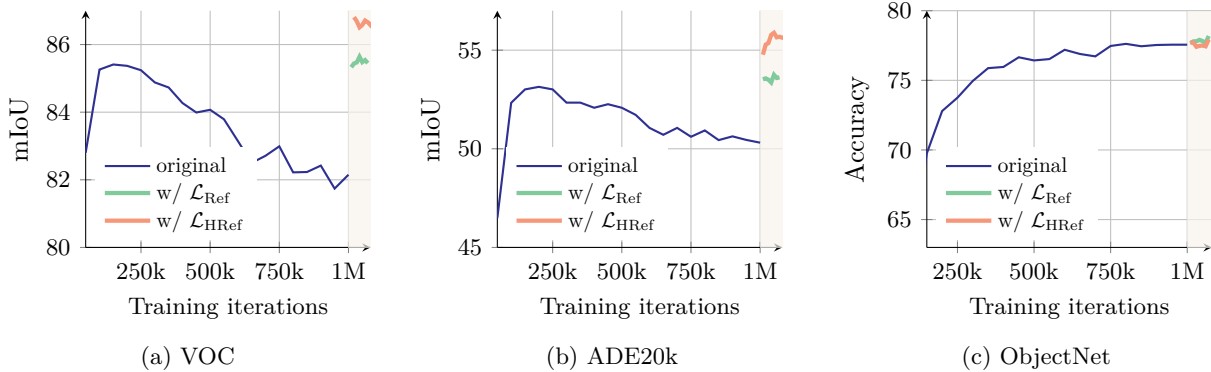

(a) VOC        (b) ADE20k        (c) ObjectNet

Figure 9: Evolution of the results on different benchmarks after applying our proposed *Gram anchoring* method. We visualize results when continuing the original training with our refinement step, noted '$\mathcal{L}_{\text{Ref}}$'. We also plot results obtained when using higher-resolution features for the Gram objective as introduced in following Sec. 4.3 and noted '$\mathcal{L}_{\text{HRef}}$'. We highlight the iterations which use the Gram objective.

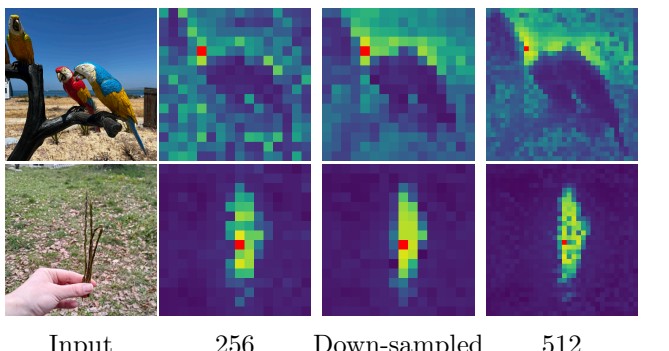

Input      256     Down-sampled     512

(a) Gram matrices at different input resolutions.

|  | Teacher | IN1k | ADE | NYU |
|---|---|---|---|---|
| Method | Iteration | Linear | mIoU | RMSE |
| Baseline | — | **88.2** | 50.3 | 0.307 |
| GRAM | 200k | 88.0 | **55.7** | **0.281** |
| GRAM | 100k | 87.9 | **55.7** | 0.284 |
| GRAM | 1M | 88.1 | 54.9 | 0.290 |

(b) Ablation of Gram teachers when using $\mathcal{L}_{\text{HRef}}$.

Figure 10: Quantitative and qualitative study of the impact of high-resolution Gram. We show (a) the improved cosine maps after down-sampling the high-resolution maps into smaller ones, and (b) the quantitative improvements brought by varying the training iteration of the Gram teacher when using $\mathcal{L}_{\text{HRef}}$.

## 4.3 Leveraging Higher-Resolution Features

Recent work shows that a weighted average of patch features can yield stronger local representations by smoothing outlier patches and enhancing patch-level consistency (Wysoczańska et al., 2024). On the other hand, feeding higher-resolution images into the backbone produces finer and more detailed feature maps. We leverage the benefits of both observations to compute high-quality features for the Gram teacher. Specifically, we first input images at twice the normal resolution into the Gram teacher, then 2× down-sample the resulting feature maps with the bicubic interpolation to achieve the desired smooth feature maps that match the size of the student output. Fig. 10a visualizes the Gram matrices of patch features obtained with images at resolutions 256 and 512, as well as those obtained after 2× down-sampling features from the 512-resolution (denoted as 'downsamp.'). We observe that the superior patch-level consistency in the higher-resolution features is preserved through down-sampling, resulting in smoother and more coherent patch-level representations. As a side note, our model can seamlessly process images at varying resolutions without requiring adaptation, thanks to the adoption of Rotary Positional Embeddings (RoPE) (Su et al., 2024).

**High-resolution refinement step.** To exploit the smoothed high-resolution features obtained via down-sampling, we compute their Gram matrix and substitute it for $\mathbf{X}_G$ in $\mathcal{L}_{\text{Gram}}$. We denote the resulting objective as $\mathcal{L}_{\text{HRef}}$. This approach enables the Gram objective to effectively distill the improved patch

Table 3: Results before and after our refinement step with and without high-resolution features $\mathcal{L}_{\text{Ref}}$ on classification and dense tasks.

| | Classification | | | Segmentation | | Depth |
|---|---|---|---|---|---|---|
| | IN1k k-nn | IN1k Linear | ObjectNet | ADE20k | VOC | NYU $\downarrow$ |
| Baseline (1M iter.) | **84.8** | **88.2** | 77.6 | 50.3 | 82.2 | 0.307 |
| w/ $\mathcal{L}_{\text{Ref}}$ (70k iter) | 84.7 | 88.0 | 78.2 | 53.6 | 85.5 | 0.285 |
| w/ $\mathcal{L}_{\text{HRef}}$ (70k iter) | **84.8** | 88.0 | **77.9** | **55.7** | **86.7** | **0.281** |

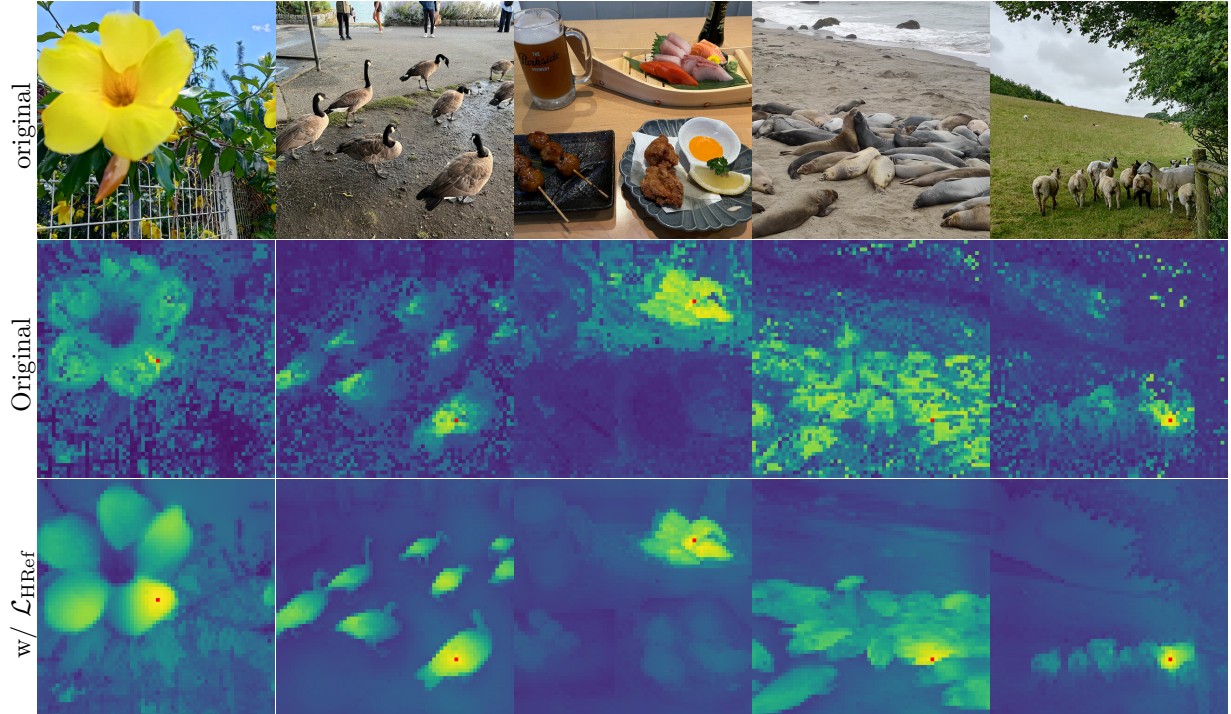

Figure 11: Qualitative effect of Gram anchoring. We visualize cosine maps before and after using the refinement objective $\mathcal{L}_{\text{HRef}}$. The input resolution of the images is $1024 \times 1024$ pixels.

consistency of smoothed high-resolution features into the student model. We visualize the difference when using $\mathcal{L}_{\text{Ref}}$ or $\mathcal{L}_{\text{HRef}}$ in Fig. 7 and observe that the cosine maps are denser and more accurate when using the smoothed high-resolution features. We quantitatively discuss the impact in Fig. 9 and Tab. 3; the distillation translates into better predictions on dense tasks, yielding significant additional gains on top of the benefit brought by $\mathcal{L}_{\text{Ref}}$ (+2 mIoU on ADE20k). We also ablate the choice of Gram teacher in Fig. 10b. Interestingly, choosing the Gram teacher from 100k or 200k does not significantly impact the results, but using a much later Gram teacher (1M iterations) is detrimental because the patch-level consistency of such a teacher is inferior.

Finally, we illustrate the effect of Gram anchoring to patch-level consistency in Fig. 11 which visualizes the Gram matrices patch features obtained with the initial training and high-resolution Gram anchoring refinement. We observe great improvements in feature correlations that our high-resolution refinement procedure brings about.

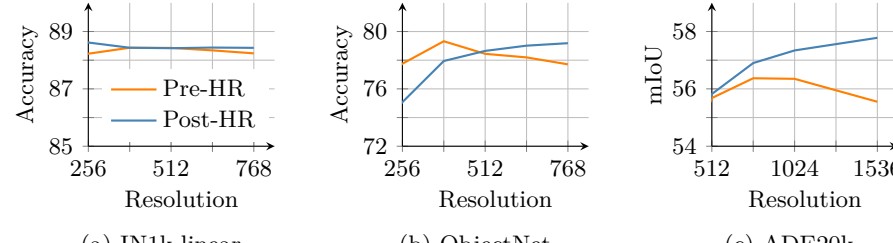

Figure 12: Effect of high resolution adaptation. Results before ('Pre-HR') and after ('Post-HR') resolution scaling (Sec. 5.1) on (a) linear classification on ImageNet, (b) applied OOD to ObjectNet, (c) linear semantic segmentation on ADE20k, and (d) segmentation tracking on DAVIS at different evaluation resolutions.

## 5    Post-Training

This section presents *post-training* stages. This includes a high-resolution adaptation phase enabling effective inference at different input resolutions (Sec. 5.1), model distillation producing quality and efficient smaller-sized models (Sec. 5.2), and text alignment adding zero-shot capabilities to DINOv3 (Sec. 5.3).

### 5.1    Resolution Scaling

We train our model at a relatively small resolution of 256, which gives us a good trade-off between speed and effectiveness. For a patch size of 16, this setup leads to the same input sequence length as DINOv2, which was trained with resolution 224 and patch size 14. However, many contemporary computer vision applications require processing images at significantly higher resolutions, often $512 \times 512$ pixels or greater, to capture intricate spatial information. The inference image resolution is also not fixed in practice and varies depending on specific use cases. To address this, we extend our training regime with a high-resolution adaptation step (Touvron et al., 2019). To ensure high performance across a range of resolutions, we utilize *mixed resolutions*, sampling differently-sized pairs of global and local crops per mini-batch. Specifically, we consider global crop sizes from $\{512, 768\}$ and local crop sizes from $\{112, 168, 224, 336\}$ and train the model for 10k additional iterations.

Similar to the main training, a key component of this high-resolution adaptation phase is the addition of Gram anchoring, using the 7B teacher as Gram teacher. We found this component to be essential: without it, the model performance on dense prediction tasks degrades significantly. The Gram anchoring encourages the model to maintain consistent and robust feature correlations across spatial locations, which is crucial when dealing with the increased complexity of high-resolution inputs.

Empirically, we observe that this relatively brief but targeted high-resolution step substantially enhances the overall model's quality and allows it to generalize across a wide range of input sizes, as shown visually in Fig. 4. In Fig. 12, we compare our 7B model before and after adaptation. We find that resolution scaling leads to a small gain on ImageNet classification (a) with relatively stable performance w.r.t. resolution. However, in ObjectNet OOD transfer (b), we observe that the performance tends to degrade slightly for lower resolutions, while improving for higher resolutions. This is largely compensated by the improvement in the quality of local features at high resolution, shown by the positive trend in segmentation on ADE20k (c) and tracking on DAVIS (d). Adaptation leads to local features that *improve with image size*, leveraging the richer spatial information available at larger resolutions and effectively enabling high-resolution inference. Interestingly, the adapted model supports resolutions way beyond the maximum training resolution of 768— we visually observe stable feature maps at resolutions above 4k (*c.f.* Fig. 4).

### 5.2    Model Distillation

**A Family of Models for Multiple Use-Cases**    We perform knowledge distillation of the ViT-7B model into smaller Vision Transformer variants (ViT-S, ViT-B, and ViT-L), which are highly valued by the community for their improved manageability and efficiency. Our distillation approach uses the same training

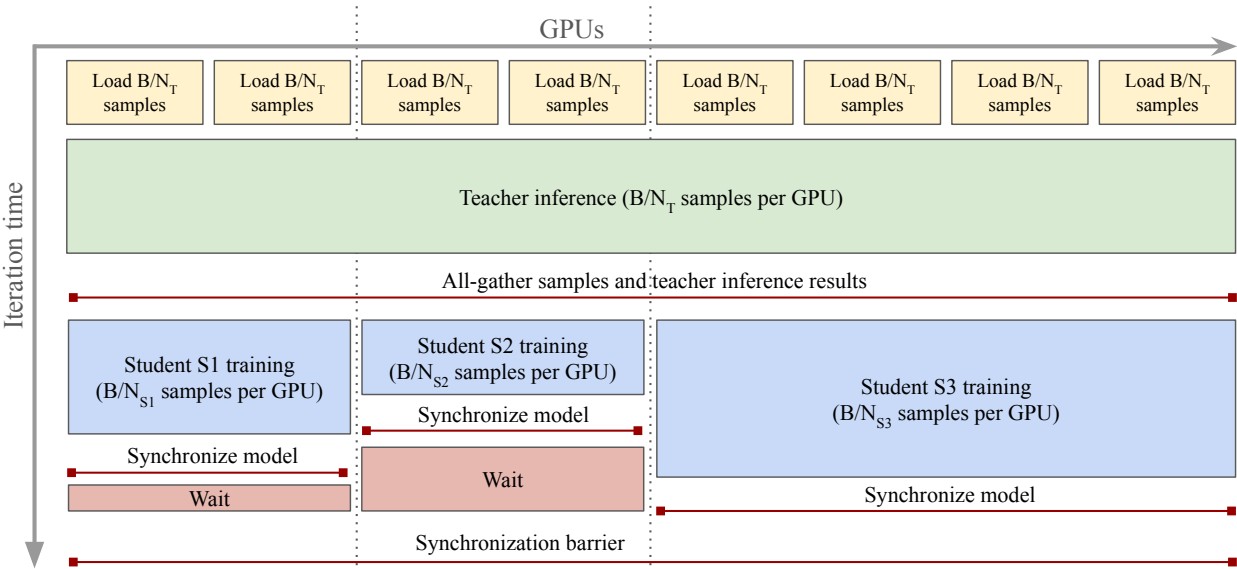

Figure 13: Multi-student distillation procedure. In this diagram, we distill 3 students in parallel: we first share teacher inference across all T nodes to save compute, and gather inputs and results on all GPUs. Then, smaller groups perform student training. We adjust the size of these groups such that the training step has the same duration across all students $Si$, minimizing idle time waiting at the synchronization barrier.

objective as in the first training phase, ensuring consistency in learning signals. However, instead of relying on an exponential moving average (EMA) of model weights, we use the 7B model directly as the teacher to guide the smaller student models. In this case, the teacher model is fixed. We do not observe patch-level consistency issues and therefore do not apply the Gram anchoring technique. This strategy enables the distilled models to inherit the rich representational power of the large teacher while being more practical for deployment and experimentation.

Our ViT-7B model is distilled into a series of ViT models with sizes covering a broad range of compute budgets, and allowing proper comparison with concurrent models. They include the standard ViT-S (21M params), B (86M), L (0.3B), along with a custom ViT-S+ (29M) and a custom ViT-H+ (0.8B) model to close the performance gap with the self-distilled 7B teacher model. Indeed, we observe in DINOv2 that smaller student models can reach a performance on par with their teacher as the distillation. As a result, the distilled models deliver frontier-level performance for a fraction of the inference compute as we see in Tab. 16. We train the models for 1M iterations then perform 250k iterations of learning rate cooldown following a cosine schedule before applying the high-resolution phase described in Sec. 5.1 above without Gram anchoring.

**Efficient Multi-Student Distillation** As the inference cost for a large teacher can be orders of magnitude higher than for students (see Fig. 16a), we design a parallel distillation pipeline that allows training multiple students at the same time and sharing the teacher inference across all nodes involved in the training (see Fig. 13 for a diagram). Let $C_T$ and $C_S$ be respectively the cost of running the teacher inference and the student training on a single sample, in single-teacher/single-student distillation with batch-size $B$ where each of the $N$ GPUs processes a $B/N$ slice of the data, the teacher inference costs $B/N \times C_T$ and the student training costs $B/N \times C_S$ per GPU. In multi-student distillation, we proceed as follows. Each student $Si$ is assigned a set of $N_{Si}$ GPUs for training, and all $N_T = \sum N_{Si}$ GPUs are part of the global inference group. At each iteration, we first run the teacher inference on the global group for a $B/N_T \times C_T$ compute cost per GPU. We then run an *all-gather* collective operation to share the input data and inference result with all compute nodes. Finally, each student group separately performs student training for a $B/N_{Si} \times C_{Si}$ cost.

The above calculations shows that adding an additional student to the distillation pipeline will (1) reduce the per-GPU compute at each iteration, thus globally improving distillation speed, and (2) increase the overall compute only by the training cost of the new student, since the total teacher inference cost is now fixed. The implementation only requires setting up GPU process groups carefully, adapting data-loaders and teacher inference to ensure inputs and outputs are synchronized across groups using NCCL collectives. As the groups are synchronized at each iteration, in order to maximize speed, we adapt the number of GPUs for each student such that their iteration times are roughly the same. With this procedure, we seamlessly train multiple students, and produce a whole family of distilled models from our flagship 7B model.

### 5.3   Aligning DINOv3 with Text

Open-vocabulary image-text alignment has received significant interest and enthusiasm from the research community, thanks to its potential to enable flexible and scalable multimodal understanding. A large body of work has focused on improving the quality of CLIP (Radford et al., 2021), which originally learned only a global alignment between image and text representations. While CLIP has demonstrated impressive zero-shot capabilities, its focus on global features limits its ability to capture fine-grained, localized correspondences. More recent works (Zhai et al., 2022b) have shown that effective image-text alignment can be achieved with pre-trained self-supervised visual backbones. This makes it possible to leverage these powerful models in multi-modal settings, facilitating richer and more precise text-to-image associations that extend beyond global semantics while also reducing computational costs, since the visual encoding is already learned.

We align a text encoder with our DINOv3 model by adopting the training strategy previously proposed in Jose et al. (2025). This approach follows the LiT training paradigm (Zhai et al., 2022b), training a text representation from scratch to match images to their captions with a contrastive objective, while keeping the vision encoder frozen. To allow for some flexibility on the vision side, two transformer layers are introduced on top of the frozen visual backbone. A key enhancement of this method is the concatenation of the mean-pooled patch embeddings with the output CLS token before matching to the text embeddings. This enables aligning both global and local visual features to text, leading to improved performance on dense prediction tasks without requiring additional heuristics or tricks. Furthermore, we use to the same data curation protocol as established in Jose et al. (2025) to ensure consistency and comparability.

## 6   Results

In this section, we evaluate our flagship DINOv3 7B model on a variety of computer vision tasks. Throughout our experiments, unless otherwise specified, *we keep DINOv3 frozen* and solely use its representations. We demonstrate that with DINOv3, finetuning is not necessary to obtain strong performance. This section is organized as follows. We first probe the quality of DINOv3's dense (Sec. 6.1) and global (Sec. 6.2) image representations using lightweight evaluation protocols and compare it to the strongest available vision encoders. We show that DINOv3 learns exceptional dense features while offering robust and versatile global image representations. Then, we consider DINOv3 as a basis for developing more complex computer vision systems (Sec. 6.3). We show with little effort on top of DINOv3, we are able to achieve results competitive with or exceeding the state of the art in tasks as diverse as object detection, semantic segmentation, 3D view estimation, or relative monocular depth estimation.

### 6.1   DINOv3 provides Exceptional Dense Features

We first investigate the raw quality of DINOv3's dense representations using a diverse set of lightweight evaluations. In all cases, we utilize the frozen patch features of the last layer, and evaluate them using (1) qualitative visualizations (Sec. 6.1.1), (2) dense linear probing (Sec. 6.1.2: segmentation, depth estimation), (3) non-parametric approaches (Sec. 6.1.3: 3D correspondence estimation, Sec. 6.1.4: object discovery, Sec. 6.1.5: tracking), and (4) lightweight attentive probing (Sec. 6.1.6: video classification).

**Baselines**   We compare the dense features of DINOv3 with those of the strongest publicly available image encoders, both weakly- and self-supervised ones. We consider the weakly-supervised encoders Perception

Table 4: Dense linear probing results on semantic segmentation and monocular depth estimation with frozen backbones. We report the mean Intersection-over-Union (mIoU) metric for the segmentation benchmarks ADE20k, Cityscapes, and VOC, and the Root Mean Squared Error (RMSE) metric for the depth benchmarks NYUv2 and KITTI.

| Method | ViT | Segmentation | | | Depth | |
|---|---|---|---|---|---|---|
| | | ADE20k | Citysc. | VOC | NYUv2 ↓ | KITTI ↓ |
| *Agglomerative backbones* | | | | | | |
| AM-RADIOv2.5 | g/14 | 53.0 | 78.4 | 85.4 | 0.340 | 2.918 |
| PEspatial | G/14 | 49.3 | 73.2 | 82.7 | 0.362 | 3.082 |
| *Weakly-supervised backbones* | | | | | | |
| SigLIP 2 | g/16 | 42.7 | 64.8 | 72.7 | 0.494 | 3.273 |
| PEcore | G/14 | 38.9 | 61.1 | 69.2 | 0.590 | 4.119 |
| *Self-supervised backbones* | | | | | | |
| Franca | g/14 | 46.3 | 68.7 | 82.9 | 0.445 | 3.140 |
| DINOv2 | g/14 | 49.5 | 75.6 | 83.1 | 0.372 | 2.624 |
| Web-DINO | 7B/14 | 42.7 | 68.3 | 76.1 | 0.466 | 3.158 |
| DINOv3 | 7B/16 | **55.9** | **81.1** | **86.6** | **0.309** | **2.346** |

Encoder (PE) Core (Bolya et al., 2025) and SigLIP 2 (Tschannen et al., 2025), which use CLIP-style image-text contrastive learning. We also compare to the strongest self-supervised methods: DINOv3's predecessor DINOv2 (Oquab et al., 2024) with registers (Darcet et al., 2024), Web-DINO (Fan et al., 2025), a recent scaling effort of DINO, and Franca (Venkataramanan et al., 2025), the best open-data SSL model. Finally, we compare to the agglomerative models AM-RADIOv2.5 (Heinrich et al., 2025), distilled from DINOv2, CLIP (Radford et al., 2021), DFN (Fang et al., 2024a), and Segment Anything (SAM) (Kirillov et al., 2023), and to PEspatial, distilling SAM 2 (Ravi et al., 2025) into PEcore. For each baseline, we report the performance of the strongest model available and specify the architecture in the tables.

### 6.1.1 Qualitative Analysis

We start by analyzing DINOv3's dense feature maps qualitatively. To this end, we project the dense feature space into 3 dimensions using principal component analysis (PCA), and map the resulting 3D space into RGB. Because of the sign ambiguity in PCA (eight variants) and the arbitrary mapping between principal components and colors (six variants), we explore all combinations and report the visually most compelling one. The resulting visualization is shown in Fig. 14. Compared to other vision backbones, it can be seen that the features of DINOv3 are sharper, containing much less noise, and showing superior semantical coherence.

### 6.1.2 Dense Linear Probing

We perform linear probing on top of the dense features for two tasks: semantic segmentation and monocular depth estimation. In both cases, we train a linear transform on top of the frozen patch outputs of DINOv3. For semantic segmentation, we evaluate on the ADE20k (Zhou et al., 2017), Cityscapes (Cordts et al., 2016), and PASCAL VOC 2012 (Everingham et al., 2012) datasets and report the mean intersection-over-union (mIoU) metric. For depth estimation, we use the NYUv2 (Silberman et al., 2012) and KITTI (Geiger et al., 2013) datasets and report the root mean squared error (RMSE). For segmentation, models are evaluated with input resolution adapted to 1024 tokens for ADE20k and VOC (*i.e.* $448 \times 448$ for patch size 14, $512 \times 512$ for patch size 16), and 4096 tokens for Cityscapes. For depth estimation, models are evaluated at the original image resolution. Please refer to Apps. E.1 and E.2 for more details.

**Results (Tab. 4)** The segmentation results demonstrate the superior quality of our dense features. On the general ADE20k dataset, DINOv3 outperforms the self-supervised baselines by more than 6 mIoU points, and the weakly supervised baselines by more than 13 points. Furthermore, DINOv3 surpasses PEspatial by more than 6 points, and AM-RADIOv2.5 by nearly 3 points. These results are remarkable as both are

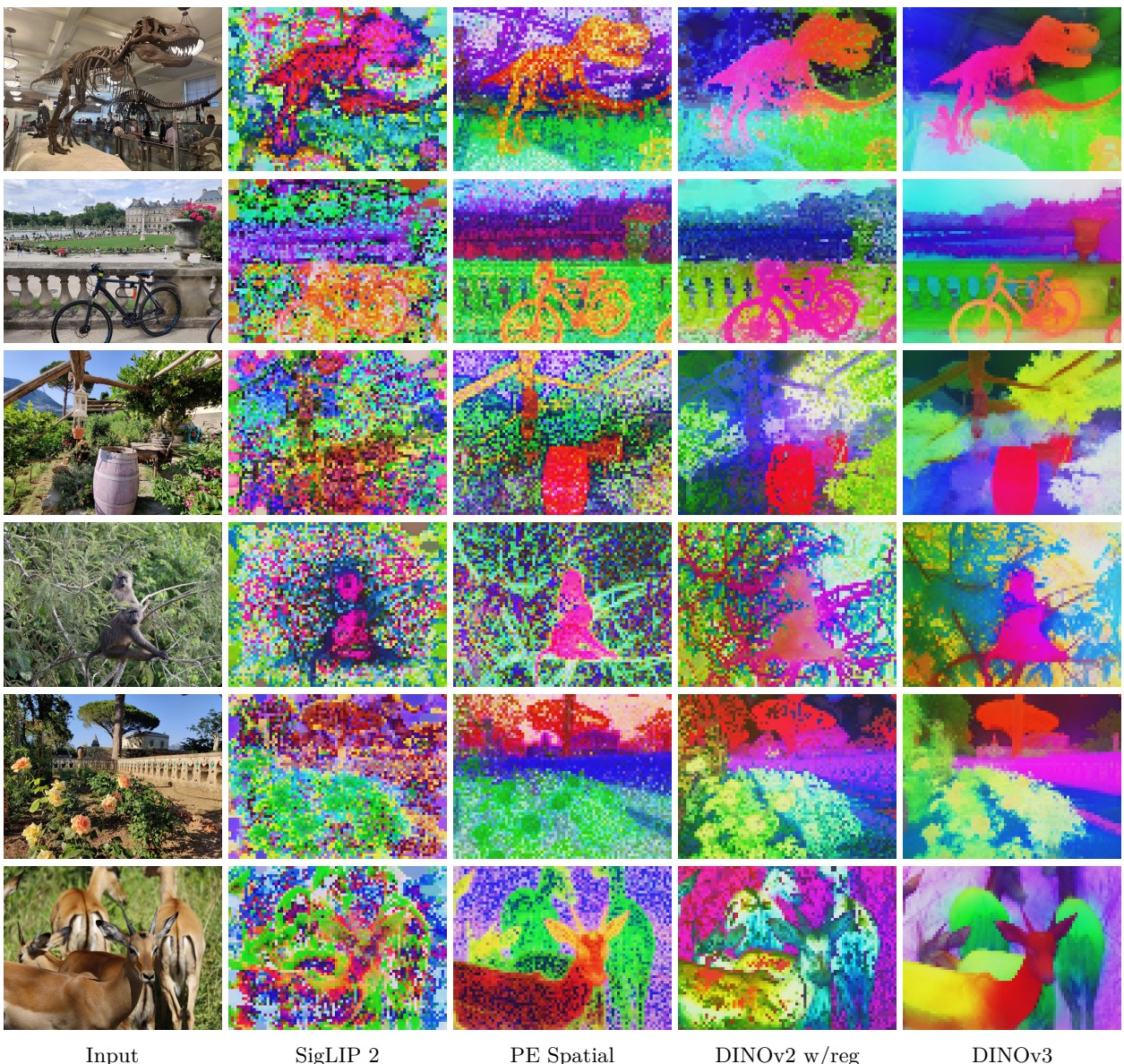

|  Input | SigLIP 2 | PE Spatial | DINOv2 w/reg | DINOv3 |

Figure 14: Comparison of dense features. We compare several vision backbones by projecting their dense outputs using PCA and mapping them to RGB. From left to right: SigLIP 2 ViT-g/16, PEspatial ViT-G/14, DINOv2 ViT-g/14 with registers, DINOv3 ViT-7B/16. Images are forwarded at resolution 1280×960 for models using patch 16 and 1120×840 for patch 14, *i.e.* all feature maps have size 80×60.

strong baselines, being distilled from the heavily supervised segmentation model SAM (Kirillov et al., 2023). Similar results are observed on the self-driving benchmark Cityscapes, with DINOv3 achieving the best mIoU of 81.1, surpassing AM-RADIOv2.5 by 2.5 points, and all other backbones by at least 5.5 points.

On monocular depth estimation, DINOv3 again outperforms all other models by significant margins: the weakly-supervised models PEcore and SigLIP 2 are still lagging, with DINOv2 and the more advanced models derived from SAM are the closest competitors. Interestingly, while PEspatial and AM-RADIO show strong performance on NYU, their performance is lower than DINOv2's on KITTI. Even there, DINOv3 outperforms its predecessor DINOv2 by 0.278 RMSE.

Table 5: Evaluation of 3D consistency of dense representations. We estimate 3D keypoint correspondences across views following the evaluation protocol of Probe3D (Banani et al., 2024). To measure performance, we report the correspondence recall, *i.e.* the percentage of correspondences falling into a specified distance.

| Method | ViT | Geometric NAVI | Semantic SPair |
|---|---|---|---|
| *Agglomerative backbones* | | | |
| AM-RADIOv2.5 | g/14 | 59.4 | 56.8 |
| PEspatial | G/14 | 53.8 | 49.6 |
| *Weakly-supervised backbones* | | | |
| SigLIP 2 | g/16 | 49.4 | 42.6 |
| PEcore | G/14 | 39.9 | 23.1 |
| *Self-supervised backbones* | | | |
| Franca | g/14 | 54.6 | 51.0 |
| DINOv2 | g/14 | 60.1 | 56.1 |
| Web-DINO | 7B/14 | 55.0 | 32.2 |
| DINOv3 | 7B/16 | **64.4** | **58.7** |

Both sets of evaluations show the outstanding representation power of the dense features of DINOv3 and reflect the visual results from Fig. 14. With only a linear predictor, DINOv3 allows robust prediction of object categories and masks, as well as physical measurements of the scene such as relative depth. These results show that the features are not only visually sharp and properly localized, they also represent many important properties of the underlying observations in a linearly separable way. Finally, the absolute performance obtained with a linear classifier on ADE20k (55.9 mIoU) is itself impressive, as it is not far from the absolute the state-of-the-art (63.0 mIoU) on this dataset.

### 6.1.3 3D Correspondence Estimation

Understanding the world in 3D has always been an important goal of computer vision Image foundation models have recently fueled research in 3D understanding by offering *3D-aware features*. In this section, we evaluate the *multi-view consistency* of DINOv3—that is, whether patch features of the same keypoint in different views of an object are similar—following the protocol defined in Probe3D (Banani et al., 2024). We distinguish between *geometric* and *semantic* correspondence estimation. The former refers to matching keypoints for the *same object instance* while the latter refers to matching keypoints for different instances of the *same object class*. We evaluate geometric correspondence on the NAVI dataset (Jampani et al., 2023) and semantic correspondence on the SPair dataset (Min et al., 2019), and measure performance with correspondence recall in both cases. Please refer to App. E.3 for more experimental details.

**Results (Tab. 5)** For geometric correspondences, DINOv3 outperforms all other models and improves over the second best model (DINOv2) by 4.3% recall. Other SSL scaling endeavors (Franca and WebSSL) lag behind DINOv2, showing that it is still a strong baseline. Weakly-supervised models (PEcore and SigLIP 2) do not fare well on this task, indicating a lack of 3D awareness. For models with SAM distillation, AM-RADIO nearly reaches the performance of DINOv2, but PEspatial still lags behind it ($-11.6\%$ recall), and even falls behind Franca ($-0.8\%$ recall). This suggests that self-supervised learning is a key component for strong performance on this task. For semantic correspondences, the same conclusions apply. DINOv3 performs best, outperforming both its predecessor ($+2.6\%$ recall) and AM-RADIO ($+1.9\%$ recall). Overall, these impressive performance on keypoint matching are very promising signals for downstream use of DINOv3 in other 3D-heavy applications.

### 6.1.4 Unsupervised Object Discovery

Powerful self-supervised features facilitate discovering object instances in images without requiring *any* annotations (Vo et al., 2021; Siméoni et al., 2021; Seitzer et al., 2023; Wang et al., 2023c; Siméoni et al., 2025). We

Table 6: Unsupervised object discovery. We apply TokenCut (Wang et al., 2022c) on the output patch features of different backbones and report the CorLoc metric. We also visualize predicted masks obtained with DINOv3 (red overlay on input images at res. 1024), obtained *without annotations and post-processing.*

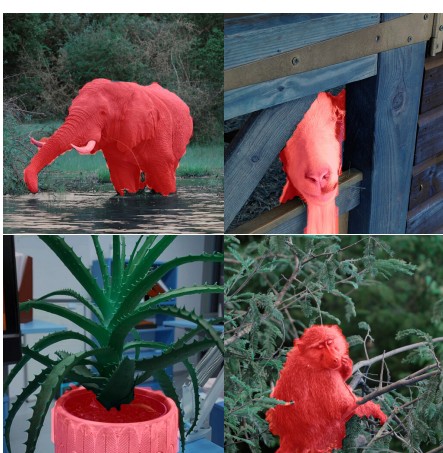

| Method | ViT | VOC07 | VOC12 | COCO |
|---|---|---|---|---|
| *Agglomerative backbones* | | | | |
| AM-RADIOv2.5 | g/14 | 55.0 | 59.7 | 45.9 |
| PEspatial | G/14 | 51.2 | 56.0 | 43.9 |
| *Weakly-supervised backbones* | | | | |
| SigLIPv2 | g/16 | 20.5 | 24.7 | 18.6 |
| PEcore | G/14 | 14.2 | 18.2 | 13.5 |
| *Self-supervised backbones* | | | | |
| DINO | S/16 | 61.1 | 66.0 | 48.7 |
| DINO | B/16 | 60.1 | 64.4 | 50.5 |
| DINOv2 | g/14 | 55.6 | 60.4 | 45.4 |
| Web-DINO | 7B/14 | 26.1 | 29.7 | 20.9 |
| DINOv3 | 7B/16 | **66.1** | **69.5** | **55.1** |

test this capability for different vision encoders via the task of unsupervised object discovery, which requires class-agnostic segmentation of objects in images (Russell et al., 2006; Tuytelaars et al., 2010; Cho et al., 2015; Vo et al., 2019). In particular, we use the non-parametric graph-based TokenCut algorithm (Wang et al., 2023c), which has shown strong performance on a variety of backbones. We run it on three widely used datasets: VOC 2007, VOC 2012 (Everingham et al., 2015), and COCO-20k (Lin et al., 2014; Vo et al., 2020). We follow the evaluation protocol defined by Siméoni et al. (2021) and report the CorLoc metric. To properly compare backbones with different feature distributions, we perform a search over the main Token-Cut hyperparameter, namely the cosine similarity threshold applied when constructing the patch graph used for partitioning. Originally, the best object discovery results were obtained with DINO (Caron et al., 2021) using the keys of the last attention layer. However, this hand-crafted choice does not consistently generalize to other backbones. For simplicity, we always employ the output features for all models.

**Results (Tab. 6)** The original DINO has set a very high bar for this task. Interestingly, while DINOv2 has shown very strong performance for pixel-wise dense tasks, it fails at object discovery. This can in part be attributed to the artifacts present in the dense features (*c.f.* Fig. 14). DINOv3, with its clean and precise output feature maps outperforms both its predecessors, with a 5.9 CorLoc improvement on VOC 2007, and all other backbones, whether self-, weakly-supervised or agglomerative. This evaluation confirms that DINOv3's dense features are both semantically strong and well localized. We believe that this will pave the way for more class-agnostic object detection approaches, especially in scenarios where annotations are costly or unavailable, and where the set of relevant classes is not confined to a predefined subset.

### 6.1.5 Video Segmentation Tracking

Beyond static images, an important property of visual representations is their *temporal consistency, i.e.* whether the features evolve in a stable manner through time. To test for this property, we evaluate DINOv3 on the task of video segmentation tracking: given ground-truth instance segmentation masks in the first frame of a video, the goal is to propagate these masks to subsequent frames. We use the DAVIS 2017 (Pont-Tuset et al., 2017), YouTube-VOS (Xu et al., 2018), and MOSE (Ding et al., 2023) datasets. We evaluate performance using the standard $\mathcal{J}\&\mathcal{F}$-mean metric, which combines region similarity ($\mathcal{J}$) and contour accuracy ($\mathcal{F}$) (Perazzi et al., 2016). Following Jabri et al. (2020), we use a non-parametric label propagation algorithm that considers the similarity between patch features across frames. We evaluate at three input resolutions, using a short side length of 420/480 (S), 840/960 (M), and 1260/1440 (L) pixels for models with patch size 14/16 (matching the number of patch tokens). The $\mathcal{J}\&\mathcal{F}$ score is always computed at the native resolution of the videos. See App. E.5 for more detailed experimental settings.

Table 7: Video segmentation tracking evaluation. We report the $\mathcal{J}\&\mathcal{F}$-mean on DAVIS, YouTube-VOS, and MOSE at multiple resolutions. For models with patch size 14/16, the small, medium and large resolutions correspond to a video short side of 420/480, 840/960, 1260/1440 pixels.

| Method | ViT | DAVIS | | | YouTube-VOS | | | MOSE | | |
|---|---|---|---|---|---|---|---|---|---|---|
| | | S | M | L | S | M | L | S | M | L |
| *Agglomerative backbones* | | | | | | | | | | |
| AM-RADIOv2.5 | g/14 | 66.5 | 77.3 | 81.4 | 70.1 | 78.1 | 79.2 | 44.0 | 52.6 | 54.3 |
| PEspatial | G/14 | 68.4 | 74.5 | 70.5 | 68.5 | 67.5 | 55.6 | 39.3 | 40.2 | 34.0 |
| *Weakly-supervised backbones* | | | | | | | | | | |
| SigLIP 2 | g/16 | 56.1 | 62.3 | 62.9 | 52.0 | 57.3 | 55.1 | 28.0 | 30.3 | 29.2 |
| PEcore | G/14 | 48.2 | 53.1 | 49.8 | 34.7 | 33.0 | 25.3 | 17.8 | 19.0 | 15.4 |
| *Self-supervised backbones* | | | | | | | | | | |
| Franca | g/14 | 61.8 | 66.9 | 66.5 | 67.3 | 70.5 | 67.9 | 40.3 | 42.6 | 41.9 |
| DINOv2 | g/14 | 63.9 | 73.6 | 76.6 | 65.6 | 73.5 | 74.6 | 40.4 | 47.6 | 48.5 |
| Web-DINO | 7B/14 | 57.2 | 65.8 | 69.5 | 43.9 | 49.6 | 50.9 | 24.9 | 29.9 | 31.1 |
| DINOv3 | 7B/16 | **71.1** | **79.7** | **83.3** | **74.1** | **80.2** | **80.7** | **46.0** | **53.9** | **55.6** |

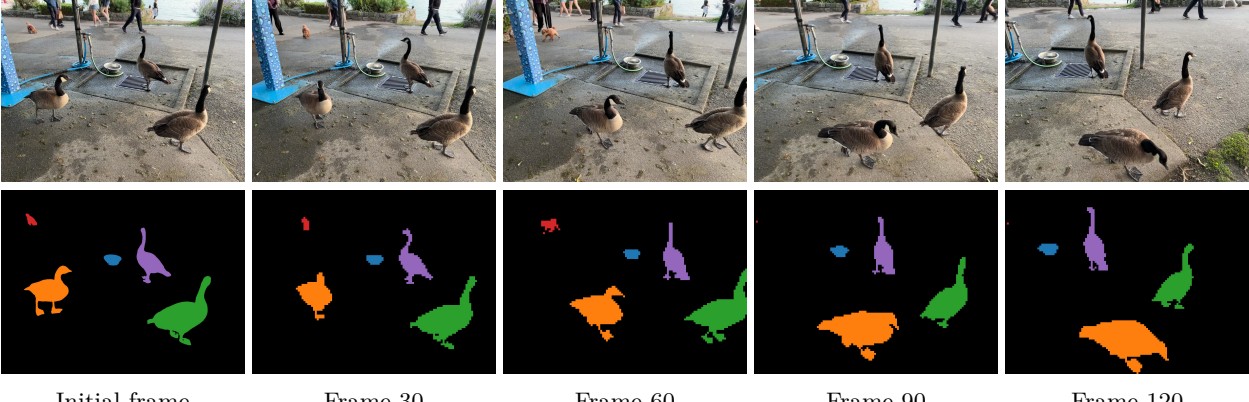

| Initial frame | Frame 30 | Frame 60 | Frame 90 | Frame 120 |

Figure 15: Segmentation tracking example. Given the ground-truth instance segmentation masks for the initial frame, we propagate the instance labels to subsequent frames according to patch similarity in the feature space of DINOv3. The input resolution is 2048×1536 pixels, resulting in 128×96 patches.

**Results (Tab. 7)** Aligned with all previous results, weakly-supervised backbones do not deliver convincing performance. PEspatial, distilled from the video model SAMv2, provides satisfactory performance, surpassing DINOv2 on smaller resolutions, but falling short on larger ones. Across resolutions, DINOv3 outperforms all competitors, with a staggering 83.3 $\mathcal{J}\&\mathcal{F}$ on DAVIS-L, 6.7 points above DINOv2. Furthermore, performance as a function of resolution follows a healthy trend, confirming that our model is able to make use of more input pixels to output precise, high-resolution feature maps (*c.f.* Figs. 3 and 4). In contrast, performance at higher resolutions stays almost flat for SigLIP 2 and PEcore, and degrades for PEspatial. Interestingly, our image model, without any tuning on video, allows to properly track objects in time (see Fig. 15). This makes it a great candidate to embed videos, allowing to build strong video models on top.

### 6.1.6 Video Classification

The previous results have shown the low-level temporal consistency of DINOv3's representations, allowing to accurately track objects in time. Going beyond, we evaluate in this section the suitability of its dense features for high-level video classification. Similar to the setup of V-JEPA 2 (Assran et al., 2025), we train an *attentive probe*—a shallow 4-layer transformer-based classifier—on top of patch features extracted from each frame.

Table 8: Video classification evaluation using attentive probes. We report top-1 accuracy on UCF101, Something-Something V2 (SSv2), and Kinetics-400 (K400). For each model, we report performance for evaluating a single clip per video, or applying test-time augmentation (TTA) by averaging the predicted probabilities from multiple clips.

| Method | ViT | UCF101 | | SSv2 | | K400 | |
|---|---|---|---|---|---|---|---|
| | | Single | TTA | Single | TTA | Single | TTA |
| *Agglomerative backbones* | | | | | | | |
| AM-RADIOv2.5 | g/14 | 92.8 | 92.5 | 69.1 | 70.0 | 84.8 | 85.2 |
| PEspatial | G/14 | 92.7 | 92.8 | 66.4 | 68.4 | 83.5 | 84.8 |
| *Weakly-supervised backbones* | | | | | | | |
| SigLIP 2 | g/16 | 93.6 | **94.2** | 68.8 | 70.2 | 86.9 | 87.7 |
| PEcore | G/14 | 93.1 | 93.3 | 69.0 | 70.4 | **87.9** | **88.8** |
| *Self-supervised backbones* | | | | | | | |
| DINOv2 | g/14 | 93.5 | 93.8 | 67.4 | 68.4 | 84.4 | 85.6 |
| V-JEPA 2 | g/16 | **94.0** | 93.8 | **73.8** | **75.4** | 83.3 | 84.3 |
| Web-DINO | 7B/14 | 93.9 | 94.1 | 67.3 | 68.1 | 86.8 | 87.2 |
| DINOv3 | 7B/16 | 93.5 | 93.5 | 70.1 | 70.8 | 87.8 | 88.2 |

This enables reasoning over temporal and spatial dimensions as the features are extracted independently per frame. During evaluation, we either take a single clip per video, or use test-time augmentation (TTA) by averaging the predictions of 3 spatial and 2 temporal crops per video. See App. E.6 for experimental details. We run this evaluation on three datasets: UCF101 (Soomro et al., 2012), Something-Something V2 (Goyal et al., 2017), and Kinetics-400 (Kay et al., 2017), and report top-1 accuracy. As an additional baseline, we report the performance of V-JEPA v2, a state-of-the-art SSL model for video understanding.

**Results (Tab. 8)** In line with the previous experiment, we find that DINOv3 can successfully be used to extract strong video features. Because this evaluation involves training multiple transformer blocks, differences between backbones are partially attenuated and the ranking is less pronounced. Nonetheless, DINOv3 performs competitively with PEcore and SigLIP 2, and clearly surpasses other baselines (e.g., DINOv2 and AM-RADIO) across datasets. UCF101 and K400 are largely appearance-driven benchmarks, where category-level object understanding accounts for most of the performance. In contrast, SSV2 emphasizes understanding fine-grained temporal dynamics; correspondingly, V-JEPA 2—pretrained specifically to capture motion and temporal structure—achieves stronger results than the image models. Interestingly, the margin between DINOv3 and weakly-supervised models is larger on SSV2, further supporting the suitability of DINOv3 for video-centric downstream tasks.

## 6.2 DINOv3 has Robust and Versatile Global Image Descriptors

In this section, we evaluate DINOv3's ability to capture global image statistics. To this end, we consider classic classification benchmarks using linear probes (Sec. 6.2.1) and instance retrieval benchmarks (Sec. 6.2.2). Again, we compare to the strongest publicly available image encoders. In addition to the models from the previous section, we evaluate the two weakly supervised models AIMv2 (Fini et al., 2024), trained using joint auto-regressive pixel and text prediction, and the massive EVA-CLIP-18B (Sun et al., 2024).

### 6.2.1 Image Classification with Linear Probing

We train a linear classifier on top of DINOv3's output CLS token to evaluate the model on classification benchmarks. We consider the ImageNet1k (Deng et al., 2009) dataset and its variants to evaluate out-of-distribution robustness, and a suite of datasets from different domains to understand DINOv3's ability to distinguish fine-grained classes. See App. E.7 for evaluation details.

Table 9: Classification accuracy of linear probes trained on ImageNet1k with frozen backbones. Weakly- and self-supervised models are evaluated with image resolution adapted to 1024 patch tokens (*i.e.* $448 \times 448$ for patch size 14, $512 \times 512$ for patch size 16). For reference, we also list results from Dehghani et al. (2023) using a different evaluation protocol (marked with $^*$).

| Method | ViT | ImageNet | | | Rendition | | Hard | | |
| --- | --- | --- | --- | --- | --- | --- | --- | --- | --- |
| | | Val | V2 | ReaL | R | S | A | C $\downarrow$ | Obj. |
| *Supervised backbones* | | | | | | | | | |
| Zhai et al. (2022a)* | G/14 | 89.0 | 81.3 | 90.6 | 91.7 | — | 78.8 | — | 69.6 |
| Chen et al. (2023)* | e/14 | 89.3 | 82.5 | 90.7 | 94.3 | — | 81.6 | — | 71.5 |
| Dehghani et al. (2023)* | 22B/14 | 89.5 | 83.2 | 90.9 | 94.3 | — | 83.8 | — | 74.3 |
| *Agglomerative backbones* | | | | | | | | | |
| AM-RADIOv2.5 | g/14 | 88.0 | 80.2 | 90.3 | 83.8 | 67.1 | 81.3 | 27.1 | 68.4 |
| *Weakly-supervised backbones* | | | | | | | | | |
| PEcore | G/14 | **89.3** | **81.6** | 90.4 | **92.2** | **71.9** | **89.0** | 22.7 | **80.2** |
| SigLIP 2 | g/16 | 89.1 | 81.6 | **90.5** | **92.2** | 71.8 | 84.6 | 30.0 | 78.6 |
| AIMv2 | 3B/14 | 87.9 | 79.5 | 89.7 | 82.3 | 67.1 | 74.5 | 29.5 | 69.0 |
| EVA-CLIP | 18B/14 | 87.9 | 79.3 | 89.5 | 85.2 | 64.0 | 81.6 | 33.0 | 71.9 |
| *Self-supervised backbones* | | | | | | | | | |
| Web-DINO | 7B/14 | 85.9 | 77.1 | 88.6 | 75.6 | 64.0 | 71.6 | 31.2 | 69.7 |
| Franca | g/14 | 84.8 | 75.3 | 89.2 | 67.6 | 49.5 | 56.5 | 40.0 | 54.5 |
| DINOv2 | g/14 | 87.3 | 79.5 | 89.9 | 81.1 | 65.4 | 81.7 | 24.1 | 66.4 |
| DINOv3 | 7B/16 | 88.4 | 81.4 | 90.4 | 91.1 | 71.3 | 86.9 | **19.6** | 79.0 |

**Domain Generalization from ImageNet (Tab. 9)** In this experiment, we train on ImageNet-*train*, use ImageNet-*val* as a *validation set* to select hyperparameters, and transfer the best found classifier to different test datasets: ImageNet-**V2** (Recht et al., 2019) and **ReaL** (Beyer et al., 2020) are alternative sets of images and labels for ImageNet, used to test overfitting on the ImageNet validation set; **R**endition (Hendrycks et al., 2021a) and **S**ketch (Wang et al., 2019) show stylized and artificial versions of the ImageNet classes; **A**dversarial (Hendrycks et al., 2021b) and **Obj**ectNet (Barbu et al., 2019) contain deliberately-chosen difficult examples; **C**orruptions (Hendrycks & Dietterich, 2019) measures robustness to common image corruptions. For reference, we also list linear probing results from Dehghani et al. (2023) for ViTs trained using supervised classification on the massive JFT dataset (3B–4B images). Note that these results follow a slightly different evaluation protocol and are not directly comparable to our results.

DINOv3 significantly surpasses all previous self-supervised backbones, with gains of +10% on ImageNet-R, +6% on -Sketch, +13% on ObjectNet over the previously strongest SSL model DINOv2. We note that the strongest weakly-supervised models, SigLIP 2 and PE, are now better than the strongest supervised ones (ViT-22B) on hard OOD tasks like ImageNet-A and ObjectNet. DINOv3 reaches comparable results on ImageNet-R and -Sketch, and, on the hard tasks ImageNet-A and ObjectNet, is closely behind PE, while exceeding SigLIPv2. On ImageNet, while validation scores are 0.7–0.9 points behind SigLIPv2 and PE, the performance on the "cleaner" test sets -V2 and -ReaL is virtually the same. Notably, DINOv3 achieves the best robustness to corruptions (ImageNet-C). All in all, *this is the first time that a SSL model has reached comparable results to weakly- and supervised models on image classification*—a domain which used to be the strong point of (weakly-)supervised training approaches. This is a remarkable result, given that models like ViT-22B, SigLIP 2, and PE are trained using massive human-annotated datasets. In contrast, DINOv3 learns purely from images, which makes it feasible to further scale/improve the approach in the future.

**Finegrained Classification (Tab. 10)** We also measure DINOv3's performance when training linear probes on several datasets for fine-grained classification. In particular, we report the accuracy on 3 large datasets, namely Places205 (Zhou et al., 2014) for scene recognition, and iNaturalist 2018 (Van Horn et al., 2018) and iNaturalist 2021 (Van Horn et al., 2021)) for detailed plant and animal-species recognition, as well

Table 10: Finegrained classification benchmarks. sFine averages over 12 datasets; see Tab. 27 for full results.

| Method | ViT | sFine | Places | iNat18 | iNat21 |
|---|---|---|---|---|---|
| *Agglomerative backbones* | | | | | |
| AM-RADIOv2.5 | g/14 | 93.9 | 70.2 | 79.0 | 83.7 |
| *Weakly-supervised backbones* | | | | | |
| SigLIP 2 | g/16 | 93.7 | 70.5 | 80.7 | 82.7 |
| PEcore | G/14 | **94.5** | **71.3** | **86.6** | 87.0 |
| AIMv2 | 3B/14 | 92.9 | 70.7 | 80.8 | 83.2 |
| EVA CLIP | 18B/14 | 92.9 | 71.1 | 80.7 | 83.5 |
| *Self-supervised backbones* | | | | | |
| Franca | g/14 | 87.7 | 64.6 | 61.4 | 70.6 |
| DINOv2 | g/14 | 92.6 | 68.2 | 80.7 | 86.1 |
| Web-DINO | 7B/14 | 90.2 | 69.6 | 65.3 | 74.1 |
| DINOv3 | 7B/16 | 93.0 | 70.0 | 85.6 | **89.8** |

Table 11: Instance recognition benchmarks. See Tab. 28 for additional metrics.

| Oxford-H | Paris-H | Met (GAP) | AmsterTime |
|---|---|---|---|
| 47.4 | 82.7 | 30.5 | 45.8 |
| | | | |
| 23.3 | 57.9 | 13.9 | 16.6 |
| 24.3 | 65.6 | 10.6 | 19.4 |
| 16.5 | 52.4 | 29.5 | 14.6 |
| 27.1 | 65.6 | 0.5 | 18.9 |
| | | | |
| 14.3 | 51.6 | 27.2 | 21.1 |
| 58.2 | 84.6 | 44.6 | 48.9 |
| 31.2 | 80.3 | 35.2 | 30.6 |
| **64.5** | **85.4** | **51.1** | **59.4** |

as the average over 12 smaller datasets covering scenes, objects, and textures (as in Oquab et al. (2024), here termed sFine). See also Tab. 27 for individual results on those datasets.

We find that, again, DINOv3 surpasses all previous SSL methods. It also shows competitive results compared to the weakly-supervised methods, indicating its robustness and generalization capability across diverse finegrained classification tasks. Notably, DINOv3 attains the highest accuracy on the difficult iNaturalist21 dataset at 89.8%, outperforming even the best weakly-supervised model PEcore with 87.0%.

### 6.2.2 Instance Recognition

To evaluate the instance-level recognition capabilities of our model, we adopted a non-parametric retrieval approach. Here, database images are ranked by their cosine similarity to a given query image, using the output CLS token. We benchmark performance across several datasets: the Oxford and Paris datasets for landmark recognition (Radenović et al., 2018), the Met dataset featuring artworks from the Metropolitan Museum (Ypsilantis et al., 2021), and AmsterTime, which consists of modern street view images matched to historical archival images of Amsterdam (Yildiz et al., 2022). Retrieval effectiveness is quantified using mean average precision for Oxford, Paris, and AmsterTime, and global average precision for Met. See App. E.9 for more evaluation details.

**Results (Tabs. 11 and 28)** Across all evaluated benchmarks, DINOv3 achieves the strongest performance by large margins, *e.g.* improving over the second best model DINOv2 by +10.8 points on Met and +7.6 points on AmsterTime. On this benchmark, weakly-supervised models are lagging far behind DINOv3, with the exception of AM-RADIO, which is distilled from DINOv2 features. These findings highlight the robustness and versatility of DINOv3 for instance-level retrieval tasks, spanning both traditional landmark datasets and more challenging domains such as art and historical image retrieval.

### 6.3 DINOv3 is a Foundation for Complex Computer Vision Systems

The previous two sections already provided solid signal for the quality of DINOv3 in both dense and global tasks. However, these results were obtained under "model probing" experimental protocols, using lightweight linear adapters or even non-parametric algorithms to assess the quality of features. While such simple evaluations allowed to remove confounding factors from involved experimental protocols, they are not enough to evaluate the full potential of DINOv3 as a foundational component in a larger computer vision system. Thus, in this section, we depart from the lightweight protocols, and instead train more involved downstream decoders and consider stronger, task-specific baselines. In particular, we use DINOv3 as a basis for (1) object detection with Plain-DETR (Sec. 6.3.1), (2) semantic segmentation with Mask2Former (Sec. 6.3.2),

Table 12: Comparison with state-of-the-art systems on object detection. We train a detection adapter on top of a *frozen* DINOv3 backbone. We show results on the validation set of the COCO and COCO-O datasets, and report the mAP across IoU thresholds, as well as the effective robustness (ER). Our detection system based on DINOv3 sets a new state of the art. As the InternImage-G detection model has not been released, we were unable to reproduce their results or compute COCO-O scores.

| Model | Detector | FT | Parameters | | | COCO | | COCO-O | |
| | | | Encoder | Decoder | Trainable | Simple | TTA | mAP | ER |
|---|---|---|---|---|---|---|---|---|---|
| EVA-02 | Cascade | 🔥 | 300M | — | 300M | 64.1 | — | 63.6 | 34.7 |
| InternImage-G | DINO | 🔥 | 6B | — | 6B | 65.1 | 65.3 | — | — |
| EVA-02 | Co-DETR | 🔥 | 300M | — | 300M | 65.4 | 65.9 | 63.7 | 34.3 |
| PEspatial | DETA | 🔥 | 1.9B | 50M | 2B | 65.3 | 66.0 | 64.0 | 34.7 |
| DINOv3 | Plain-DETR | ❄ | 7B | 100M | 100M | **65.6** | **66.1** | **66.4** | **36.8** |

(3) monocular depth estimation with Depth Anything (Sec. 6.3.3), and (4) 3D understanding with the Visual Geometry Grounded Transformer (Sec. 6.3.4). These tasks are only intended as explorations for what is possible with DINOv3. Still, we find that building on DINOv3 unlocks competitive or even state-of-the-art results with little effort.

### 6.3.1 Object Detection

As a first task, we tackle the long-standing computer vision problem of object detection. Given an image, the goal is to provide bounding boxes for all instances of objects of pre-defined categories. This task requires both precise localization and good recognition, as boxes need to match the object boundaries and correspond to the correct category. While performance on standard benchmarks like COCO (Lin et al., 2014) is mostly saturated, we propose to tackle this task with a *frozen* backbone, only training a small decoder on top.

**Datasets and Metrics**   We evaluate DINOv3 on object detection capabilities with the COCO dataset (Lin et al., 2014), reporting results on the COCO-VAL2017 split. Additionally, we evaluate out-of-distribution performance on the COCO-O evaluation dataset (Mao et al., 2023). This dataset contains the same classes but provides input images under six distribution shift settings. For both datasets, we report mean Average Precision (mAP) with IoU thresholds in $[0.5 : 0.05 : 0.95]$. For COCO-O, we additionally report the effective robustness (ER). Since COCO is a small dataset, comprising only 118k training images, we leverage the larger Objects365 dataset (Shao et al., 2019) for pre-training the decoder, as is common practice.

**Implementation**   We build upon the Plain-DETR (Lin et al., 2023b), but make the following modification: We do not fuse the transformer encoder into the backbone, but keep it as a separate module, similar to the original DETR (Carion et al., 2020), which allows us to keep the DINOv3 backbone completely frozen during training and inference. To the best of our knowledge, this makes it *the first competitive detection model to use a frozen backbone.* We train the Plain-DETR detector on Objects365 for 22 epochs at resolution 1536, then one epoch at resolution 2048, followed by 12 epochs on COCO at resolution 2048. At inference time, we run at resolution 2048. Optionally, we also apply test-time augmentation (TTA) by forwarding the image at multiple resolutions (from 1536 to 2880). See App. E.10 for full experimental details.

**Results (Tab. 12)**   We compare our system with four models: EVA-02 with a Cascade detector (Fang et al., 2024b), EVA-02 with Co-DETR (Zong et al., 2023), InternImage-G with DINO (Wang et al., 2023b), and PEspatial with DETA (Bolya et al., 2025). We find that our lightweight detector (100M parameters) trained on top of a frozen DINOv3 backbone manages to reach state-of-the-art performance. For COCO-O, the gap is pronounced, showing that the detection model can effectively leverage the robustness of the DINOv3. Interestingly, our model outperforms all previous models with much fewer trained parameters, with the smallest comparison point still using more than 300M trainable parameters. We argue that achieving

Table 13: Comparison with state-of-the-art systems for semantic segmentation on ADE20k. We evaluate the model in a single- or multi-scale setup (respectively Simple and TTA). Following common practice, we run this evaluation at resolution 896 and report mIoU scores. BEIT3, ONE-PEACE and DINOv3 use a Mask2Former with ViT-Adapter architecture, and the decoder parameters take into account both. We report results on further datasets in Tab. 29.

| Model | FT | Parameters | | | mIoU | |
|---|---|---|---|---|---|---|
| | | Encoder | Decoder | Trainable | Simple | TTA |
| BEIT3 | 🔥 | 1.0B | 550M | 1.6B | 62.0 | 62.8 |
| InternImage-H | 🔥 | 1.1B | 230M | 1.3B | 62.5 | 62.9 |
| ONE-PEACE | 🔥 | 1.5B | 710M | 2.2B | 62.0 | **63.0** |
| DINOv3 | ❄ | 7B | 927M | 927M | **62.6** | **63.0** |

such strong performance without specializing the backbone is an enabler for various practical applications: A single backbone forward can provide features that support multiple tasks, reducing compute requirements.

### 6.3.2 Semantic Segmentation

Following the previous experiment, we now evaluate on semantic segmentation, another long-standing computer vision problem. This task also requires strong, well localized representations, and expects a dense per-pixel prediction. However, opposed to object detection, the model does not need to differentiate instances of the same object. Similar to detection, we train a decoder on top of a *frozen* DINOv3 model.

**Datasets and Metrics**   We focus our evaluation on the ADE20k dataset (Zhou et al., 2017), which contains 150 semantic categories across 20k training images and 2k validation images. We measure performance using the mean Intersection over Union (mIoU). To train the segmentation model, we additionally use the COCO-Stuff (Caesar et al., 2018) and Hypersim (Roberts et al., 2021) datasets. Those contain 164k images with 171 semantic categories, and 77k images with 40 categories respectively.

**Implementation**   To build a decoder that maps DINOv3 features to semantic categories, we combine ViT-Adapter (Chen et al., 2022) and Mask2Former (Cheng et al., 2022), similar to prior work (Wang et al., 2022b; 2023b;a). However, in our case, the DINOv3 backbone remains frozen during training. In order to avoid altering the backbone features, we further modify the original ViT-Adapter architecture by removing the injector component. Compared to baselines, we also increase the embedding dimensions from 1024 to 2048, to support processing the 4096-dimensional output of the DINOv3 backbone. We start by pre-training the segmentation decoder on COCO-Stuff for 80k iterations, followed by 10k iterations on Hypersim (Roberts et al., 2021). Finally, we train for 20k iterations on the training split of ADE20k and report results on the validation split. All training is done at an input resolution of 896. At inference time we consider two setups: single-scale, *i.e.* we forward images at training resolution, or multi-scale, *i.e.* we average predictions at multiple image ratios between ×0.9 and 1.1 the original training resolution. We refer to App. E.11 for more experimental details.

**Results (Tab. 13)**   We compare our model's performance with several state-of-the-art baselines, including BEIT-3 (Wang et al., 2022b), InternImage-H (Wang et al., 2023b) and ONE-PEACE (Wang et al., 2023a), and report results on additional datasets in Tab. 29. Our segmentation model based on the frozen DINOv3 backbone reaches state-of-the-art performance, equaling that of ONE-PEACE (63.0 mIoU). It also improves over all prior models on the COCO-Stuff (Caesar et al., 2018) and VOC 2012 (Everingham et al., 2012) datasets. As semantic segmentation requires accurate per-pixel predictions, vision transformer backbones pose a fundamental problem. Indeed, the 16 pixel-wide input patches make the granularity of the prediction relatively coarse—encouraging solutions like ViT-Adapter. On the other hand, we have shown that we can obtain high-quality feature maps, even at very high resolutions up to 4096 (*c.f.* Figs. 3 and 4); this corresponds to dense feature maps 512-tokens wide. We hope that future work will be able to leverage these

Table 14: Comparison with state-of-the-art systems for relative monocular depth estimation. By combining DINOv3 with Depth Anything V2 (Yang et al., 2024b), we obtain a SotA model for relative depth estimation.

| Method | FT | NYUv2 | | KITTI | | ETH3D | | ScanNet | | DIODE | |
|---|---|---|---|---|---|---|---|---|---|---|---|
| | | ARel ↓ | $\delta_1$ ↑ | ARel ↓ | $\delta_1$ ↑ | ARel ↓ | $\delta_1$ ↑ | ARel ↓ | $\delta_1$ ↑ | ARel ↓ | $\delta_1$ ↑ |
| MiDaS | 🔥 | 11.1 | 88.5 | 23.6 | 63.0 | 18.4 | 75.2 | 12.1 | 84.6 | 33.2 | 71.5 |
| LeReS | 🔥 | 9.0 | 91.6 | 14.9 | 78.4 | 17.1 | 77.7 | 9.1 | 91.7 | 27.1 | 76.6 |
| Omnidata | 🔥 | 7.4 | 94.5 | 14.9 | 83.5 | 16.6 | 77.8 | 7.5 | 93.6 | 33.9 | 74.2 |
| DPT | 🔥 | 9.8 | 90.3 | 10.0 | 90.1 | 7.8 | 94.6 | 8.2 | 93.4 | **18.2** | 75.8 |
| Marigold | 🔥 | 5.5 | 96.4 | 9.9 | 91.6 | 6.5 | 96.0 | 6.4 | 95.1 | 30.8 | 77.3 |
| DAv2 (ViT-g) | 🔥 | 4.4 | 97.9 | 7.5 | 94.7 | 13.1 | 86.5 | — | — | — | — |
| DINOv3 | ❄️ | **4.3** | **98.0** | **7.3** | **96.7** | **5.4** | **97.5** | **4.4** | **98.1** | 25.6 | **82.2** |

high-resolution features to reach state-of-the-art performance without having to rely on heavy decoders like ViT-Adapter with Mask2Former.

### 6.3.3 Monocular Depth Estimation

We now consider building a system for monocular depth estimation. To do so, we follow the setup of Depth Anything V2 (DAv2) (Yang et al., 2024b), a recent state-of-the-art method. The key innovation of DAv2 is to use a large collection of synthetically generated images with ground truth depth annotations. Critically, this relies on DINOv2 as a feature extractor that is able to bridge the *sim-to-real* gap, a capability that other vision backbones like SAM (Kirillov et al., 2023) do not show (Yang et al., 2024b). Thus, we swap DINOv2 with DINOv3 in the DAv2 pipeline to see if we can achieve similar results.

**Implementation** Like DAv2, we use a Dense Prediction Transformer (DPT) (Ranftl et al., 2021) to predict a pixelwise depth field, using features from four equally spaced layers of DINOv3 as input. We train the model using the set of losses from DAv2 on DAv2's synthetic dataset, increasing the training resolution to $1024 \times 768$ to make use of DINOv3's high resolution capabilities. In contrast to DAv2, we *keep the backbone frozen* instead of finetuning it, testing the out-of-the-box capabilities of DINOv3. We also found it beneficial to scale up the DPT head to obtain the full potential DINOv3 7B's larger features. See App. E.12 for details.

**Datasets and Metrics** We evaluate our model on 5 real-world datasets (NYUv2 (Silberman et al., 2012), KITTI (Geiger et al., 2013), ETH3D (Schöps et al., 2017), ScanNet (from Ke et al. (2025)) and DIODE (Vasiljevic et al., 2019)) in the zero-shot scale-invariant depth setup, similar to Ranftl et al. (2020); Ke et al. (2025); Yang et al. (2024b). We report the standard metrics absolute relative error (ARel) (lower is better) and $\delta_1$ (higher is better). We refer to Yang et al. (2024a) for a description of those metrics.

**Results (Tab. 14)** We compare to the state of the art for relative depth estimation: MiDaS (Ranftl et al., 2020), LeReS (Yin et al., 2021), Omnidata (Eftekhar et al., 2021), DPT (Ranftl et al., 2021), Marigold in the ensemble version (Ke et al., 2025) and DAv2. Our depth estimation model reaches a new state-of-the-art on all datasets, only lacking behind in ARel on DIODE compared to DPT. Remarkably, this is possible using a *frozen backbone*, whereas all other baselines need to finetune the backbone for depth estimation. In addition, this validates that DINOv3 inherits DINOv2's *strong sim-to-real capabilities*, a desirable property that opens up the possibility for downstream tasks to use synthetically generated training data.

### 6.3.4 Visual Geometry Grounded Transformer with DINOv3

Finally, we consider 3D understanding with the recent Visual Geometry Grounded Transformer (VGGT) (Wang et al., 2025). Trained on a large set of 3D-annotated data, VGGT learns to estimate all important 3D attributes of a scene, such as camera intrinsics and extrinsics, point maps, or depth maps,

Table 15: 3D understanding using Visual Geometry Grounded Transformer (VGGT) (Wang et al., 2025). Simply by swapping DINOv2 for DINOv3 ViT-L as the image feature extractor in the VGGT pipeline, we are able to obtain state-of-the-art results on various 3D geometry tasks. We reproduce baseline results from Wang et al. (2025). We also report methods using ground truth camera information, marked with *. Camera pose estimation results are reported with AUC@30.

(a) Camera pose estimation.

| Method | Re10K | CO3Dv2 |
|---|---|---|
| DUSt3R | 67.7 | 76.7 |
| MASt3R | 76.4 | 81.8 |
| VG GSfM v2 | 78.9 | 83.4 |
| CUT3R | 75.3 | 82.8 |
| FLARE | 78.8 | 83.3 |
| VGGT | 85.3 | 88.2 |
| DINOv3 | **86.3** | **89.6** |

(b) Multi-view estimation on DTU.

| Method | Acc.↓ | Comp.↓ | Overall↓ |
|---|---|---|---|
| Gipuma* | 0.283 | 0.873 | 0.578 |
| CIDER* | 0.417 | 0.437 | 0.427 |
| MASt3R* | 0.403 | 0.344 | 0.374 |
| GeoMVSNet* | 0.331 | 0.259 | 0.295 |
| DUSt3R | 2.677 | 0.805 | 1.741 |
| VGGT | 0.389 | 0.374 | 0.382 |
| DINOv3 | **0.375** | **0.361** | **0.368** |

(c) View matching on ScanNet-1500.

| Method | AUC@5 | AUC@10 |
|---|---|---|
| SuperGlue | 16.2 | 33.8 |
| LoFTR | 22.1 | 40.8 |
| DKM | 29.4 | 50.7 |
| CasMTR | 27.1 | 47.0 |
| Roma | 31.8 | 53.4 |
| VGGT | 33.9 | 55.2 |
| DINOv3 | **35.2** | **56.1** |

in a single forward pass. Using a simple, unified pipeline, it reaches state-of-the-art results on many 3D tasks while being more efficient than specialized methods—constituting a major advance in 3D understanding.

**Implementation**   VGGT uses a DINOv2-pretrained backbone to obtain representations for different views of a scene, before fusing them with a transformer. Here, we simply swap the DINOv2 backbone with DINOv3, using our ViT-L variant (see Sec. 7) to match DINOv2 ViT-L/14 in the original work. We run the same training pipeline as VGGT, including finetuning of the image backbone. We switch the image resolution from $518 \times 518$ to $592 \times 592$ to accommodate DINOv3's patch size 16 and keep the the results comparable to VGGT. We additionally adopt a small number of hyperparameter changes detailed in App. E.13.

**Datasets and Metrics**   Following Wang et al. (2025), we evaluate on camera pose estimation on the Re10K (Zhou et al., 2018) and CO3Dv2 (Reizenstein et al., 2021) datasets, dense multi-view estimation on DTU (Jensen et al., 2014), and two-view matching on ScanNet-1500 (Dai et al., 2017). For camera pose estimation and two-view matching, we report the standard area-under-curve (AUC) metric. For multi-view estimation, we report the smallest L2-distance between prediction to ground truth as "Accuracy", the smallest L2-distance from ground truth to prediction as "Completeness" and their average as 'Overall'. We refer to Wang et al. (2025) for details about method and evaluation.

**Results (Tab. 15)**   We find that VGGT equipped with DINOv3 *further improves over the previous state-of-the-art* set by VGGT on all three considered tasks—using DINOv3 leads to clear and consistent gains. This is encouraging, given that we only applied minimal tuning for DINOv3. These tasks span different levels of visual understanding: high-level abstraction of scene content (camera pose estimation), dense geometric prediction (multi-view depth estimation), and fine-grained pixel-level correspondence (view matching). Together with the previous results on correspondence estimation (Sec. 6.1.3) and depth estimation (Sec. 6.3.3), we take this as further empirical evidence for the strong suitability of DINOv3 as a basis for 3D tasks. Additionally, we anticipate further improvements from using the larger DINOv3 7B model.

## 7   Evaluating the Full Family of DINOv3 Models

In this section, we provide quantitative evaluations on the family of models distilled from our 7B-parameters model (See Sec. 5.2). This family includes variants based on the Vision Transformer (ViT) and the ConvNeXt (CNX) architectures. We provide the detailed parameter counts and inference FLOPs for all models in Fig. 16a. These models cover a wide range of computational budgets to accommodate a broad spectrum of users and deployment scenarios. We conduct a thorough evaluation of all ViT (Sec. 7.1) and ConvNeXt variants (Sec. 7.2) to assess their performance across tasks. For both, we also provide an extended set of results in Tabs. 36 and 37.

| Model | #Params | Inference GFLOPs | |
| | | Res. 256 | Res. 512 |
| --- | --- | --- | --- |
| CNX-Tiny | 29M | 5 | 20 |
| CNX-Small | 50M | 11 | 46 |
| CNX-Base | 89M | 20 | 81 |
| CNX-Large | 198M | 38 | 152 |
| ViT-S | 21M | 12 | 63 |
| ViT-S+ | 29M | 16 | 79 |
| ViT-B | 86M | 47 | 216 |
| ViT-L | 300M | 163 | 721 |
| ViT-H+ | 840M | 450 | 1903 |
| ViT-7B | 6716M | 3550 | 14515 |

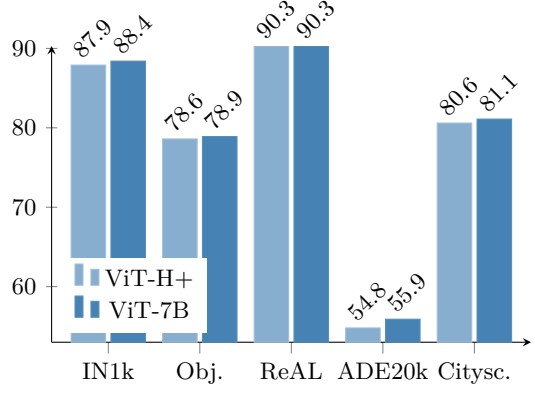

(a) DINOv3 family of models.      (b) ViT-H+ v.s. ViT-7B.

Figure 16: (a) Presentation of the distilled models' characteristics. CNX stands for ConvNeXT. We present per model the number of parameters and the GFLOPs estimated on images of size $256 \times 256$ and $512 \times 512$. (b) We compare DINOv3 ViT-H+ to its 7B-sized teacher; despite having almost $10\times$ less parameters, the ViT-H+ is close to DINOv3 7B in performance.

Figure 2 provides an overview comparison of the DINOv3 family versus other model collections. The DINOv3 family significantly outperforms all others on dense prediction tasks. This includes specialized models distilled from supervised backbones like AM-RADIO and PEspatial. At the same time, our models achieve similar results on classification tasks, making them the optimal choice across compute budgets.

In Sec. 7.1 detail our ViT models and compare them to other open-source alternatives. Then, in Sec. 7.2, we discuss the ConvNeXt models. Finally, following Sec. 5.3, we trained a text encoder aligned with the output of our ViT-L model. We present multi-modal alignment results for this model in Sec. 7.3.

## 7.1 A Vision Transformer for Every Use Case

Our ViT family spans architectures from the compact ViT-S to the larger 840 million parameter ViT-H+ models. The former is designed to run efficiently on resource-constrained devices such as laptops, the latter delivers state-of-the-art performance for more demanding applications. We compare our ViT models to the best open-source image encoders of corresponding size, namely DINOv2 (Oquab et al., 2024), SigLIP 2 (Tschannen et al., 2025) and Perception Encoder (Bolya et al., 2025). For a fair comparison, we ensure that the input sequence length is equivalent across models. Specifically, for model with a patch size of 16 we input images of size $512 \times 512$ versus $448 \times 448$ when models are using patch size 14.

Our empirical study clearly demonstrates that DINOv3 models consistently outperform their counterparts on dense prediction tasks. Most notably, on the ADE20k benchmark, the DINOv3 ViT-L model achieves an improvement of over 6 mIoU points compared to the best competitor DINOv2. The ViT-B variant shows a gain of approximately 3 mIoU points against the next best competitor. These substantial improvements highlight the effectiveness of DINOv3's local features in capturing fine-grained spatial details. Furthermore, evaluations on depth estimation tasks also reveal consistent performance gains over competing approaches. This underscores the versatility of the DINOv3 family across different dense vision problems. Importantly, our models achieve competitive results on global recognition benchmarks such as ObjectNet and ImageNet-1k. This indicates that the enhanced dense task performance does not come at the expense of global task accuracy. This balance confirms that DINOv3 models provide a robust and well-rounded solution, excelling across both dense and global vision tasks without compromise.

On another note, we want to also validate if the largest models that we distill capture all the information from the teacher. To this end, we run a comparison of our largest ViT-H+ with the 7B teacher. As shown

Table 16: Comparison of our family of models against open-source alternatives of comparable size. We showcase our ViT-{S, S+, B, L, H+} models on a representative set of global and dense benchmarks: classification (IN-ReAL, IN-R, ObjectNet), retrieval (Oxford-H), segmentation (ADE20k), depth (NYU), tracking (DAVIS at 960px), and keypoint matching (NAVI, SPair). We match the number of patch tokens for a fair comparison of models of different patch size. See Tabs. 36 and 37 for more results of our models.

| Size | Model | Global Tasks | | | | Dense Tasks | | | | |
|---|---|---|---|---|---|---|---|---|---|---|
| | | IN-ReAL | IN-R | Obj. | Ox.-H | ADE20k | NYU↓ | DAVIS | NAVI | SPair |
| S | DINOv2 | 87.3 | 54.0 | 47.8 | 39.5 | 45.5 | 0.446 | 73.6 | 53.4 | 51.6 |
| S | DINOv3 | 87.0 | 60.4 | 50.9 | 49.5 | 47.0 | 0.403 | 72.7 | 56.3 | 50.4 |
| S+ | DINOv3 | 88.0 | 68.8 | 54.6 | 50.0 | 48.8 | 0.399 | 75.5 | 57.1 | 55.2 |
| B | PEcore | 87.5 | 68.4 | 57.9 | 20.9 | 37.4 | 0.641 | 44.5 | 41.8 | 13.7 |
| B | SigLIP 2 | 89.3 | 80.6 | 66.9 | 15.3 | 41.6 | 0.512 | 63.2 | 45.4 | 32.8 |
| B | DINOv2 | 89.0 | 68.4 | 57.3 | 51.0 | 48.4 | 0.416 | 72.9 | 56.9 | 57.1 |
| B | DINOv3 | 89.3 | 76.7 | 64.1 | 58.5 | 51.8 | 0.373 | 77.2 | 58.8 | 57.2 |
| L | PEcore | 90.1 | 87.7 | 74.9 | 24.0 | 39.7 | 0.650 | 48.2 | 42.1 | 19.2 |
| L | SigLIP 2 | 90.1 | 89.2 | 75.0 | 16.5 | 43.6 | 0.484 | 66.3 | 47.8 | 41.9 |
| L | DINOv2 | 89.7 | 79.1 | 64.7 | 55.7 | 48.8 | 0.394 | 73.4 | 59.9 | 57.0 |
| L | DINOv3 | 90.2 | 88.1 | 74.8 | 63.1 | 54.9 | 0.352 | 79.9 | 62.3 | 61.2 |
| SO400m | SigLIP 2 | 90.3 | 90.4 | 76.2 | 20.6 | 44.0 | 0.402 | 64.8 | 48.8 | 38.7 |
| H+ | DINOv3 | 90.3 | 90.0 | 78.6 | 64.5 | 54.8 | 0.352 | 79.3 | 63.3 | 56.3 |

in Fig. 16b, the largest student achieves performance that is on par with the 8 times larger ViT-7B model. This result not only validates the effectiveness of our distillation process but also demonstrates that, when guided by a high-quality teacher, smaller models can learn to deliver comparable levels of performance. This finding reinforces our belief that *training very large models benefits the broader community*. The strength of larger models can be successfully distilled into more efficient, smaller models with little or no loss of quality.

## 7.2 Efficient ConvNeXts for Resource-Constrained Environments

In this section, we evaluate the quality of our ConvNeXt (CNX) models distilled from the 7B teacher. ConvNeXt models are highly efficient in terms of FLOPs and are well-suited for deployment on devices optimized for convolutional computations. Furthermore, transformer models often do not lend themselves well to quantization (Bondarenko et al., 2021), whereas quantization of convolutional nets is a well explored subject. We distill CNX architectures of size T, S, B, and L (see Fig. 16a) and compare them to the original ConvNeXt models (Liu et al., 2022). These baselines achieve high performance on ImageNet-1k as they were trained in a supervised fashion using ImageNet-22k labels, and thus represent a strong competitor. For this experiment, we provide results for global tasks at input resolutions 256 and 512, for ADE20k at resolution 512, and for NYU at resolution 640.

**Results (Tab. 17)** We find that on in-distribution image classification, our models slightly lag behind the supervised ones at resolution 256 (*e.g.* −0.7 IN-ReAL for CNX-T). However, the trend is reversed at resolution 512, with the supervised ConvNeXts significantly degrading, whereas our models scale with increased input resolution. For out-of-distribution classification (IN-R, ObjectNet), there are significant gaps between the two model families for all sizes—a testament to the robustness of the DINOv3 CNX models. Furthermore, the DINOv3 models offer very large improvement on dense tasks. Indeed, for CNX-T, our model yields a +17.9 mIoU (42.7 versus 24.8) improvement, and for CNX-L, our model gets +14.5 mIoU (47.8 versus 33.3). The combination of high performance and computational efficiency makes the distilled ConvNeXt models especially promising for real-world applications where resource constraints are critical. Aside from that, the distillation of the ViT-7B model into smaller ConvNeXt models is particularly exciting, as it bridges two fundamentally different architectures. While ViT-7B is based on transformer blocks with

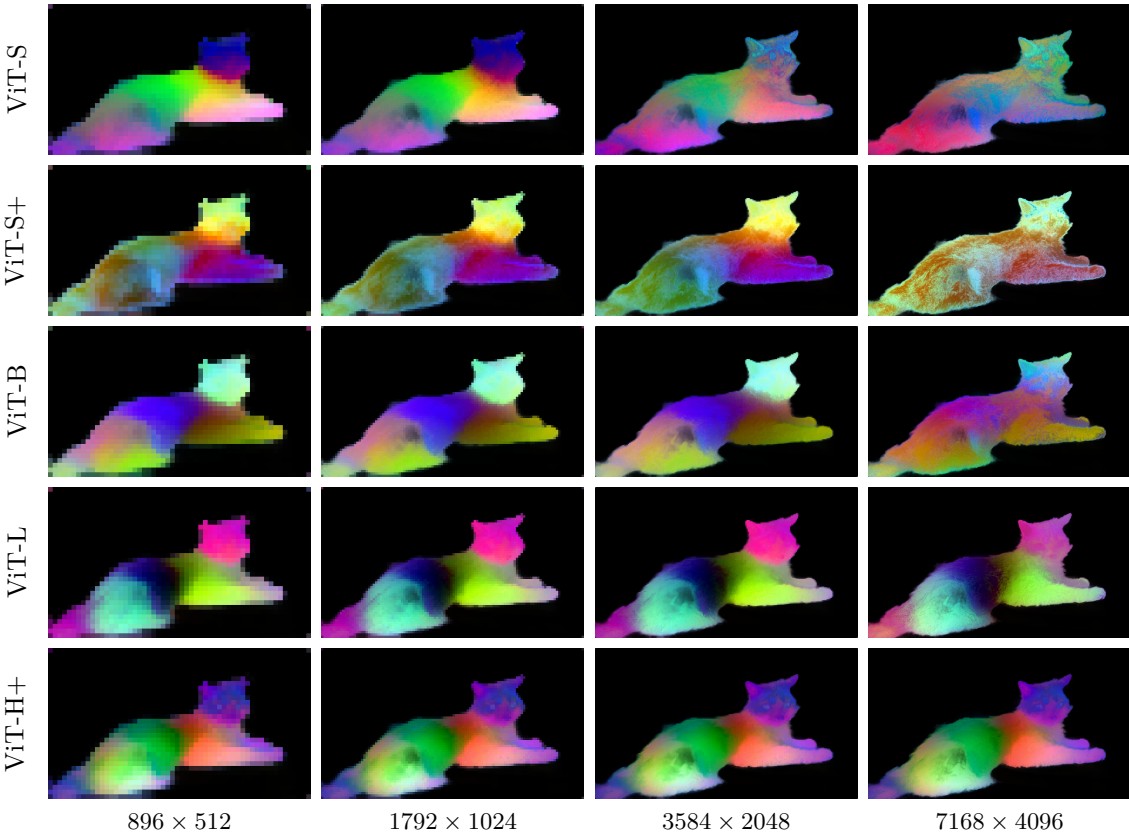

Figure 17: Stability of the features at multiple resolutions for the DINOv3 ViT family of models. Top-to-bottom: ViT-S, S+, B, L, H+. We run inference on an image at multiple resolutions, then perform principal component analysis on the features computed on a $1792 \times 1024$ image ($112 \times 64$ image tokens). We then project features at all resolutions onto the principal components 5–7 that we map to the RGB space for visualization. While the models are functional at all resolutions, we observe that the features remain consistent across a large range of resolutions before drifting: for example, ViT-S+ features are stable between $896 \times 512$ and $3584 \times 2048$ inputs, while ViT-L barely starts drifting on the largest resolution $7168 \times 4096$. ViT-H+ remains stable throughout the whole tested range.

a CLS token, ConvNeXt relies on convolutional operations without a CLS token, making this transfer of knowledge non-trivial. This achievement highlights the versatility and effectiveness of our distillation process.

### 7.3 Zero-shot Inference with DINOv3-based `dino.txt`

As detailed in Sec. 5.3, we train a text encoder to align both the CLS token and the output patches of the distilled DINOv3 ViT-L model to text, following the recipe of dino.txt by Jose et al. (2025). We evaluate the quality of the alignment both at the global- and patch-level on standard benchmarks. We report the zero-shot classification accuracy using CLIP protocol (Radford et al., 2021) on the ImageNet-1k, ImageNet-Adversarial, ImageNet-Rendition and ObjectNet benchmarks. For image-text retrieval, we evaluate on the COCO2017 dataset (Tsung-Yi et al., 2017) and report Recall@1 on both image-to-text (I → T) and text-to-image (T → I) tasks. To probe the quality of patch-level alignment, we evaluate our model on the open-vocabulary segmentation task using the common benchmarks ADE20k, Cityscapes ('City.'), COCO-Stuff ('Suff') and Pascal Context ('C59') for which we report the mIoU metric. We provide more details for the evaluation protocol and the implementation details in App. E.14.

**Results (Tab. 18)** We compare our text-aligned DINOv3 ViT-L with competitors in the same size class. For a fair comparison, we ensure that all models operate on the same sequence length of 576 tokens, and eval-

Table 17: Evaluation of our distilled DINOv3 ConvNeXt models. We compare our models to off-the-shelf ConvNeXts trained supervised on ImageNet-22k (Liu et al., 2022). For global tasks, we give results at input resolutions 256 and 512, as we found the supervised models to significantly degrade at resolution 512. See also Tabs. 36 and 37 for more results of our models.

| Size | Model | Global Tasks | | | | | | Dense Tasks | |
|------|-------|------|------|------|------|------|------|--------|------|
| | | IN-ReAL | | IN-R | | Obj. | | ADE20k | NYU↓ |
| | | 256 | 512 | 256 | 512 | 256 | 512 | | |
| T | Sup. | 87.3 | 83.0 | 45.0 | 33.0 | 44.5 | 27.1 | 24.8 | 0.666 |
| T | DINOv3 | 86.6 | 87.7 | 73.7 | 74.1 | 52.6 | 58.7 | 42.7 | 0.448 |
| S | Sup. | 88.9 | 86.8 | 52.8 | 39.1 | 50.8 | 40.0 | 22.6 | 0.630 |
| S | DINOv3 | 87.9 | 88.7 | 73.7 | 74.1 | 52.6 | 58.7 | 44.8 | 0.432 |
| B | Sup. | 89.3 | 87.8 | 57.3 | 46.2 | 53.6 | 46.5 | 26.5 | 0.596 |
| B | DINOv3 | 88.5 | 89.2 | 77.2 | 78.2 | 56.2 | 61.3 | 46.3 | 0.420 |
| L | Sup. | 89.6 | 88.1 | 58.4 | 46.6 | 55.0 | 47.7 | 33.3 | 0.567 |
| L | DINOv3 | 88.9 | 89.4 | 81.3 | 82.4 | 59.3 | 65.2 | 47.8 | 0.403 |

Table 18: Comparing our text-aligned DINOv3 ViT-L to the state-of-the-art. Our model achieves excellent dense alignment performance while staying competitive in global alignment tasks. All compared models are of ViT-L size and operate on the same sequence length of 576. Evaluation protocol details in App. E.14.

| Method | Res./Patch | Classification | | | | Retrieval | | Segmentation | | | |
|--------|-----------|------|------|------|------|-----------------|-----------------|--------|-------|-------|------|
| | | IN1k | A | R | Obj. | I $\rightarrow$ T | T $\rightarrow$ I | ADE20k | City. | Stuff | C59 |
| EVA-02-CLIP | 336/14 | 80.4 | 82.7 | 93.0 | 78.5 | 64.8 | 46.2 | 10.2 | 13.6 | 11.0 | 16.3 |
| dino.txt | 336/14 | 81.6 | 83.2 | 88.8 | 74.5 | 62.5 | 45.0 | 20.1 | 30.5 | 21.1 | 30.7 |
| SigLIP 2 | 384/16 | 83.2 | 84.0 | **95.7** | 78.2 | **73.9** | **55.6** | 10.8 | 16.7 | 10.1 | 14.6 |
| PE-Core | 336/14 | **83.5** | **88.7** | 95.0 | **82.2** | 73.4 | 55.3 | 16.6 | 19.8 | 18.9 | 27.8 |
| DINOv3 dino.txt | 384/16 | 82.3 | 85.4 | 93.0 | 80.5 | 63.7 | 45.6 | **24.6** | **36.9** | **24.7** | **36.9** |

uate them in their native input resolution using the same code. Compared to Jose et al. (2025), which aligns DINOv2 to text, DINOv3 leads to significantly better performance on all benchmarks. On global alignment tasks, we compare favorably strong baselines such as EVA-02-CLIP (Sun et al., 2023) but slightly behind SigLIP2 (Tschannen et al., 2025) and Perception Encoder (Bolya et al., 2025). On dense alignment tasks, our text-aligned model shows excellent performance on all the challenging benchmarks evaluated thanks to the clean feature maps of DINOv3. Results obtained by comparing the patch features of DINOv3 dino.txt significantly outperform all baselines—which require doing more complicated patch feature extraction.

# 8 DINOv3 on Geospatial Data

Our self-supervised learning recipe is generic and can be applied to any image domain. In this section, we showcase this universality by building a DINOv3 7B model for satellite images, which have very different characteristics (*e.g.* object texture, sensor noise, and focal views) than the web images on which DINOv3 was initially developed.

## 8.1 Pre-Training Data and Benchmarks

Our satellite DINOv3 7B model is pre-trained on SAT-493M, a dataset of 493 millions of $512 \times 512$ images sampled randomly from Maxar RGB ortho-rectified imagery at 0.6 meter resolution. We use the exact same set of hyper-parameters that are used for the web DINOv3 7B model, except for the RGB mean and std

Table 19: Evaluation of different backbones for high-resolution canopy height prediction. All models are trained with a DPT decoder. Results are presented either for experiments with the decoder trained on SatLidar and evaluated on IID samples (SatLidar Val) and OOD test sets (SatLidar Test, Neon and São Paulo), or for experiments with the decoder trained and evaluated on the Open-Canopy dataset. We list mean absolute error (MAE) and the block $R^2$ metric from Tolan et al. (2024). For completeness, we additionally evaluate the original decoder of Tolan et al. (2024) that was trained on Neon dataset (denoted by *). [†] We did not use the vegetation mask for the Open-Canopy evaluation, and performed inference on full images.

| Method | Arch. | SatLidar | | | | | | | | Open Canopy[†] |
| | | SatLidar Val | | SatLidar Test | | Neon Test | | São Paulo | | |
| | | MAE↓ | $R^2$↑ | MAE↓ | $R^2$↑ | MAE↓ | $R^2$↑ | MAE↓ | $R^2$↑ | MAE↓ |
|---|---|---|---|---|---|---|---|---|---|---|
| Tolan et al. (2024)* | ViT-L | 2.8 | 0.86 | 4.0 | 0.61 | 2.7 | 0.73 | **5.4** | 0.42 | — |
| Tolan et al. (2024) | ViT-L | 2.4 | 0.90 | 3.4 | 0.81 | 2.9 | 0.69 | **5.4** | 0.48 | 2.42 |
| DINOv3 Web | ViT-7B | 2.4 | 0.90 | 3.6 | 0.74 | 2.7 | 0.75 | 5.9 | 0.34 | 2.17 |
| DINOv3 Sat | ViT-L | **2.2** | 0.91 | **3.2** | 0.81 | **2.4** | **0.81** | 5.8 | 0.42 | 2.07 |
| DINOv3 Sat | ViT-7B | **2.2** | **0.92** | **3.2** | **0.82** | 2.6 | 0.74 | 5.5 | **0.51** | **2.02** |

normalization that are adapted for satellite images, and the training length. Similar to the web model, our training pipeline for the satellite model consists of 100k iterations of initial pre-training with global crops ($256 \times 256$), followed by 10k iterations using Gram regularization, and finalized with 8k steps of high resolution fine-tuning at resolution 512. Also similar to the web model, we distill our 7B satellite model into a more manageable ViT-Large model to facilitate its use in low-budget regime.

We evaluate DINOv3 satellite and web models on multiple earth observation tasks. For the task of global canopy height mapping, we use the Satlidar dataset described in App. E.15, which consists of one million $512 \times 512$ images with LiDAR ground truths split into train/val/test splits with ratios 8/1/1. The splits include the Neon and São Paulo dataset used by Tolan et al. (2024). For national-scale canopy height mapping, we evaluate on Open-Canopy (Fogel et al., 2025), which combines SPOT 6-7 satellite imagery and aerial LiDAR data over 87,000 km$^2$ across France. Since images in this dataset have 4 channels including the additional infra-red (IR) channel, we adapt our backbone by taking the average of the three channels in the weights of the patch embed module and adding it to the weights as the fourth channel. We trained a DPT decoder on $512 \times 512$ crops of images resized to 1667 to match the Maxar ground sample resolution.

Semantic geospatial tasks are assessed with GEO-Bench (Lacoste et al., 2023), which comprises six classification and six segmentation tasks spanning various spatial resolutions and optical bands. The GEO-Bench tasks are diverse, including the detection of rooftop-mounted photovoltaic systems, classifying local climate zones, measuring drivers of deforestation, and detecting tree crowns. For high-resolution semantic tasks, we consider the land cover segmentation dataset LoveDA (Wang et al., 2022a), the object segmentation dataset iSAID (Zamir et al., 2019), and the horizontal detection dataset DIOR (Li et al., 2020).

## 8.2 Canopy Height Estimation

Estimating canopy height from satellite imagery is a challenging metric task, requiring accurate recovery of continuous spatial structure despite random variations in slope, viewing geometry, sun angle, atmospheric scattering, and quantization artifacts. This task is critical for global carbon monitoring and for forest and agriculture management (Harris et al., 2021). Following Tolan et al. (2024), the first work to leverage a SSL backbone trained on satellite images for this task, we train a DPT head on top of frozen DINOv3 on the SatLidar1M training set, then evaluate it on i.i.d. samples on SatLidar1M validation set as well as out-of-distribution test sets including SatLidar1M test, Neon and Sao Paulo. We additionally train and evaluate on the Open-Canopy dataset.

**Results (Tab. 19)** We compare different SSL backbones, denoting with "DINOv3 Sat" the model trained the SAT-493M dataset, and with "DINOv3 Web" the model trained on LVD-1689M (see Sec. 3.1). It

Table 20: Comparison of our DINOv3 models against strong baselines DOFA (Xiong et al., 2024), Prithvi-v2 (Szwarcman et al., 2024), and Tolan et al. (2024) in Geo-Bench tasks. While Privthi-v2 and DOFA leverage all available optical bands, our models achieve significantly better performance with only RGB inputs.

(a) Classification tasks.

| Method | Arch. | FT | Bands | m-BEnet | m-brick-kiln | m-eurosat | m-forestnet | m-pv4ger | m-so2sat | Mean |
|---|---|---|---|---|---|---|---|---|---|---|
| DOFA | ViT-L | ❄ | all | 68.7 | 98.4 | 96.6 | 55.7 | 98.2 | 61.6 | 79.9 |
| Best of Prithvi-v2 | ViT-L/H | ❄ | all | 71.2 | **98.8** | 96.4 | 54.1 | 98.1 | 59.1 | 79.6 |
| Tolan et al. (2024) | ViT-L | ❄ | RGB | 66.0 | 97.1 | 95.2 | 56.3 | 94.3 | 58.1 | 77.8 |
| DINOv3 Sat | ViT-L | ❄ | RGB | 73.0 | 96.5 | 94.1 | 60.6 | 96.0 | 57.4 | 79.6 |
| DINOv3 Sat | 7B | ❄ | RGB | 74.0 | 97.2 | 94.8 | **62.3** | 96.1 | 62.1 | 81.1 |
| DINOv3 Web | 7B | ❄ | RGB | **74.6** | 97.7 | **97.0** | 57.9 | **98.3** | **63.8** | **81.6** |

(b) Segmentation tasks.

| Method | Arch. | FT | Bands | m-cashew* | m-chesapeake | m-NeonTree | m-nz-cattle | m-pv4ger-seg | m-SA-crop | Mean |
|---|---|---|---|---|---|---|---|---|---|---|
| DOFA | ViT-L | ❄ | all | 81.2 | 61.6 | 58.5 | 77.4 | 95.1 | 35.7 | 68.3 |
| Best of Prithvi-v2 | ViT-L/H | ❄ | all | 90.2 | 69.4 | 59.1 | 81.0 | 95.3 | **41.9** | 72.8 |
| Tolan et al. (2024) | ViT-L | ❄ | RGB | 92.8 | 73.7 | 58.1 | 83.1 | 94.7 | 35.1 | 72.9 |
| DINOv3 Sat | ViT-L | ❄ | RGB | 94.2 | 75.6 | 61.8 | **83.7** | 95.2 | 36.8 | 74.5 |
| DINOv3 Sat | 7B | ❄ | RGB | 94.1 | **76.6** | 62.6 | 83.4 | 95.5 | 37.6 | 75.0 |
| DINOv3 Web | 7B | ❄ | RGB | **96.0** | 76.5 | **66.4** | **83.7** | **95.9** | 36.8 | **75.9** |

*Conversion to 6 classes following Szwarcman et al. (2024).

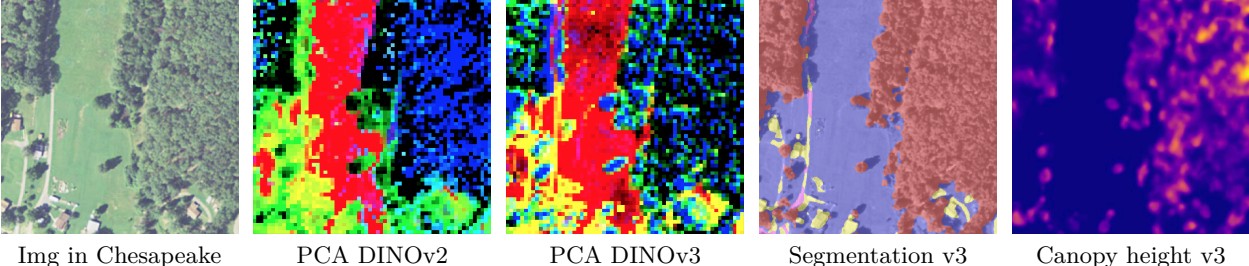

| Img in Chesapeake | PCA DINOv2 | PCA DINOv3 | Segmentation v3 | Canopy height v3 |

Figure 18: Illustration of versatile applications in remote sensing made possible by a single DINOv3 model. The PCA on DINOv3 features shows finer details than DINOv2. The segmentation map was computed using only GEO-Bench chesapeake labels. The canopy height model decoder was trained on the Open-Canopy dataset using 4 channels (RGB + InfraRed), while inference was performed on RGB channels only.

can be seen that DINOv3 satellite models yield state-of-the-art performance on most benchmarks. Our 7B satellite model sets the new state of the art on SatLidar1M val, SatLidar1M test and Open-Canopy, reducing MAE from 2.4 to 2.2, from 3.4 to 3.2 and from 2.42 to 2.02 respectively. These results show that DINOv3 training recipe is generic and can be effectively applied out-of-the-box to other domains. Interestingly, our distilled ViT-L satellite model performs comparably to its 7B counterpart, achieving comparable results on SatLidar1M and Open-Canopy while faring surprisingly better on Neon test set, reaching the lowest MAE of 2.4 compared to 2.6 of the 7B model and 2.9 of Tolan et al. (2024). Our DINOv3 7B web model reaches decent performance on the benchmarks, outperforming Tolan et al. (2024) on SatLidar1M val, Neon and Open-Canopy but stays behind the satellite model. This highlights the strength of domain-specific pretraining for physically grounded tasks like canopy height estimation, where sensor-specific priors and radiometric consistency are important.

## 8.3 Comparison to the Earth Observation State of the Art

We compare the performance of different methods for Earth observation tasks in Tab. 20 and Tab. 21. The frozen DINOv3 satellite and web models set new state-of-the-art results on 12 out of 15 classification,

Table 21: We compare the performance of DINOv3 to state-of-the-art models Privthi-v2 (Szwarcman et al., 2024), BillionFM (Cha et al., 2024) and SkySense V2 (Zhang et al., 2025) for high resolution semantic geospatial tasks. We report mIoU for the segmentation datasets LoveDA (1024×) and iSAID (896×), and mAP for the detection dataset DIOR (800×).

| Method | Arch. | FT | LoveDA | iSAID | DIOR |
|---|---|---|---|---|---|
| Prev. SotA | | 🔥 | BillionFM, ViT-G 54.4 | SkySense V2, Swin-G* **71.9** | SkySense V2, Swin-G* 79.5 |
| Decoder Arch. | | | UPerNet | UPerNet | Faster-RCNN |
| Privthi-v2 | ViT-H | 🔥 | 52.2 | 62.8 | — |
| DINOv3 Sat | ViT-L | ❄ | 54.4 | 62.9 | 72.7 |
| DINOv3 Sat | ViT-7B | ❄ | 55.3 | 64.8 | 76.6 |
| DINOv3 Web | ViT-7B | ❄ | **56.2** | 71.4 | **80.5** |

*Uses modified DINOv2 SSL with supervised pretraining alignment on OpenStreetMap, reporting +0.8 mIoU on iSAID.*

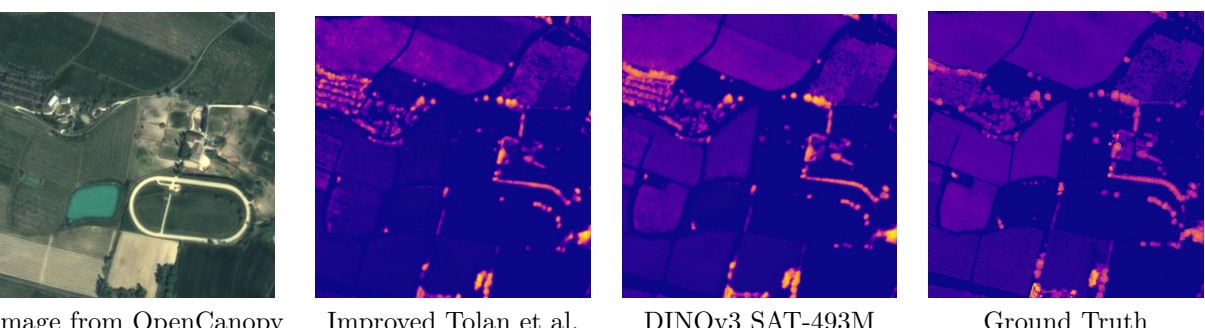

| Image from OpenCanopy | Improved Tolan et al. | DINOv3 SAT-493M | Ground Truth |
|---|---|---|---|

Figure 19: A qualitative comparison of the DINOv3 7B satellite model to Tolan et al. (2024) on the Open Canopy dataset. For both models, the decoder is trained on 448×448 input images. It can be seen that DINOv3 produces more accurate maps, for example the accurate height for the trees on the field.

segmentation, and horizontal object detection tasks. Our Geo-Bench results surpass prior models, including Prithvi-v2 (Szwarcman et al., 2024) and DOFA (Xiong et al., 2024), which use 6+ bands for Sentinel-2 and Landsat tasks, as well as task-specific fine-tuning (Tab. 20). Despite using a frozen backbone with RGB-only input, the DINOv3 satellite model outperforms previous methods on the three unsaturated classification tasks and on five of six segmentation tasks. Interestingly, the DINOv3 7B web model is very competitve on these benchmarks. It achieves comparable or stronger performance on many GEO-Bench tasks as well as on large-scale, high-resolution remote sensing benchmarks for segmentation and detection. As shown in Tab. 20 and Tab. 21, the frozen DINOv3 web model establishes new leading results Geo-Bench tasks as well as for segmentation and detection tasks on the LoveDA and DIOR datasets.

These findings have broader implications for the design of geospatial foundation models. Those have recently emphasized heuristic techniques such as multitemporal aggregation, multisensor fusion, or incorporating satellite-specific metadata (Brown et al., 2025; Feng et al., 2025). Our results show that general-purpose SSL can match or exceed satellite-specific approaches for tasks that depend on precise object boundaries (segmentation or object detection). This supports emerging evidence finding that domain-agnostic pretraining can offer strong generalization even in specialized downstream domains (Lahrichi et al., 2025).

Collectively, our results suggest task-dependent benefits of domain-specific pretraining. The DINOv3 satellite model excels in metric tasks like depth estimation, leveraging satellite-specific priors. In contrast, the DINOv3 web model achieves state-of-the-art results on semantic geospatial tasks through diverse, universal representations. The complementary strengths of both models illustrate the broad applicability and effectiveness of the DINOv3 SSL paradigm.

Table 22: Carbon footprint of model training. We report the potential carbon emission of reproducing a full model pre-training, computed using a PUE of 1.1 and carbon intensity factor of 0.385kg $CO_2$eq/KWh.

| Model | Arch. | GPU type | Power (W) | Steps | GPU hours | PUE | Total power (MWh) | Emission (tCO$_2$eq) |
|---|---|---|---|---|---|---|---|---|
| MetaCLIP | ViT-G | A100-40GB | 400W | 390k | 368,640 | 1.1 | 160 | 62 |
| DINOv2 | ViT-g | A100-40GB | 400W | 625k | 22,016 | 1.1 | 9.7 | 3.7 |
| DINOv3 | ViT-7B | H100-SXM5 | 700W | 1,000k | 61,440 | 1.1 | 47 | 18 |

## 9  Environmental Impact

To estimate the carbon emission of our pre-training, we follow the methodology used in previous work in natural language processing (Strubell et al., 2019; Touvron et al., 2023) and SSL (Oquab et al., 2024). We fix the value of all exogenous variables, *i.e.* the Power Usage Effectiveness (PUE) and carbon intensity factor of a power grid to the same value as used by Touvron et al. (2023), *i.e.* we assume a PUE of 1.1 and a carbon intensity factor of the US average of 0.385 kg $CO_2$eq/KWh. For the power consumption of GPUs, we take their thermal design power: 400W for A100 GPUs and 700W for H100 GPUs. We report the details of the computation for the pre-training of our ViT-7B in Tab. 22. For reference, we provide the analogous data for DINOv2 and MetaCLIP. As another point of comparison, the energy required to train one DINOv3 model (47 MWh) is roughly equivalent to that required for 240,000 km of driving with an average electric vehicle.

**Carbon Footprint of the Whole Project**   In order to compute the carbon footprint of the whole project, we use a rough estimate of a total 9M GPU hours. Using the same grid parameters as presented above, we estimate the total footprint to be roughly 2600 tCO$_2$eq. For comparison, a full Boeing 777 return flight between Paris and New York corresponds to approximately 560 tCO$_2$eq. Supposing 12 such flights per day, the environmental impact of our project represents half of all flights between these two cities for one day. This estimate only considers the electricity for powering the GPUs and ignores other emissions, such as cooling, manufacturing, and disposal.

## 10  Conclusion

DINOv3 represents a significant advancement in the field of self-supervised learning, demonstrating the potential to revolutionize the way visual representations are learned across various domains. By scaling dataset and model size through meticulous data preparation, design, and optimization, DINOv3 showcases the power of self-supervised learning to eliminate the dependency on manual annotations. The introduction of the Gram anchoring method effectively mitigates the degradation of dense feature maps over extended training periods, ensuring robust and reliable performance.

Together with the implementation of post-hoc polishing strategies, such as high-resolution post-training and distillation, we achieve state-of-the-art performance across a wide range of visual tasks with no fine-tuning of the image encoder. The DINOv3 suite of vision models not only sets new benchmarks but also offers a versatile solution across various resource constraints, deployment scenarios, and application use cases. The progress made with DINOv3 is a testament to the promise of self-supervised learning in advancing the state of the art in computer vision and beyond.

## 11  Broader Impact

**Data Privacy and Ethics.**   The pre-training of DINOv3 utilizes public Instagram data, restricted to posts from users aged 18 and older. This data collection and usage are conducted in strict accordance with Meta's Terms of Service and Privacy Policy. We refer reader to Meta's Privacy Policy[2] and the AI section[3] of the

---

[2]https://www.facebook.com/privacy/policy
[3]https://www.facebook.com/privacy/genai

Privacy Center for details on user consent, transparency, and the controls available to adjust preferences around data collection and use.

**Fairness and Bias Mitigation.** As a vision foundation model, DINOv3 learns representations from large-scale, real-world datasets and may inadvertently inherit or amplify historical biases present in that data. To proactively address these concerns, we conduct a fairness analysis in App. C.8. Our evaluation specifically focuses on geographical fairness and representation across diverse income brackets and global regions, ensuring the model's features remain robust and equitable across different cultural and socio-economic contexts.

**Societal and Scientific Benefits.** DINOv3 is designed as a versatile backbone to accelerate progress across a wide spectrum of visual tasks. Building on the legacy of previous foundation models, we anticipate significant downstream impact in critical scientific domains. These include, but are not limited to, remote sensing for climate monitoring (Tolan et al., 2024), histopathology (Chen et al., 2024; Vorontsov et al., 2024), and fluorescent microscopy (Moutakanni et al., 2025). By providing a high-capacity, general-purpose architecture, we aim to lower the barrier for researchers to develop specialized tools in medicine, environmental science, and beyond.

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
