## A    Appendix

## B    Artifacts and Outliers in Large-Scale Training

This section provides a discussion about the emergence of artifacts and outliers that has recently been observed in the training of large models in both the LLM (An et al., 2025) and the visual domains (Darcet et al., 2024). Borrowing the definition from An et al. (2025), outliers are typically characterized as network's activations whose values deviate significantly from the average of their distribution. During the training of DINOv3, we identified such outliers at different levels: some occurring at the patch level and others at the feature dimension level. We discuss bellow the different types of outlier observed, their impact on the training and results. We also discuss our different attempts at fixing them and our first conclusions.

### B.1    High-Norm Patch Outliers

Darcet et al. (2024) discovered that patch outliers negatively affect performance in DINOv2. These outliers are primarily characterized as high-norm tokens, often located in low-information background regions of an image. These tokens are observed to play a key role in the internal communication between patches and the CLS token. Additionally, this phenomenon affects other models as well, whether trained with supervision or not, such as CLIP (Radford et al., 2021). When scaling to a 7B model, we observe the emergence of such high-norm patches, predominantly in the background area. In this section, we present results from 7B models trained for 150k iterations, which, although limited, provide us with initial signals to guide our decisions. We plot the output patch norms (before the layer norm) in Fig. 20a, in the column '∅', with high-norm patches in yellow appearing in the sky and other low-information areas.

**Token Registers**    In order to mitigate the appearance of such token outliers, (Darcet et al., 2024) proposes a simple yet effective solution: introducing additional tokens, called registers, into the input sequence of the ViT. Their role is to take over the internal communication between patches and the CLS. Following the conclusions, we use 4 registers and do not ablate further due to the high experimental cost. Figure 20a illustrates examples of this strategy in action, where we observe the elimination of high-norm outliers, as further confirmed by the corresponding histogram of the norm distribution. Moreover, we quantitatively observe in Fig. 20b the benefit of incorporating additional register tokens on the ImageNet-1k (IN1k) benchmark.

**Integrating Biases in the Attention Mechanism**    Recent work by An et al. (2025) investigates the appearance of outliers in the LLM realm across different models and architectures. The authors analyze different types of outliers which are observed to be intrinsically linked to the attention mechanism. They study different approaches to mitigate the problem, from which we select the two shown to reduce the outliers problem, namely the attention bias and the context-aware scaling factor, which we call 'value gating' here. The attention bias consists of integrating two learnable bias terms $\mathbf{k}', \mathbf{v}' \in \mathbb{R}^D$ in the keys and values matrices, respectively. It can be defined as follows:

$$\text{Attn}(Q, K, V; \mathbf{k}', \mathbf{v}') = \text{softmax}\left(\frac{Q[K^T\mathbf{k}']}{\sqrt{d}}\right)\begin{bmatrix} V \\ \mathbf{v}' \end{bmatrix}, \tag{4}$$

with $Q, K, V \in \mathbb{R}^{T \times d}$ the query, key, and value matrices and $d = n_h \times d_h$ the dimensionality of the hidden space, with $n_h$ the number of heads and $d_h$ the dimensionality per head. Alternatively, the value gating strategy incorporates a learned input-dependent scaling factor that is produced using a linear transform $W_{\text{gate}} \in \mathbb{R}^{d_{\text{in}} \times h}$. It is applied to modulate the output of attention as

$$\text{Attn}_{\text{gated}}(Q, K, V, X) = X_{\text{att}} \cdot \sigma(XW_{\text{gate}}), \qquad X_{\text{att}} = \text{Attn}(Q, K, V), \tag{5}$$

where $X \in \mathbb{R}^{T \times d_{\text{in}}}$ is the original attention input, and $XW_{\text{gate}}$ broadcasts over dimensions per head. In Fig. 20a, we observe that the value gating strategy substantially modifies the distribution of patch norms, resulting in generally higher norm values and the elimination of clear outliers. While the attention bias mechanism mitigates the presence of high-norm tokens, it does not completely resolve the issue, as some high-norm patches persist—as visible in the top row image—when compared to our results using register

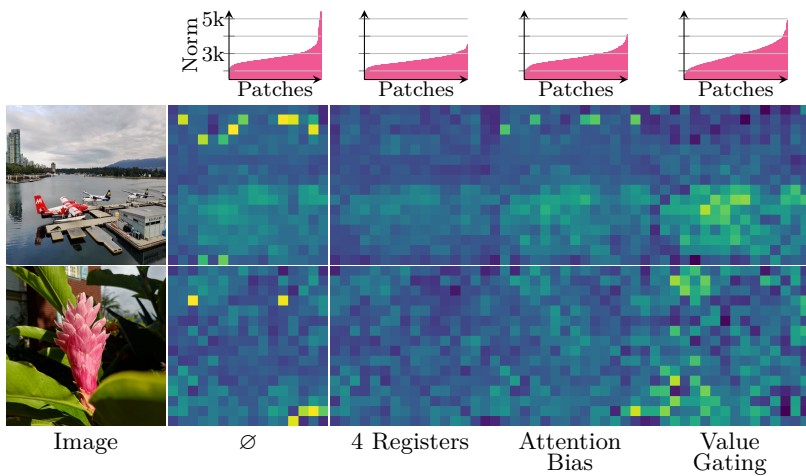

(b) Quantitative ablation.

| Outlier Strategy | IN1k (Linear) | ADE20k mIoU |
|---|---|---|
| ∅ | 86.4 | **53.2** |
| 4 Registers | **86.6** | 53.0 |
| Attention Bias | 86.5 | 52.7 |
| Value Gating | 86.3 | 52.2 |

(a) Visualization of patch norms by outlier strategy. The bottom two rows share a colormap per row, from dark blue (low) to yellow (high).

Figure 20: Impact of the different strategies to mitigate the presence of high-norm patch outliers, evaluated both (a) qualitatively and (b) quantitatively. We produce results with a 7B model trained with our recipe for 150k iterations, without any high-norm handling strategy '∅', when using four register tokens (Darcet et al., 2024), or the attention bias and value gating strategies (An et al., 2025). In (a, first row), we plot the distribution of the output patch norms (sorted by ascending values) computed for three images. We also visualize the output patch norms per image (bottom two rows), with the same colormap–min and max values are computed per image over the different outlier strategies.

tokens. Notably, the best performance is achieved with the incorporation of the register tokens, which is why we adopt this strategy for all experiments reported is the paper.

## B.2  Feature Dimension Outliers

The introduction of additional registers into the model architecture effectively resolves the issue of high-norm patch outliers. However, during the training of 7B models, we observe a distinct type of outlier that emerges not across patches, but within the feature (channel) dimension of the learned representations. Specifically, analysis of patch activations across transformer layers and training iterations reveals that a small subset of feature dimensions attain exceptionally large magnitudes, even as the norms across patches remain stable. Interestingly, these feature dimension outliers exhibit consistently high values across different patches and images, a behavior that contrasts with observations reported in (An et al., 2025). Moreover, these outlier dimensions consistently persist across the layers of a given model, increasing in magnitude with depth and reaching their maximum values in the output layer. They also progressively increase in magnitude throughout the course of training.

We conduct experiments attempting to neutralize these dimensions during both training and inference. Our findings indicate that these dimensions play a significant role during training, as applying L2-regularization to suppress them results in a performance drop. However, removing these dimensions at inference time does not lead to significant performance changes, suggesting that they primarily carry trivial or non-informative signals. Additionally, we observe that the final layer normalization is trained to substantially scale down these outlier dimensions. Thus, we recommend to apply the final layer norm to the features of the final layer for downstream use. Alternatively, applying batch normalization can also suppress these feature dimension outliers, as their elevated values are consistent across patches and images.

A word of caution applies to using features from earlier layers. As discussed above, these earlier layers are also affected by feature dimension outliers which can lead to ill-conditioned features. While the final layer normalization is well-suited to normalize the distribution of the final features, its learned parameters may be

Table 23: Details of year of publication, performance, and reference of the numbers used in Fig. 1. For all papers, we report top-1 accuracy of this algorithm with the largest model on ImageNet. For weakly- and self-supervised models, we provide linear probing performance. For dates, we use the year of first appearance on arXiv.

| Year | Supervised | | Weakly-Supervised | | Self-Supervised | |
|------|------------|-----------|------------|-----------|------------|-----------|
| | Top-1 | Reference | Top-1 | Reference | Top-1 | Reference |
| 2012 | 59.3 | Krizhevsky et al. (2012) | | | | |
| 2013 | | | | | | |
| 2014 | | | | | | |
| 2015 | 78.6 | He et al. (2016) | 34.9 | Joulin et al. (2016) | | |
| 2016 | | | | | | |
| 2017 | 80.9 | Xie et al. (2017) | | | | |
| 2018 | | | 83.6 | Mahajan et al. (2018) | 38.2 | Caron et al. (2018) |
| 2019 | 84.3 | Tan & Le (2019) | | | 68.6 | He et al. (2020) |
| 2020 | 87.5 | Kolesnikov et al. (2020) | | | 75.3 | Caron et al. (2020) |
| 2021 | 88.6 | Dosovitskiy et al. (2020) | 88.4 | Radford et al. (2021) | 82.3 | Zhou et al. (2021) |
| 2022 | | | | | | |
| 2023 | 89.5 | Dehghani et al. (2023) | | | 86.5 | Oquab et al. (2024) |
| 2024 | | | | | | |
| 2025 | | | 89.3 | Bolya et al. (2025) | 88.4 | This work |

Table 24: Data sampling ablation. We compare training our ViT-7B with fully heterogeneous, homogeneous batches, or mixed sampling (see Sec. 3.1). We produce results at 200k iterations.

| | Classification | | | Segmentation |
|------|------------|------------|------------|------------|
| Data sampling | k-NN IN-1k val | linear IN-1k val | linear ObjectNet | linear ADE20k |
| Mixed | 84.6 | 87.2 | 72.8 | 53.1 |
| Heterogeneous | 84.2 | 86.8 | 73.1 | 52.9 |
| Homogeneous | 84.5 | 87.1 | 72.2 | 52.5 |

suboptimal for applying it to the features of earlier layers. Indeed, we observe performance decreases for some tasks from doing so. In these cases, we found standard feature scaling techniques (*e.g.* normalization with batch norm or principal component analysis) to be effective in dealing with the feature dimension outliers. For example, for our semantic segmentation (Sec. 6.3.2) and depth estimation experiments (Sec. 6.3.3) using features from intermediate layers, we apply batch normalization.

# C    Additional Results

*Note: The qualitative visualizations included in the paper are produced using a pool of personal holiday photos and are intended for illustrative purposes only; they should not be interpreted as a systematic or exhaustive evaluation.*

## C.1    Evolution Over Years

In Fig. 1, we provide a rough evolution of state-of-the-art performance over the years. Here, we provide the precise references and performances that we reported in the figure. Please find it in Tab. 23.

Table 25: Data sampling ablation. We compare standard RoPE against RoPE with jittering when training the ViT-7B model. Results are reported at 200k iterations.

| | Classification | | | Segmentation | |
|---|---|---|---|---|---|
| | k-NN | linear | linear | linear | linear |
| Data sampling | IN-1k val | IN-1k val | ObjectNet | ADE20k | VOC |
| RoPE box jittering | **84.6** | **87.2** | **72.8** | **53.1** | **85.4** |
| Standard RoPE | 84.3 | 86.1 | **72.8** | 52.2 | 84.9 |

Table 26: Hyper-parameter ablation. We ablate the learning rate and stochastic depth drop rate. Results are reported at 500k iterations.

| | | Classification | | | Segmentation |
|---|---|---|---|---|---|
| | | k-NN | linear | linear | linear |
| Learning rate | Stochastic depth | IN-1k val | IN-1k val | ObjectNet | ADE20k |
| 0.00032 | 0.35 | 84.7 | 88.1 | 76.8 | 50.2 |
| 0.00032 | 0.4 | 84.9 | 87.8 | 75.8 | 51.5 |
| 0.0004 | 0.4 | 84.9 | 88.0 | 76.0 | 51.2 |
| 0.00048 | 0.4 | 84.9 | 87.9 | 75.7 | 51.7 |

## C.2 Data Sampling Comparison

In Tab. 24, we provide a comparison of different data sampling composition methods: 1) with a mix of heterogeneous and homogeneous batches (see Sec. 3.1), 2) fully heterogeneous batches, *i.e.*, each batch contains samples across all datasets, and 3) fully homogeneous batches, *i.e.*, each batch contains samples from only one of the data parts (ImageNet1k, ImageNet22k, Mapillary, retrieval-based curated part, clustering-based curated part). We provide results after 200k iterations. We make sure that these mixtures have the same data proportions used in our main setup. We observe that the overall best performance is obtained with our mixed setup.

## C.3 RoPE Comparison

In Tab. 25, we provide comparing RoPE with jittering (official) and the standard version without jittering. Adding box jittering improves dense tasks, while global tasks show only smaller differences.

## C.4 Hyper-parameter Discussion

When developing DINOv3, we did not conduct an extensive grid search over hyper-parameters, given the computational cost of training at scale. Instead, we performed a targeted sweep over the learning rate and report results after 500k pre-training iterations Tab. 26. Overall, performance is stable across the tested values, with a learning rate of 0.0004 yielding the best overall trade-off.

We also experimented with increasing the weight decay during training by a factor of 4–5, but observed consistently degraded performance. Another sensitive hyperparameter is stochastic depth: following DINOv2, we use a drop-path rate of 0.4. In an ablation where we reduced this value to 0.35 (reported in Tab. 26), we obtained slightly better linear probing accuracy, but performance deteriorated on dense prediction and non-parametric evaluations (e.g., kNN classification).

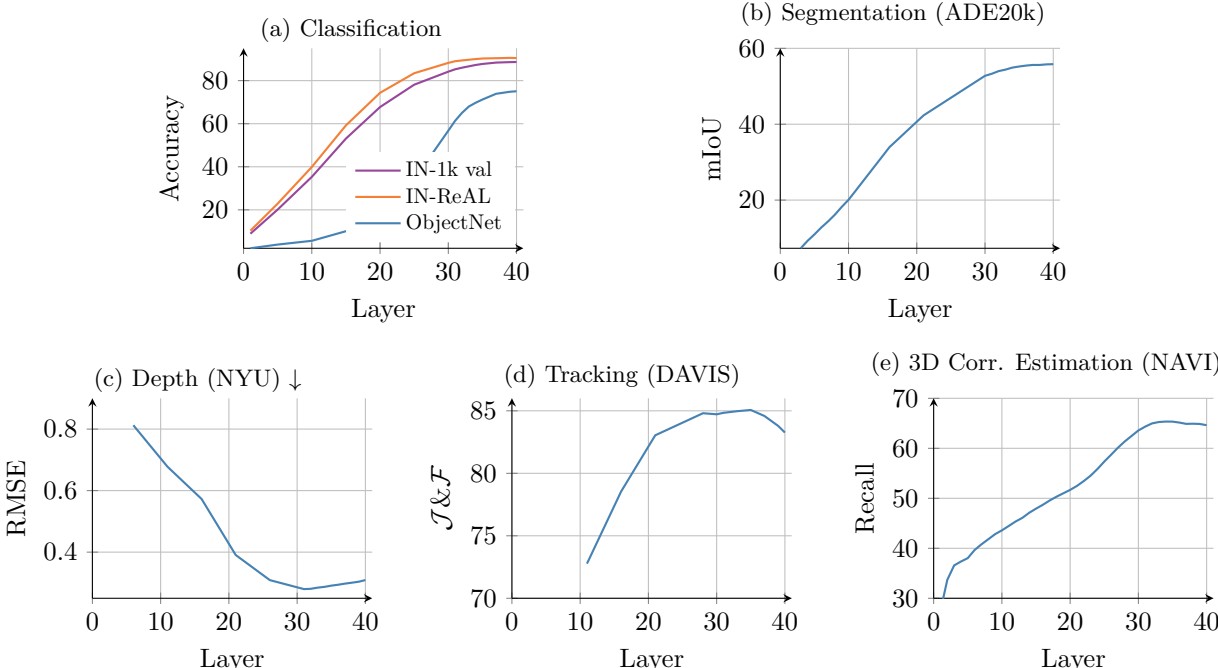

Figure 21: Results on five benchmarks using features from intermediate layers of DINOv3 7B. Evaluations (a-c) use a linear layer (see Secs. 6.1.2 and 6.2.1), while (d, e) use a non-parametric approach (see Secs. 6.1.3 and 6.1.5).

## C.5 Per-Layer Analysis

In this section, we evaluate the quality of our features across the various layers of the DINOv3 7B model. Specifically, we present results from five representative tasks: classification (IN-1k val, ImageNet-ReAL and ObjectNet), segmentation (ADE20k), depth estimation (NYU), tracking (DAVIS), and 3D correspondence estimation (NAVI). For the first 3 benchmarks, a linear layer is trained on the outputs of each backbone layer to assess feature performance as in Secs. 6.1.2 and 6.2.1. For tracking and correspondence estimation, we use non-parametric approaches as in Secs. 6.1.3 and 6.1.5.

The results are shown in Fig. 21. We find that for classification and dense tasks, performance increases smoothly over the layers. Depth estimation, tracking, and 3D correspondence estimation peak around layer 32, indicating that, for tasks where geometry plays a significant role, the downstream performance of DINOv3 can be improved by considering earlier layers. On the other hand, the performance of intermediate layers only slightly improves compared to the last one, making it a good default choice.

## C.6 Additional Results to Main Results Section

We give additional experimental results complementing the main results in Sec. 6. In Tab. 27, we show per-dataset results for finegrained classification on small datasets with linear probing (Fine-S, see Sec. 6.2.1). In Tab. 28, we give full results for the instance recognition evaluation (Sec. 6.2.2), adding more metrics. Finally, in Tab. 29, we give complementary results for our state-of-the-art semantic segmentation model (Sec. 6.3.2) on the COCO-Stuff (Caesar et al., 2018), PASCAL VOC 2012 (Everingham et al., 2012) and Cityscapes (Geiger et al., 2013) datasets.

Table 27: Per-dataset results for finegrained classification on small datasets with linear probing (sFine, see Sec. 6.2.1), following Oquab et al. (2024).

| Method | ViT | Food | C10 | C100 | SUN | Cars | Aircr. | VOC | DTD | Pets | Cal101 | Flowers | CUB | Avg |
|---|---|---|---|---|---|---|---|---|---|---|---|---|---|---|
| *Agglomerative backbones* | | | | | | | | | | | | | | |
| AM-RADIOv2.5 | g/14 | 96.5 | 99.5 | 95.0 | 82.8 | 95.4 | 91.7 | 90.3 | 88.6 | 96.7 | 98.8 | 99.7 | 91.5 | 93.9 |
| *Weakly-supervised backbones* | | | | | | | | | | | | | | |
| SigLIP | g/16 | 97.7 | 99.3 | 92.7 | 85.1 | 96.5 | 88.7 | 91.0 | 87.7 | 98.7 | 90.3 | 99.7 | 90.3 | 93.7 |
| PE-core | G/14 | 97.8 | 99.5 | 95.3 | 85.2 | 96.5 | 92.0 | 90.5 | 88.2 | 98.7 | 93.3 | 99.5 | 93.3 | 94.5 |
| AIMv2 | 3B/14 | 96.6 | 99.3 | 93.3 | 83.4 | 95.6 | 84.2 | 90.5 | 87.4 | 96.8 | 90.7 | 99.7 | 90.7 | 92.9 |
| EVA CLIP | 18B/14 | 96.9 | 99.5 | 95.4 | 85.0 | 95.4 | 81.6 | 90.2 | 87.1 | 98.4 | 90.6 | 99.6 | 90.6 | 92.9 |
| *Self-supervised backbones* | | | | | | | | | | | | | | |
| Franca | g/14 | 89.2 | 98.6 | 90.4 | 73.7 | 89.7 | 74.1 | 89.4 | 80.6 | 93.2 | 97.6 | 97.8 | 78.4 | 87.7 |
| DINOv2 | g/14 | 95.6 | 99.5 | 94.5 | 78.9 | 94.6 | 88.5 | 88.4 | 86.8 | 96.8 | 95.9 | 99.7 | 91.6 | 92.6 |
| Web-DINO | 7B/14 | 96.1 | 99.5 | 93.4 | 77.5 | 95.0 | 88.8 | 87.0 | 79.9 | 92.9 | 93.1 | 99.6 | 78.9 | 90.2 |
| DINOv3 | 7B/16 | 96.9 | 99.6 | 96.0 | 81.1 | 95.0 | 88.2 | 88.2 | 87.2 | 97.0 | 94.8 | 99.7 | 92.4 | 93.0 |

Table 28: Full results for instance recognition, presenting additional metrics for Sec. 6.2.2.

| | | Oxford | | Paris | | Met | | | AmsterTime |
|---|---|---|---|---|---|---|---|---|---|
| Method | ViT | M | H | M | H | GAP | GAP- | ACC | mAP |
| *Agglomerative backbones* | | | | | | | | | |
| AM-RADIOv2.5 | g/16 | 70.6 | 47.4 | 92.4 | 82.7 | 30.5 | 65.9 | 69.0 | 45.8 |
| *Weakly-supervised backbones* | | | | | | | | | |
| SigLIPv2 | g/16 | 45.4 | 23.3 | 77.5 | 57.9 | 13.9 | 46.0 | 54.6 | 16.6 |
| PE-core | G/14 | 50.7 | 24.3 | 81.7 | 65.6 | 10.6 | 34.8 | 44.9 | 19.4 |
| AIMv2 | 3B/14 | 35.0 | 16.5 | 73.9 | 52.4 | 29.5 | 67.3 | 69.9 | 14.6 |
| EvaCLIP | 18B/14 | 55.2 | 27.1 | 81.8 | 65.6 | 0.5 | 4.3 | 11.0 | 18.9 |
| *Self-supervised backbones* | | | | | | | | | |
| Franca | g/14 | 44.6 | 14.3 | 73.8 | 51.6 | 27.2 | 54.3 | 57.7 | 21.1 |
| DINOv2 | g/14 | 78.2 | 58.2 | 92.7 | 84.6 | 44.6 | 73.0 | 75.2 | 48.9 |
| Web-DINO | 7B/14 | 64.1 | 31.2 | 89.8 | 80.3 | 36.3 | 68.7 | 72.5 | 30.6 |
| DINOv3 | 7B/16 | 83.2 | 64.5 | 92.0 | 85.4 | 51.1 | 77.5 | 81.9 | 59.4 |

## C.7 Classification on OCR-Heavy Datasets

In this experiment, we evaluate DINOv3 on classification tasks that require some form of character recognition. These tasks include street-sign, logo, and product classification. We compare our model to the self-supervised models DINOv2 (g) and WebDINO (7B) and the best weakly-supervised one (PE-core G).

Table 29: Comparison with state-of-the-art systems on semantic segmentation on other datasets, complementary to the ADE20k results in Tab. 13. We report mIoU scores when evaluating the model in a single- or multi-scale (TTA) setup and compare against the best previously published result for each dataset: Fang et al. (2023) for COCO-Stuff 164k, Wang et al. (2023b) for Cityscapes, and Zoph et al. (2020) for VOC 2012. We use input resolutions of 1280 for COCO-Stuff, 1024 for VOC 2012, and 1280 for Cityscapes. All baselines require finetuning of the backbone, while we keep the DINOv3 backbone frozen.

| Method | FT | COCO-Stuff 164k | | Cityscapes | | VOC 2012 | |
|---|---|---|---|---|---|---|---|
| | | Single | TTA | Single | TTA | Single | TTA |
| Previous Best | 🔥 | 53.7 | 53.7 | **86.3** | **87.0** | — | 90.0 |
| DINOv3 | ❄️ | **53.8** | **54.0** | 86.1 | 86.7 | **90.1** | **90.4** |

Table 30: Comparison of DINOv3 classification performance on OCR-heavy datasets. These are notoriously hard datasets for SSL. We compare DINOv3 with the best DINOv2 model (g), WebDINO (7B), a DINO variant trained on the MetaCLIP dataset, along with the best weakly-supervised PE-core model (G).

| Model | ViT | GTSRB | Logo-2K+ | FlickrLogos-32 | RP2K | Products-10K | SOProducts |
|---|---|---|---|---|---|---|---|
| DINOv2 | g/14 | 78.2 | 52.9 | 83.6 | 91.4 | 70.8 | 57.6 |
| WebDINO | 7B/14 | 90.3 | 77.5 | 92.8 | 92.8 | 70.3 | 55.9 |
| PE-core | G/14 | 94.8 | 93.2 | 99.0 | 93.1 | 80.6 | 80.7 |
| DINOv3-7B | 7B/16 | 87.5 | 86.0 | 86.3 | 94.7 | 74.5 | 65.2 |

Table 31: Geographical fairness and diversity analysis across income buckets and regions, following the protocol of Goyal et al. (2022b).

| | | Income Buckets | | | Regions | | | |
|---|---|---|---|---|---|---|---|---|
| Method | Arch. | low | medium | high | Africa | Asia | Americas | Europe |
| SEERv2 | RG-10B | 59.7 | 78.5 | 86.6 | 65.9 | 76.3 | 81.1 | 85.6 |
| DINOv2 | ViT-g/14 | 67.4 | 83.3 | 90.5 | 74.0 | 81.6 | 86.2 | 89.7 |
| DINOv3 | ViT-7B | 69.6 | 85.7 | 90.9 | 76.7 | 83.7 | 88.0 | 90.7 |

We run this evaluation on images of resolution 512 for our model, and adjust for patch size to match the sequence length for the others. We report the result of this experiment in Tab. 30.

We see that our new model DINOv3 drastically outperforms its predecessor DINOv2. However, the gap with weakly-superpvised models remains large. Since our model does not leverage paired image-text data during training, it has a much harder time learning glyph associations. Recent work introducing the WebDINO model (Fan et al., 2025) hints at the impact of training data on the performance on these type of tasks. Indeed, we find that WebDINO achieves slightly better result than DINOv3 on some datasets (GTSRB, FlickrLogos-32). This could stem from WebDINO being trained on the MetaCLIP dataset (Xu et al., 2024) curated with OCR-type tasks in mind, highlighting a promising avenue for future improvement of SSL methods on OCR tasks.

### C.8 Fairness Analysis

We evaluate the geographical fairness and diversity of the DINOv3 7B model across different income buckets and regions, following the protocol of Goyal et al. (2022b). For reference, we include the results obtained with DINOv2 and SEERv2. The results indicate that DINOv3 delivers somewhat consistent performance across income categories, although there is a notable performance drop of 23% in the low-income bucket compared to the highest-income bucket. The medium and high-income buckets exhibit comparable performance. Regionally, DINOv3 achieves relatively good scores across different regions; however, a relative difference of over 14% is observed between Europe and Africa, which is an improvement over the relative difference of more than 17% seen with DINOv2.

## D Implementation Details

### D.1 Model

We use multi-crop (Caron et al., 2020) with 2 global crops ($256 \times 256$ px) and 8 local crops ($112 \times 112$ px) seen by the student model, resulting in a total sequence length of 3.7M tokens. The teacher EMA (exponential moving average of the student) processes the global crops only. We apply the $\mathcal{L}_{\text{DINO}}$ loss on the class token of all student local crops and both teacher global crops, and between pairs of different global crops for both models. A random proportion in $[0.1, 0.5]$ of the global crops patch tokens seen by the student are masked

with 50% probability, and we apply the $\mathcal{L}_{\text{iBOT}}$ loss between these and the visible tokens seen by the teacher EMA. We apply the $\mathcal{L}_{\text{DKoleo}}$ loss to small batches of 16 class tokens of the first global crop seen by the student. We train for 1M iterations with a global batch size of 4096 using a fully-sharded data-parallel setup in Pytorch, using bfloat16 and 8-bit floating-point matrix multiplications. We use a constant global learning rate of 0.0004 with a warmup of 100k iterations, a weight decay of 0.04, a learning rate decay factor of 0.98 per layer, a stochastic depth (layer dropout) probability of 0.4 and an EMA momentum of 0.994 for the teacher. The remaining hyperparameters can be found in the configuration files in the code release.

For the Gram anchoring step, we use a loss weight of $w_{\text{Gram}} = 2$ and update the Gram teacher every 10k steps for a maximum of three updates, increasing the EMA momentum to 0.999. For high-resolution adaptation (Sec. 5.1), we sample from the following pairs of global/local/Gram teacher crop resolutions with the following probabilities: $(512, 112, 768)$ with $p = 0.3$, $(768, 112, 1152)$ with $p = 0.3$, $(768, 168, 1152)$ with $p = 0.3$, $(768, 224, 1152)$ with $p = 0.05$, and $(768, 336, 1152)$ with $p = 0.05$. These values were obtained empirically.

## D.2 Data Composition

As detailed in Sec. 3.1, we follow the retrieval-based curation pipeline of Oquab et al. (2024) to create the second part of our pretraining data. Given a list of seed datasets that contain useful visual concepts, we retrieve samples from our 17B datapool that are similar to the images in these datasets. In Tab. 32, we provide details about the seed datasets including their size and the number of images retrieved.

During training, we sample batches in two distinct ways: homogeneous batches are drawn exclusively from ImageNet1k, while heterogeneous batches are composed of samples from other data parts, selected according to a multinomial distribution. The specific sampling ratios for each data part are detailed in Tab. 33.

# E   Experimental Details

In this section, we provide detailed descriptions of the datasets and evaluation metrics used across all benchmarks in this paper.

## E.1   Semantic Segmentation: Linear Probing

**Datasets and Metrics**   We evaluate semantic segmentation performance of DINOv3 obtained via linear probing on three benchmark datasets: ADE20k (Zhou et al., 2017), VOC 2012 (Everingham et al., 2012), and Cityscapes (Cordts et al., 2016). The evaluation metric reported is the standard mean Intersection-over-Union (mIoU). Images are processed at resolution $512 \times 512$ for ADE20k and 2012, and $1024 \times 1024$ for Cityscapes, if not mentioned otherwise. Segmentation masks are evaluated at their original resolution.

**Evaluation Protocol**   To assess the quality of the dense features, we train a linear classifier probe on the training set of each benchmark. This linear layer is applied on top of the patch output features (after layer normalization) of the frozen backbone, with the features further normalized using a trained batch normalization layer. The probe is trained for 40000 steps with a batch size of 16 by minimizing the cross entropy loss, using the AdamW optimizer with linear learning rate warmup of 1500 steps followed by cosine annealing. For all backbones, we perform a hyperparameter sweep varying the learning rate over $\{1 \times 10^{-4}, 3 \times 10^{-4}, 1 \times 10^{-3}\}$ and weight decay over $\{1 \times 10^{-4}, 1 \times 10^{-3}\}$. For evaluation, we employ a sliding window strategy; given an input side length of 512, the image is first resized to a short side of 512. The model is then applied to windows of resolution $512 \times 512$ with a stride of 341, averaging the output predictions. The final prediction is then resized to the size of the target mask.

## E.2   Depth Estimation: Linear Probing

**Datasets and Metrics**   We evaluate the quality of DINOv3 features for geometric tasks on the depth benchmarks NYUv2 (Silberman et al., 2012) and KITTI (Geiger et al., 2013) datasets. For NYUv2, we take random crops of size $544 \times 416$ as input during training, and evaluate the full-sized image during evaluation,

| Dataset / Split | Dataset Size | # Images Retrieved |
|---|---|---|
| Caltech 101 / train | 3,030 | 1,000,000 |
| CUB-200-2011 / train | 5,994 | 1,000,000 |
| DTD / train1 | 1,880 | 1,000,000 |
| FGVC-Aircraft / train | 3,334 | 970,000 |
| Flowers-102 / train | 1,020 | 1,000,000 |
| Food-101 / train | 75,750 | 1,000,000 |
| Oxford-IIIT Pet / trainval | 3,680 | 1,000,000 |
| Stanford Cars / train | 8,144 | 1,000,000 |
| SUN397 / train1 | 19,850 | 1,000,000 |
| Pascal VOC 2007 / train | 2,501 | 1,000,000 |
| ADE20K / train | 20,210 | 1,000,000 |
| Cityscapes / train | 2,975 | 600,000 |
| Pascal VOC 2012 (seg.) / trainaug | 1,464 | 1,000,000 |
| KITTI / train (Eigen) | 23,158 | 1,000,000 |
| NYU Depth V2 / train | 24,231 | 1,000,000 |
| SUN RGB-D / train | 4,829 | 1,000,000 |
| Google Landmarks v2 / train (clean) | 1,580,470 | 7,902,350 |
| AmsterTime / new | 1,231 | 270,000 |
| AmsterTime / old | 1,231 | 400,000 |
| Met / train | 397,121 | 1,000,000 |
| Revisiting Oxford / base | 4,993 | 1,000,000 |
| Revisiting Paris / base | 6,322 | 1,000,000 |
| | | 28,542,350 |

Table 32: Composition of the retrieval-based curated part in our pretraining data. We report the list of datasets and associated splits used to build this part. For each dataset, we specify the original dataset size and the number of retrieved images included in the final training data.

| Group | Group Ratio | Part | Part Size | Part Ratio in Group |
|---|---|---|---|---|
| 1 | 10% | ImageNet1k | 1.2M | 100% |
| 2 | 90% | clustering-based curated | 1.7B | 65.0447% |
| | | retrieval-based curated | 28.5M | 22.4343% |
| | | ImageNet22k | 14.2M | 10% |
| | | Mapillary | 1.4M | 1.1214% |

Table 33: Sampling ratios of different data groups and parts in our pre-training data mix.

computing the metrics only on the pre-defined center crop by Eigen et al. (2014). For KITTI, we take random crops of size $704 \times 352$ as input during training, and evaluate the full-sized image during evaluation, computing the metrics only on the pre-defined crop by Garg et al. (2016). Inputs are center padded with black pixels to round up the side lengths to the nearest length admitted by the patch size, and output predictions are bilinearly resized to the resolution of the target depth map. Results are reported using the Root Mean Squared Error (RMSE) metric.

**Evaluation Protocol**  To assess the quality of the dense features, we train a linear probe on the training set of each benchmark. This linear layer is applied on top of the patch output features (after layer normalization) of the frozen backbone, with the features further normalized using a trained batch normalization layer. The probe is trained for 38400 steps with a batch size of 16, using the AdamW optimizer with linear learning rate warmup of 12800 steps followed by cosine annealing. The probe is trained using the scale invariant loss also used in Bhat et al. (2021). For all backbones, we perform a hyperparameter sweep using the AdamW optimizer, varying the learning rate over $[1 \times 10^{-4}, 3 \times 10^{-4}, 1 \times 10^{-3}]$ and weight decay over $[1 \times 10^{-4}, 1 \times 10^{-3}]$.

### E.3   3D Correspondence Estimation

**Datasets and Metrics**  Geometric correspondence is evaluated on the NAVI dataset (Jampani et al., 2023), and semantic correspondence on the SPair dataset (Min et al., 2019). For NAVI, we use images resized to a side length of 448/512 pixels for models with patch size 14/16. For SPAir, we use images resized to a side length of 896/1024 pixels for models with patch size 14/16. To measure performance, we report the correspondence recall, *i.e.* the percentage of correspondences falling into a specified distance.

**Evaluation Protocol**  For NAVI, we follow the protocol defined in Probe3D (Banani et al., 2024). Specifically, we subsample 1/4 of the object views, and for each source view select another dest. view within a maximum rotation of 120 degrees to create an image pair (source, dest.) to perform patch matching on. For each image pair, each patch of source (within the object) is matched to a patch in dest. The top-1000 matches with highest cosine similarity are kept for evaluation, and a 3D distance error is computed for each match based on the known camera pose and depth maps of both images. This allows to compute recall errors with varying thresholds, for which we use thresholds of 1cm, 2cm, and 5cm. We then compute the average recall across thresholds as the correspondence recall.

For each evaluated backbone, we use the features of the final layer, and evaluate them with and without the final layer norm applied. This is because we noticed bad performance for some baseline models when applying the final layer norm. We report the maximum of both results.

### E.4   Unsupervised Object Discovery

**Datasets and Metrics**  For this task, the objective is to generate a single bounding box per image that highlights any object depicted in the scene. We follow the protocol of Siméoni et al. (2021) for unsupervised object discovery and evaluate all backbones on the detection benchmarks VOC07 (Everingham et al., 2007), VOC12 (Everingham et al., 2012), and COCO20K (Lin et al., 2014; Vo et al., 2020). COCO20K is a subset of the COCO2014 trainval dataset (Lin et al., 2014), consisting of 19,817 randomly selected images, as proposed in (Vo et al., 2020) and commonly used for this task. Each image is processed close to its original resolution, resized only to a side length rounded up to the nearest multiple of patch size. For each image, a single bounding box is generated. For evaluation, we use the *Correct Localization* (CorLoc) metric, which computes the percentage of correctly localized boxes. A predicted box is considered correct if its *intersection over union* (IoU) with any ground-truth bounding box exceeds 0.5.

**Evaluation Protocol**  To evaluate the quality of the image encoders, we employ the TokenCut strategy (Wang et al., 2023c). This method organizes image patches into a fully connected graph, where the edges represent similarity scores between pairs of patches, computed using backbone features learned by the transformer. The salient object patches are identified by applying the Normalized Cut algorithm, which solves a graph-cut problem. A bounding box is then fitted to the resulting salient object mask. All images are input to the encoders at their full resolution, and we use the patch output features for all image encoder. To

account for differences in feature distributions among models, we perform a sweep over TokenCut's unique hyperparameter: the similarity threshold used in graph construction. Specifically, we vary the threshold between 0 and 0.4 in increments of 0.05.

### E.5 Video Segmentation Tracking

**Datasets and Metrics**  For this task, we use the DAVIS 2017 (Pont-Tuset et al., 2017), YouTube-VOS (Xu et al., 2018) and MOSE (Ding et al., 2023) datasets. DAVIS defines a training set of 60 videos and a validation set of 30 videos for which all frames are annotated with ground-truth instance segmentation masks. For YouTube-VOS, only the training set is annotated and publicly available, while the validation set is gated behind an evaluation server. To mimic the DAVIS setup, we take a random subset of 2758 annotated videos (80%) as the training set and the remaining 690 videos (20%) as the validation set. In a similar fashion, we split the MOSE dataset into 1206 videos for validation and 301 for testing. We evaluate with different input image sizes, namely 420/480 (S), 840/960 (M), 1260/1440 (L) for models with patch size 14/16. For all datasets, we evaluate performance using the standard $\mathcal{J}\&\mathcal{F}$-mean metric (Perazzi et al., 2016), which combines the region similarity ($\mathcal{J}$) and contour accuracy ($\mathcal{F}$) scores. Only the objects annotated in the first frame are tracked and evaluated, while objects that appear later in the video are ignored, even if their ground-truth masks are annotated.

**Evaluation Protocol**  Similar to Rajasegaran et al. (2025), we implement a non-parametric protocol for label propagation based on patch similarity, which is computed as a cosine similarity between features extracted from a frozen DINOv3 backbone. We assume that the first frame of the video is labeled with instance segmentation masks, which we represent as a one-hot vector per patch. For each frame, we compute the cosine similarity between all its patch features, all patches of the first frame, and all patches of a small number of past frames. Focusing on a single patch in the current frame, we consider the $k$ most similar patches within a spatial neighborhood, and compute a weighted average of their labels to obtain a prediction for the current patch. After processing one frame, we move to the next one, treating the previous predictions as soft instance segmentation labels. When forwarding individual frames through the backbone, we resize the image such that the shortest side matches a certain size, preserving aspect ratio up to the nearest multiple of the patch size.[4] Patch similarity and label propagation are computed at the resolution of the resulting features, then the mask probabilities are bilinearly resized to the native resolution for computing $\mathcal{J}\&\mathcal{F}$. We consider several hyperparameter combinations, *e.g.* the number of past frames to use as context, the number of neighbors $k$, and the size of the spatial neighborhood, as summarized in Tab. 34. We perform hyperparameter selection on the training set of DAVIS, and then apply the best combination to the test splits of all datasets.

### E.6 Video Classification

**Datasets and Metrics**  We evaluate DINOv3 on video classification using the UCF101 (Soomro et al., 2012), Something-Something V2 (Goyal et al., 2017), and Kinetics-400 (Kay et al., 2017) datasets. At a high level, we extract a fixed number of frames from each video, encode them with a frozen backbone, collect all patch features into a flat sequence, which we then feed to a shallow transformer-based classifier trained with regular supervision on a set of labeled videos. In previous work, *e.g.* Assran et al. (2025), this protocol is referred to as an *attentive probe*, a hint to the *linear probes* used for image classification. In the following paragraphs, we describe our implementation of the protocol.

**Training**  At training time, we select 16 frames at random temporal locations from each video, keeping track of the corresponding timestamps. We also sample the parameters of a spatial crop that covers between 40% and 100% of the area–these parameters will be shared across all frames of the video to avoid jittering. We then process each frame with DINOv3 as an independent 256×256 image, extracting 16×16 patch features and discarding the CLS token. For each patch, we keep track of its spatial coordinates defined on a $[0, 1]^2$

---

[4]For example, DAVIS videos are natively 480×854 and we want to process them at resolution 960. For a model with patch size 16, we resize the frames to 960×1712 with a slight horizontal stretch, resulting in a 60×107 feature map. Instead, for a model with patch size 14, we resize the frames to 966×1708 with a slight vertical stretch, resulting in a 69×122 feature map.

Table 34: List of hyperparameters evaluated for video segmentation tracking on the training split of DAVIS 2017 (Pont-Tuset et al., 2017). The best hyperparameters, highlighted, are applied to all datasets.

| Max context length | Neighborhood mask size | Neighborhood mask shape | Top-K | Temperature |
|---|---|---|---|---|
| 7 | 12 | Square | 5 | 0.2 |
| 7 | 12 | Circle | 5 | 0.2 |
| 7 | 5 | Square | 5 | 0.2 |
| 7 | 24 | Square | 5 | 0.2 |
| 7 | $\infty$ | — | 5 | 0.2 |
| 7 | 12 | Square | 5 | 0.01 |
| 7 | 12 | Square | 5 | 0.1 |
| 7 | 12 | Square | 5 | 0.7 |
| 4 | 12 | Square | 5 | 0.2 |
| 10 | 12 | Square | 5 | 0.2 |
| 15 | 12 | Square | 5 | 0.2 |
| 7 | 12 | Square | 3 | 0.2 |
| 7 | 12 | Square | 10 | 0.2 |
| 7 | 12 | Square | 15 | 0.2 |
| 15 | 12 | Circle | 10 | 0.1 |
| 15 | 24 | Circle | 10 | 0.1 |
| 15 | 36 | Circle | 10 | 0.1 |
| 15 | $\infty$ | — | 10 | 0.1 |

box. The patch features from all frames are linearly projected to 1024 dimensions, concatenated into a flat sequence of length $16 \times 16 \times 16 = 4096$, and then fed to four self-attention blocks that model the spatial and temporal relationships between the patches. To ensure the model has access to positional information, we inject the timestamp and spatial coordinates of each patch both as an additive sin-cos embedding before the blocks (Vaswani et al., 2017), and as a 3D factorized RoPE with random spatial rotations in each attention head (Heo et al., 2024). After the four blocks, we apply a cross-attention block with a single position-less learnable query to aggregate the information from all patches into a single vector, which is then linearly projected to obtain the final classification logits. The stack of self-attention blocks, the cross-attention block, the positional embeddings, and the final projection constitute the video classifier, which we train for 20 epochs with batch size 64 with a standard cross-entropy loss. In practice, we train a set of classifiers in parallel, one for each combination of learning rate $\{1 \cdot 10^{-4}, 2 \cdot 10^{-4}, 5 \cdot 10^{-4}, 1 \cdot 10^{-3}, 2 \cdot 10^{-3}, 5 \cdot 10^{-3}\}$ and weight decay $\{10^{-3}, 10^{-2}, 10^{-1}\}$. For each dataset, we use 90% of the training set to update the model parameters, 10% of the training set to choose the best combination of learning rate and weight decay, and finally report performance of the chosen model on the validation split.

**Inference** At inference time, we follow a deterministic strategy to sample a single clip per video: we take the first frame, the last frame, and uniformly-spaced frames in between, for a total of 16. From each frame, we crop the largest center square and resize it to $256 \times 256$ pixels, possibly losing information from the sides of rectangular videos. We then feed these frames to DINOv3 and to the classifier to obtain a prediction for the video. Alternatively, we follow Assran et al. (2025) and perform test-time augmentation (TTA) by selecting multiple frame sequences and multiple spatial crops, processing them independently and then averaging the class probabilities to obtain the final prediction. Clip sampling is exemplified in Fig. 22.

**Baselines** For the chosen baseline models, we use the same evaluation protocol, *i.e.* feature extraction, classifier architecture, training procedure and inference protocol, with a few differences. The input resolution is $256 \times 256$ pixels for models that use patch size 16, and $224 \times 224$ pixels for patch size 14. This way, all backbones yield an identical number of tokens, and therefore afford the same amount of computation in the classifier. All models process videos frame by frame independently, since they are trained on images. The only exception is V-JEPA 2, to which we feed whole clips to extract time-aware features. Since V-JEPA 2

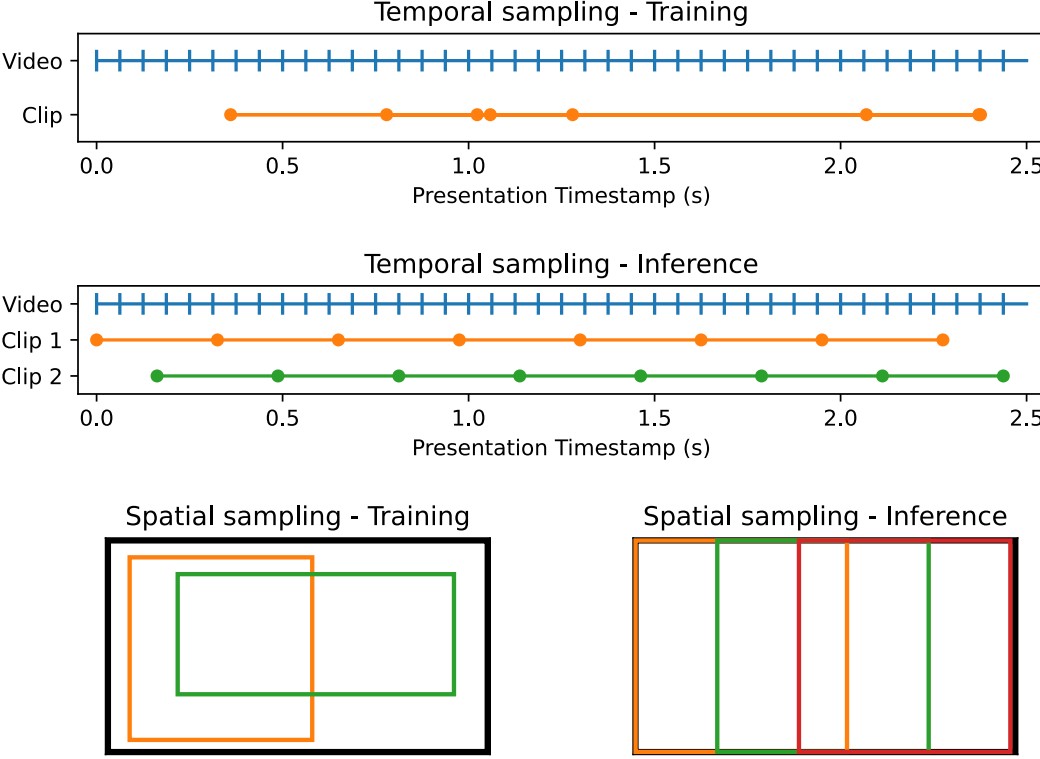

Figure 22: Sampling clips for video classification. Choosing a clip for training or inference means determining the coordinates of a spatial crop and which frames/timestamps to sample. At training time, we sample clips at random by choosing random frames from the whole video and by applying a spatial crop that covers $\geq 40\%$ of the area. At inference time, we select clips in a deterministic way. Spatially, we take the three largest square crops aligned to the left, middle and right. Temporally, we take two overlapping sets of frames such that they cover as much of the video as possible and their timestamps interleave.

reduces the temporal axis by a factor of two, *e.g.* yielding 8 time steps given 16 frames as input, we duplicate each patch token to match the sequence length of other models.

### E.7  Image Classification with Linear Probing

**Datasets and Metrics**  We evaluate the global quality of the DINOv3 model using the widely adopted linear probing evaluation. We train a linear transform on the training set of ImageNet-1k (Deng et al., 2009) and evaluate results on the val set. We assess the generalization quality of the model by evaluating the transfer to classification test sets: ImageNet-**V2** (Recht et al., 2019) and **ReaL** (Beyer et al., 2020), which provide alternative sets of images and labels for ImageNet, designed to test for overfitting on the original ImageNet validation set. Additionally, we consider the **R**endition (Hendrycks et al., 2021a) and **S**ketch (Wang et al., 2019) datasets, which present stylized and artificial versions of ImageNet classes; the **A**dversarial (Hendrycks et al., 2021b) and **Obj**ectNet (Barbu et al., 2019) datasets, which contain deliberately challenging examples; and the **C**orruptions (Hendrycks & Dietterich, 2019) dataset, which measures robustness to common image corruptions. We report top-1 classification accuracy as the evaluation metric for all datasets but ImageNet-C, for which we report the mean corruption error (mCE, see (Hendrycks & Dietterich, 2019)).

For fine-grained datasets, we consider the same collection of 12 smaller datasets from Oquab et al. (2024), which we refer to as "sFine" here: Food-101 (Bossard et al., 2014), CIFAR-10 (Krizhevsky, 2009), CIFAR-100 (Krizhevsky, 2009), SUN397 (Xiao et al., 2010), StanfordCars (Krause et al., 2013), FGVC-Aircraft (Maji et al., 2013), VOC 2007 (Everingham et al., 2007), DTD (Cimpoi et al., 2014), Oxford Pets (Parkhi et al., 2012), Caltech101 (Fei-Fei et al., 2004), Flowers (Nilsback & Zisserman, 2008), and CUB200 (Welinder et al., 2010), as well as the larger datasets Places205 (Zhou et al., 2014), iNaturalist 2018 (Van Horn et al., 2018), and iNaturalist 2021 (Van Horn et al., 2021).

**Evaluation Protocol**   For the larger datasets ImageNet, Places205, iNaturalist 2018 and iNaturalist 2021, we use the following procedure. For each baseline, we train a linear layer on the final features of the CLS token (after the layer norm) using the ImageNet-1k training set (Deng et al., 2009). Specifically, we use SGD with a momentum of 0.9, and train for 10 epochs with a batch size of 1024. We sweep the learning rates $\{1 \times 10^{-4}, 2 \times 10^{-4}, 5 \times 10^{-4}, 1 \times 10^{-3}, 2 \times 10^{-3}, 5 \times 10^{-3}, 1 \times 10^{-2}, 2 \times 10^{-2}, 5 \times 10^{-2}, 1 \times 10^{-1}, 2 \times 10^{-1}, 5 \times 10^{-1}, 1 \times 10^{0}, 2 \times 10^{0}, 5 \times 10^{0}\}$ and weight decay values $\{0, 1e-5\}$ and use the validation set of ImageNet-1k to select the best combination. During training, we use random resize crop augmentation with standard Inception-crop parameters. For the datasets in sFine, following Oquab et al. (2024), we use a more lightweight evaluation using scitkit-learn's LogisticRegression implementation with the L-BFGS solver.

In both cases, we evaluate models at resolutions resulting in 1024 patch tokens, that is, $448 \times 448$ for patch size 14, and $512 \times 512$ for patch size 16. The images are resized such that the shorter side matches the chosen side length, then take the central square crop.

### E.8   Image Classification with k-Nearest Neighbors

**Datasets and Metrics**   We also evaluate DINOv3 on ImageNet classification using a non-parametric approach based on $k$-Nearest Neighbors ($k$-NN). We report top-1 classification accuracy.

**Evaluation Protocol**   For each image in the test set, we retrieve the $k$ nearest neighbors in the training set based on the cosine similarity between CLS tokens. The classification decision is based on majority voting over classes associated with the retrieved images in the training set. We use $k = 10$.

### E.9   Instance Recognition

**Datasets and Metrics**   We use the Oxford and Paris datasets for landmark recognition (Radenović et al., 2018), the Met dataset featuring artworks from the Metropolitan Museum (Ypsilantis et al., 2021), and AmsterTime, which consists of modern street view images matched to historical archival images of Amsterdam (Yildiz et al., 2022). In Tab. 11, we report mean average precision (mAP) for Oxford-Hard, Paris-Hard, and AmsterTime, and global average precision (GAP) for Met. In Tab. 28, we additionally give mAP for Oxford-Medium and Paris-Medium, as well as the additional metrics GAP- and accuracy (see (Ypsilantis et al., 2021)). For Oxford and Paris, we resize all images such that the larger side is 224/256 pixels long for models with patch size 14/16, keeping the aspect ratio, then pad the image with zeros on the shorter side using torchvision's CenterCrop operation to yield an image of resolution of $224 \times 224$ / $256 \times 256$. For AmsterTime, we resize all images such that the shorter side is 256/292 pixels long, keeping the aspect ratio, then take a square center crop with side length 224/256 for models with patch size 14/16. For Met, we evaluate all images close to their original resolution, resizing only to a side length rounded up to the nearest multiple of patch size (resulting in a long side of 504/512 pixels for patch size 14/16 as the original long side is always 500 pixels).

**Evaluation Protocol**   The image similarity is computed using cosine distance between the CLS tokens computed for query and target images. We follow the evaluation protocols of (Radenović et al., 2018) for Oxford and Paris, of (Yildiz et al., 2022) for AmsterTime, and of Ypsilantis et al. (2021) for Met. For Met, this includes tuning the hyperparameters k and $\tau$ with a grid search, optimizing GAP on the validation set of Met, and whitening the features using a PCA estimated on the training set of Met.

### E.10 Object Detection

**Datasets and Metrics**   We evaluate DINOv3 on object detection on the COCO (Lin et al., 2014) and COCO-O (Mao et al., 2023) datasets. COCO is a standard benchmark for object detection, covering 80 object categories, and containing 118k training images and 5k validation images. COCO-O is an evaluation-only dataset with the same categories as COCO, but in more challenging visual conditions, such as scenes with significant occlusions, cluttered backgrounds, and varying lighting conditions. For training the object detection model, we also leverage the Objects365 (Shao et al., 2019) dataset, which contains 2.5M images and covers 365 object categories, a subset of which maps directly to COCO classes. For both COCO and COCO-O, we report mean Average Precision (mAP) computed at IoU thresholds of $[0.5 : 0.05 : 0.95]$.

**Architecture**   Our approach builds upon the Plain-DETR (Lin et al., 2023b) implementation, with several modifications. We do not fuse the transformer encoder into the *backbone*, but keep it as a separate module, similar to the original DETR (Carion et al., 2020). This allows us to keep the DINOv3 backbone completely frozen during training and inference, making it the first competitive detection model to do so. From a DINOv3 ViT-7B/16 backbone, we extract features from four intermediate layers, namely $[10, 20, 30, 40]$. For each patch, we concatenate intermediate features channel-wise, giving a feature dimension of $4 \cdot 4096 = 16384$, which is further increased by the windowing strategy described below. The backbone features feed into the *encoder*, which is a stack of 6 self attention blocks with embedding dimension of 768. The *decoder* is a stack of 6 cross attention blocks with the same embedding dimension, where 1500 "one-to-one" queries and 1500 "one-to-many" queries attend to the patch tokens of the encoders to predict bounding boxes and class labels.

**Image Pre-Processing**   Training is performed in three stages as described below, one with a base image resolution of 1536 pixels and two with a base resolution of 2048. Following DETR, we apply random horizontal flipping ($p = 0.5$), followed by either (i) random resizing, where the shortest side is uniformly sampled between 920 pixels (resp. 1228) and the base resolution of the stage (1536 or 2048), or (ii) a random crop retaining 60–100% of the original image area, followed by resizing as in (i). At evaluation time, images are resized so that the shortest side is 2048 without additional augmentation, and both sides are rounded up to the nearest multiple of the patch size.

**Windowing Strategy**   We then apply a windowing strategy that combines a global view of the image with smaller views, to allow the backbone to process objects at all scales. The number of windows is fixed to $3 \times 3$, and their sizes vary according to the input resolution. As an example, for an image of size $1536 \times 2304$:

1. The image is divided into $3 \times 3$ non-overlapping windows of size $512 \times 768$. Each window is forwarded through the backbone, resulting in $32 \times 48$ patch tokens of dimension 16384. The features of all windows are spatially reassembled into a $(3 \cdot 32) \times (3 \cdot 48)$ feature map.

2. The whole image is resized to $512 \times 768$ and forwarded through the backbone, resulting in a feature map of $32 \times 48$ patch tokens of dimension 16384. These features are then bilinearly upsampled to $96 \times 144$, matching the size of the windowed feature map.

3. Finally, the features maps from steps 1 and 2 are concatenated channel-wise, resulting in a $96 \times 144$ feature map of dimension $2 \cdot 16384 = 32768$. This feature map is then flattened as a sequence of $96 * 144$ tokens and fed to the encoder.

**Training**   We follow a training curriculum in three stages, using the Objects365 dataset (Shao et al., 2019) and the COCO dataset (Lin et al., 2014) at increasing resolutions. Throughout training, we use the AdamW optimizer (Loshchilov & Hutter, 2017) with a weight decay of 0.05. Following DETR, we use the Focal Loss (Lin et al., 2018) as classification loss, with a weight of 2, L1 loss as bounding box loss with a weight of 1, complemented by the GIoU (Rezatofighi et al., 2019) loss with a weight of 2. The stages are as follows:

1. We begin training on Objects365 at base resolution 1536 pixels. We train for 22 epochs with global batch size of 32, which we distribute over 32 GPUs. After an initial warmup of 1000 steps, the learning rate is set to $5 \cdot 10^{-5}$ and is divided by 10 after the 20th epoch.

2. We then continue training on Objects365 at base resolution 2048 pixels. We train for for 4 epochs with learning rate $2.5 \cdot 10^{-5}$.

3. We finish training by doing 12 epochs on COCO at base resolution 2048. After a linear warmup of 2000 iterations, the learning rate follows a cosine decay schedule, starting at $2.5 \cdot 10^{-5}$ and reaching $2.5 \cdot 10^{-6}$ at the 8th epoch. In this part we use the IA-BCE classification loss (Cai et al., 2024) instead of the simple Focal Loss from DETR. We observed this loss to increase the model performance at transfer time, but not at pretraining time. As this loss mixes class and box information, it brings its full potential if the box predictions are already well initialized. The GIoU loss weight is set to 4 in this part to encourage better box alignment.

**Test-Time Augmentation**  At test time, we follow the inference procedure described above, resizing images such that the short side is 1536 or 2048. At those resolutions, the COCO mAP is 65.4 and 65.6, respectively. Alternatively, we can apply the test-time augmentation (TTA) strategy from Bolya et al. (2025), which consists in flipping and resizing the image to multiple resolutions, and merging the predictions with SoftNMS (Bodla et al., 2017). Specifically, each image is processed at resolutions of $[1536, 1728, 1920, 2112, 2304, 2496, 2688, 2880]$, yielding an mAP of 66.1.

### E.11 Semantic Segmentation

**Datasets and Metrics**  We evaluate DINOv3 on semantic segmentation on the ADE20k (Zhou et al., 2017), Cityscapes (Cordts et al., 2016), COCO-Stuff (Caesar et al., 2018), and VOC 2012 (Everingham et al., 2012) datasets. ADE20k is a widely used benchmark for semantic segmentation with 150 semantic categories, varying from outdoor scenery to images of people and objects inside a house. In addition, COCO-Stuff and Hypersim (Roberts et al., 2021) datasets are used for pre-training the model. COCO-Stuff is a larger dataset (118k training images) than ADE20k containing 80 thing classes and 91 stuff classes, while Hypersim is a photorealistic synthetic dataset presenting indoor scenes with 40 semantic categories, with sharper and more accurate annotations. More than half of the Hypersim images contain 21 or more objects, making it a good candidate for helping the model learn rich information of the scenes. The evaluation metric reported is mIoU for all datasets.

**Architecture**  We adapt the ViT-Adapter and Mask2former configurations that other baselines use (Wang et al., 2023a), with several differences. First, to ensure that our backbone remains frozen and its activations are not altered, we remove the injector component of the ViT-Adapter. This makes our backbone output features to be directly used in the extractor module. Second, the embedding dimensions in the Mask2former decoder are scaled to 2048 instead of the default 1024 to adapt to our backbone output dimension of 4096, while other baselines' backbones usually present an output dimension of 1024 or 1536. As inputs to the decoder, we extract features from four intermediate layers of the DINOv3 7B/16 backbone, namely layers $[10, 20, 30, 40]$. We apply the final layer norm to the features of all layers and add a learned batch normalization.

**Training Protocol**  For generating results on COCO-Stuff, we train the model using a cosine scheduler, with a 6k linear warmup and a maximum learning rate of 1.5e-5. The model is trained for 80k iterations, at resolution 1280 pixels. As for training on the other datasets—ADE20k, Cityscapes and VOC 2012—we first pre-train the decoder on COCO-Stuff for 80k iterations, with a 6k linear warmup and a learning rate of 1.5e-5, following a cosine scheduler. This helps the model learn diverse semantic categories (171 categories) on a larger dataset than ADE20k. The model is then trained on Hypersim for 10k iterations at a learning rate of 2.5e-5 following a cosine scheduler with a 1.5k linear warmup. Corresponding to roughly 2 epochs, this step helps our model learn high-quality image-to-mask correspondence due to their photorealistic synthetic nature. Finally, our model is trained on ADE20k for 20k iterations with a learning rate of 3e-5, again with a 1.5k linear warmup and a cosine schedule. We report our final result on the validation set. For Cityscapes and VOC 2012, learning rates of 1.5e-5 and 1e-5 are used respectively. For all training, a batch size of 16 and the AdamW optimizer is used.

**Inference**  For single-scale evaluation, sliding inference is used for evaluating the models. First, the image is resized to the training resolution (*e.g.* a $400 \times 500$ image will be resized to $896 \times 1120$ pixels for ADE20k, since the model was trained at resolution 896). Then, a sliding window approach with a stride (*e.g.* stride of 596 pixels for ADE20k) is used on square crops (*e.g.* $896 \times 896$ pixels) to generate a prediction for each crop, sliding through the image. The results are then aggregated by averaging and rescaled to the original image size to generate a final prediction. For test-time augmentation, both ADE20k and VOC 2012 images are rescaled to ratios of [0.9, 0.95, 1.0, 1.05, 1.1] of the evaluation resolution, and each image is also flipped horizontally to generate a total of 10 predictions per sample. After sliding inference on each image, the results are rescaled to the original image shape and averaged. COCO-Stuff 164K's TTA mIoU was obtained by simply using an additional horizontally flipped image per sample, and for Cityscapes, ratios of [1.0, 1.5, 2.0] of the evaluation resolution were applied.

## E.12  Monocular Depth Estimation

**Implementation Details**  Our approach differs from Depth Anything v2 (DAv2) (Yang et al., 2024b) primarily in the configuration of image resolution, which is set to $768 \times 1024 pixels$, and the network architecture. The backbone is kept frozen throughout training, while a dropout rate of 0.05 is applied at the end of the DPT head (Ranftl et al., 2021). As input to the decoder, we extract features from four intermediate layers of the DINOv3 7B/16 backbone, namely layers $[10, 20, 30, 40]$. We apply the final layer norm to the features of all layers and add a learned batch normalization. The depth estimation output is discretized into 256 uniformly distributed bins covering the range from 0.001m to 100m. Training employs a base learning rate of 1e-3, scheduled using PolyLR with a power of 3.5 and an initial linear warmup phase lasting 12k iterations. To enhance robustness and generalization, we apply a suite of augmentations: Gaussian blur, Gaussian noise, AutoContrast, AutoEqualise, ColorJitter, rotation, and left-right flip.

**Datasets and Metrics**  We train the model on the dataset of DAv2, which consists of synthetically generated images from the IRS, TartanAir, BlendedMVS, Hypersim, and VKITTI2 datasets. We evaluate on five datasets: NYUv2 (Silberman et al., 2012), KITTI (Geiger et al., 2013), ETH3D (Schöps et al., 2017), ScanNet from (Ke et al., 2025), and DIODE (Vasiljevic et al., 2019). We adopt the zero-shot scale-invariant depth setup, and report the standard metrics absolute relative error and $\delta_1$ (see (Yang et al., 2024a)).

## E.13  Visual Geometry Grounded Transformer with DINOv3

**Implementation Details**  Compared to the original VGGT (Wang et al., 2025), we adopt the following changes: (1) we use an image size of 592 instead of 518; this is to match the number of patch tokens that DINOv2 produces, (2) adopting a smaller learning rate, specifically from 0.0002 to 0.0001, and (3) using a concatenation of the four intermediate layers of DINOv3 ViT-L rather than just the last layer as input to the downstream modules. Interestingly, we found that using four intermediate layers brings a benefit for DINOv3, whereas doing the same for DINOv2 brings no additional performance gains. We also experimented with a version closer to the original VGGT setup (image size 512, same learning rate, final layer), and already found this untuned version to improve over the original VGGT work across all tested benchmarks.

## E.14  Zero-shot inference with DINOv3-based `dino.txt`

### E.14.1  Zero-shot Classification and Retrieval

**Implementation Details**  We evaluate the zero-shot inference performance of our DINOv3 `dino.txt` model and other text-aligned encoders using the global representation for each image. For `dino.txt`-based approaches, this global representation is formed by concatenating the `[CLS]` token and the average of patch embeddings at the model's output. For other models, we simply take the output `[CLS]` token. We then compare these image representations to the text queries encoded by the respective text encoders. To ensure a fair comparison between encoders, we standardize the sequence length to 576 across all models. Specifically, input images are resized to 336 pixels for models with a patch size of 14, and to 384 pixels for models with a patch size of 16.

**Datasets and Metrics: Classification** For zero-shot classification, we follow the protocol introduced in CLIP, using the ImageNet-1K (Krizhevsky et al., 2012) (IN1K), ImageNet-Adversarial (Hendrycks et al., 2021b) (A), ImageNet-Rendition (Hendrycks et al., 2021a) (R), and ObjectNet (Barbu et al., 2019) (Obj.) benchmarks. At test time, for all baselines, we input the class names into the text encoder to obtain text embeddings. We then compute the cosine similarity between these text embeddings and the global descriptors produced by the image encoder, and return the class name that yields the highest similarity. Classification accuracy is used as the evaluation metric.

**Datasets and Metrics: Retrieval** We evaluate image-text retrieval on the cross-modal retrieval benchmarks COCO2017 (Tsung-Yi et al., 2017). The dataset contains pairs of images and their corresponding descriptive captions. The retrieval task is performed in both directions: finding the most similar image of a given text query (T → I), and conversely, retrieving the most relevant caption of a given image (I → T). Performance is measured using the Recall@1 metric, which is 1 if the top-ranked retrieved item matches the ground-truth pair, and 0 otherwise.

### E.14.2 Open-vocabulary segmentation

**Implementation Details** To obtain pixel-level features that are aligned with text, we employ different strategies depending on the model. For the `dino.txt` models, pixel-text alignment is achieved inherently, allowing us to use the output patch features directly and compare them to the segment of the text embedding that was aligned with the average patch during training. In contrast, most general-purpose image-text encoders (Radford et al., 2021; Zhai et al., 2023; Sun et al., 2023; 2024; Fang et al., 2024a; Bolya et al., 2025) do not apply supervisory signals to the final patch embeddings, which often results in poor-quality output patches. To address this limitation, we adopt the MaskCLIP (Zhou et al., 2022) strategy for evaluating these models on segmentation tasks. MaskCLIP extracts patch embeddings in the aligned space by forwarding the value embeddings from the last attention layer, effectively bypassing the final attention operation.

**Datasets and Metrics** We assess the performance of text-aligned models on the open-vocabulary segmentation task using the ADE20K (Zhou et al., 2019) (ADE20K), Cityscapes (Cordts et al., 2016) (City.), COCO-Stuff (Caesar et al., 2018) (Stuff), and PASCAL Context (Mottaghi et al., 2014) (C59) datasets, employing mean intersection-over-union (mIoU) as the evaluation metric. For evaluation, we adopt the sliding window protocol from TCL (Cha et al., 2023). To ensure fair comparison across models with different patch sizes, we standardize the sequence length by adjusting both the image resizing and sliding window parameters. For models with a patch size of 14, input images are resized so that their shorter side is 448 pixels, and we use a window size of (336, 336) with a stride of 168. For models with a patch size of 16, images are resized to a shorter side of 512 pixels, with a window size of (384, 384) and a stride of 192.

### E.15 Geospatial

**Evaluation details** In all of the evaluations, we keep the backbone frozen and only train lightweight classifiers or decoders that are specialized for the tasks. For GEO-Bench classification, we train a linear classifier for 2400 iterations with a batch size of 32. We use SGD optimizer, cosine learning rate annealing, and select the best learning rate between 1e−5 and 1. Unless otherwise specified, segmentation evaluations use a DPT decoder (Ranftl et al., 2021), with a learning rate selected based on performance on the validation set with a grid search of four values in [3e−5, 1e−4, 3e−4, 1e−3].

On LoveDA and iSAID datasets, we train an UPerNet decoder (Xiao et al., 2018) for 80k iterations, with a batch size of 8, and a linear warm-up of 1500 iterations in line with (Wang et al., 2024a). All other hyperparameters such as crop size and weight decay are the same as in (Wang et al., 2024a). Following previous work (Tolan et al., 2024; Wang et al., 2022a), we use a DPT head for canopy height prediction evaluations and train a Faster RCNN (Ren et al., 2015) detector for 12 epochs for object detection tasks.

**Satlidar dataset** The Satlidar dataset consists in one Million of 512×512 Maxar images and corresponding dense lidar measurements collected from different locations as described in Tab. 35. The images were extracted from larger tiles, the numbers of tiles for each sub-dataset are specified in the table.

Table 35: Description of the Satlidar dataset.

| Subdataset | Path | Amount of tiles | Purpose |
|---|---|---|---|
| Kalimantan | https://daac.ornl.gov/cgi-bin/dsviewer.pl?ds_id=1540 | 86 | train/val/test |
| OpenDC | https://opendata.dc.gov/datasets/2020-lidar-classified-las/about | 68 | train/val/test |
| Brazil | https://daac.ornl.gov/cgi-bin/dsviewer.pl?ds_id=1644 | 37 | train/val/test |
| Mozambique | https://daac.ornl.gov/cgi-bin/dsviewer.pl?ds_id=1521 | 144 | train/val/test |
| Neon | https://data.neonscience.org/data-products/DP3.30015.001 | 5366 | train/val/test |
| CA20Graup | https://portal.opentopography.org/datasetMetadata?otCollectionID=OT.092021.6339.1 | 99 | train/val/test |
| CA17Duvall | https://portal.opentopography.org/datasetMetadata?otCollectionID=OT.042020.6339.2 | 56 | train/val/test |
| Netherlands | https://geotiles.citg.tudelft.nl/ | 13 | train/val/test |
| Sao Paulo | https://daac.ornl.gov/CMS/guides/LiDAR_Forest_Inventory_Brazil.html | 4 | test |
| CA brande | https://doi.org/10.5069/G9C53J18 | 1 | test |

# F   Reference Results for DINOv3 Family of Models

We provide reference results for the full DINOv3 family of models on global and dense tasks in Tabs. 36 and 37.

# G   Memory usage for model inference

We provide memory usage numbers for inference in float32 precision for our family of models. The total usage corresponds to the amount allocated for weight parameters, and buffer memory required for intermediate activations during inference computation. We measure these values with PyTorch on an Nvidia GPU and report the results in Tab. 38.

Table 36: Overview of benchmark results for global tasks for the DINOv3 family of models.

| Benchmark | Res. | Metric | ViT-7B | ViT-H+ | ViT-L | ViT-B | ViT-S+ | ViT-S | CNX-L | CNX-B | CNX-S | CNX-T |
|---|---|---|---|---|---|---|---|---|---|---|---|---|
| *k-NN Classification* | | | | | | | | | | | | |
| IN-1k val | 256 | T1 Acc | 85.3 | 85.7 | 85.3 | 83.2 | 80.9 | 79.3 | 82.6 | 82.3 | 81.8 | 78.9 |
| IN-1k val | 512 | T1 Acc | 85.1 | 85.4 | 85.4 | 83.5 | 82.2 | 80.8 | 84.0 | 83.7 | 82.8 | 80.5 |
| *Linear Classification: ImageNet and Domain Generalization (Sec. 6.2.1)* | | | | | | | | | | | | |
| IN-1k val | 256 | T1 Acc | 88.7 | 88.1 | 87.3 | 84.9 | 81.8 | 80.2 | 85.8 | 84.6 | 83.8 | 81.3 |
| IN-1k val | 512 | T1 Acc | 88.4 | 87.9 | 87.4 | 85.3 | 83.0 | 82.0 | 86.4 | 85.4 | 84.6 | 82.7 |
| IN-V2 | 256 | T1 Acc | 81.5 | 80.5 | 80.2 | 75.7 | 71.7 | 69.7 | 77.2 | 75.5 | 73.8 | 71.0 |
| IN-V2 | 512 | T1 Acc | 81.3 | 80.6 | 80.1 | 76.5 | 72.8 | 71.4 | 78.0 | 76.6 | 75.1 | 72.6 |
| IN-ReaL | 256 | T1 Any Acc | 90.6 | 90.4 | 90.2 | 88.9 | 87.1 | 85.9 | 88.9 | 88.5 | 87.9 | 86.6 |
| IN-ReaL | 512 | T1 Any Acc | 90.3 | 90.3 | 90.2 | 89.3 | 88.0 | 87.0 | 89.4 | 89.2 | 88.7 | 87.7 |
| IN-R | 256 | T1 Acc | 90.7 | 90.2 | 87.8 | 76.5 | 65.4 | 59.8 | 81.3 | 77.2 | 73.7 | 66.7 |
| IN-R | 512 | T1 Acc | 91.0 | 90.0 | 88.1 | 76.7 | 66.8 | 60.4 | 82.4 | 78.2 | 74.1 | 66.8 |
| IN-S | 256 | T1 Acc | 70.8 | 70.5 | 67.6 | 58.9 | 50.6 | 45.9 | 63.1 | 60.4 | 57.8 | 52.2 |
| IN-S | 512 | T1 Acc | 71.2 | 70.0 | 67.8 | 58.5 | 51.0 | 45.4 | 63.4 | 60.4 | 57.3 | 51.1 |
| IN-A | 256 | T1 Acc | 86.0 | 83.5 | 78.1 | 57.9 | 34.0 | 26.3 | 63.3 | 53.3 | 44.9 | 28.2 |
| IN-A | 512 | T1 Acc | 86.9 | 84.6 | 82.0 | 68.4 | 51.6 | 44.0 | 70.1 | 62.0 | 55.7 | 40.3 |
| IN-C | 256 | mCE ↓ | 19.8 | 21.4 | 26.1 | 38.5 | 48.2 | 52.8 | 30.8 | 36.2 | 41.1 | 52.1 |
| IN-C | 512 | mCE ↓ | 19.6 | 21.6 | 25.5 | 39.6 | 47.4 | 53.2 | 31.1 | 37.1 | 43.0 | 55.1 |
| ObjectNet | 256 | T1 Any Acc | 75.5 | 75.3 | 70.3 | 58.4 | 46.2 | 42.7 | 59.3 | 56.2 | 52.6 | 45.3 |
| ObjectNet | 512 | T1 Any Acc | 78.9 | 78.6 | 74.8 | 64.1 | 54.6 | 50.9 | 65.2 | 61.3 | 58.7 | 51.0 |
| *Linear Classification: Finegrained Classification (Sec. 6.2.1)* | | | | | | | | | | | | |
| Places205 | 256 | T1 Acc | 70.5 | 69.3 | 68.5 | 66.3 | 63.6 | 62.6 | 66.7 | 65.6 | 64.7 | 63.5 |
| Places205 | 512 | T1 Acc | 70.0 | 69.0 | 68.2 | 66.3 | 64.3 | 63.4 | 67.4 | 66.0 | 65.5 | 64.3 |
| iNaturalist18 | 256 | T1 Acc | 83.7 | 81.3 | 80.8 | 76.2 | 68.6 | 66.5 | 67.1 | 64.7 | 62.4 | 59.2 |
| iNaturalist18 | 512 | T1 Acc | 85.6 | 83.1 | 83.5 | 80.1 | 75.5 | 73.9 | 67.2 | 66.2 | 66.1 | 64.5 |
| iNaturalist21 | 256 | T1 Acc | 88.5 | 85.6 | 85.2 | 81.2 | 74.3 | 72.8 | 72.7 | 69.8 | 67.2 | 64.2 |
| iNaturalist21 | 512 | T1 Acc | 89.8 | 87.0 | 87.6 | 84.9 | 80.6 | 79.3 | 74.0 | 72.5 | 71.4 | 70.5 |
| sFine | 256 | Avg T1 Acc | 93.3 | 92.6 | 92.5 | 91.4 | 89.9 | 88.8 | 89.9 | 89.3 | 89.0 | 88.3 |
| sFine | 512 | Avg T1 Acc | 93.0 | 92.3 | 92.5 | 91.6 | 90.4 | 89.6 | 90.4 | 89.9 | 89.7 | 89.1 |
| Food | 256 | T1 Acc | 96.8 | 96.5 | 96.1 | 94.4 | 92.1 | 90.7 | 94.6 | 93.4 | 92.2 | 90.3 |
| Food | 512 | T1 Acc | 96.9 | 96.7 | 96.3 | 94.8 | 93.2 | 92.1 | 95.2 | 94.1 | 93.4 | 91.8 |
| C10 | 256 | T1 Acc | 99.6 | 99.4 | 99.3 | 98.5 | 97.6 | 97.4 | 98.8 | 98.0 | 98.0 | 96.4 |
| C10 | 512 | T1 Acc | 99.6 | 99.4 | 99.3 | 98.5 | 97.9 | 97.5 | 98.6 | 98.1 | 97.0 | 95.1 |
| C100 | 256 | T1 Acc | 95.8 | 94.9 | 93.7 | 90.6 | 88.6 | 87.2 | 91.5 | 90.0 | 88.9 | 85.5 |
| C100 | 512 | T1 Acc | 96.0 | 95.1 | 94.1 | 91.1 | 88.6 | 87.7 | 91.5 | 89.3 | 86.7 | 83.0 |
| SUN | 256 | T1 Acc | 81.7 | 80.7 | 80.3 | 78.4 | 75.6 | 74.6 | 77.8 | 77.2 | 77.1 | 75.6 |
| SUN | 512 | T1 Acc | 81.1 | 80.7 | 80.2 | 78.7 | 75.8 | 75.0 | 80.0 | 78.9 | 78.4 | 77.2 |
| Cars | 256 | T1 Acc | 94.7 | 94.4 | 94.5 | 94.2 | 93.0 | 91.9 | 91.6 | 91.3 | 91.1 | 91.4 |
| Cars | 512 | T1 Acc | 95.0 | 94.5 | 94.6 | 94.5 | 93.5 | 92.4 | 92.3 | 91.9 | 92.3 | 92.1 |
| Aircraft | 256 | T1 Acc | 88.6 | 85.6 | 85.3 | 85.6 | 82.0 | 79.8 | 77.3 | 78.4 | 76.6 | 77.0 |
| Aircraft | 512 | T1 Acc | 88.2 | 85.7 | 87.6 | 87.6 | 85.5 | 84.2 | 78.8 | 78.7 | 79.1 | 79.7 |
| VOC | 256 | T1 Acc | 89.2 | 88.7 | 89.4 | 89.1 | 87.7 | 87.2 | 88.1 | 88.4 | 88.5 | 87.8 |
| VOC | 512 | T1 Acc | 88.2 | 88.3 | 89.2 | 88.8 | 88.0 | 87.5 | 89.7 | 89.4 | 89.7 | 88.9 |
| DTD | 256 | T1 Acc | 87.5 | 86.2 | 87.1 | 84.3 | 82.8 | 81.4 | 81.7 | 81.3 | 80.2 | 79.7 |
| DTD | 512 | T1 Acc | 87.2 | 86.5 | 86.6 | 85.8 | 83.3 | 82.6 | 83.4 | 83.2 | 82.6 | 83.4 |
| Pets | 256 | T1 Acc | 97.1 | 96.8 | 97.1 | 96.2 | 95.5 | 94.9 | 96.3 | 95.8 | 95.4 | 95.3 |
| Pets | 512 | T1 Acc | 97.0 | 96.6 | 97.1 | 96.3 | 96.0 | 95.1 | 96.4 | 96.3 | 96.1 | 95.7 |
| Caltech101 | 256 | T1 Acc | 97.0 | 96.7 | 96.5 | 96.0 | 95.6 | 94.0 | 95.4 | 93.9 | 94.5 | 95.2 |
| Caltech101 | 512 | T1 Acc | 94.8 | 93.5 | 94.0 | 93.1 | 93.9 | 93.1 | 96.6 | 95.9 | 96.3 | 96.6 |
| Flowers | 256 | T1 Acc | 99.7 | 99.7 | 99.7 | 99.6 | 99.6 | 99.6 | 99.6 | 99.7 | 99.6 | 99.7 |
| Flowers | 512 | T1 Acc | 99.7 | 99.7 | 99.7 | 99.6 | 99.6 | 99.6 | 99.7 | 99.7 | 99.7 | 99.7 |
| CUB | 256 | T1 Acc | 92.0 | 91.1 | 91.3 | 90.0 | 88.4 | 86.6 | 86.5 | 84.5 | 86.1 | 85.2 |
| CUB | 512 | T1 Acc | 92.4 | 91.5 | 91.6 | 90.6 | 90.0 | 88.9 | 83.1 | 83.8 | 85.2 | 86.4 |
| *Instance Recognition (Sec. 6.2.2)* | | | | | | | | | | | | |
| Oxford M | 256 | mAP | 83.2 | 83.0 | 82.6 | 80.5 | 75.8 | 74.4 | 71.3 | 68.7 | 73.9 | 71.2 |
| Oxford H | 256 | mAP | 64.5 | 64.7 | 63.7 | 59.6 | 52.5 | 49.6 | 47.7 | 43.2 | 51.6 | 46.6 |
| Paris M | 256 | mAP | 92.0 | 93.1 | 91.7 | 91.3 | 88.1 | 85.7 | 91.8 | 90.2 | 89.9 | 90.5 |
| Paris H | 256 | mAP | 85.4 | 87.1 | 83.1 | 80.9 | 74.9 | 70.5 | 83.4 | 80.7 | 79.4 | 78.6 |
| Met | Orig | GAP | 51.1 | 32.9 | 41.7 | 39.1 | 11.5 | 30.3 | 10.7 | 10.7 | 14.8 | 12.0 |
| Met | Orig | GAP- | 77.5 | 65.3 | 67.7 | 65.1 | 31.0 | 52.9 | 41.3 | 40.1 | 43.7 | 40.2 |
| Met | Orig | Acc | 81.9 | 72.0 | 73.0 | 69.2 | 38.4 | 56.4 | 50.9 | 49.3 | 50.7 | 47.5 |
| AmsterTime | 256 | mAP | 59.4 | 57.0 | 56.5 | 54.5 | 49.0 | 50.5 | 38.1 | 40.2 | 40.6 | 42.1 |

Table 37: Overview of benchmark results for dense tasks for the DINOv3 family of models.

| Benchmark | Res. | Metric | ViT-7B | ViT-H+ | ViT-L | ViT-B | ViT-S+ | ViT-S | CNX-L | CNX-B | CNX-S | CNX-T |
|---|---|---|---|---|---|---|---|---|---|---|---|---|
| *Linear Semantic Segmentation (Sec. 6.1.2)* | | | | | | | | | | | | |
| ADE20k | 512 | mIoU | 55.9 | 54.8 | 54.9 | 51.8 | 48.8 | 47.0 | 47.8 | 46.3 | 44.8 | 42.7 |
| ADE20k | 1024 | mIoU | 57.5 | 56.2 | 56.5 | 52.7 | 49.1 | 46.4 | 49.6 | 47.0 | 45.8 | 41.5 |
| Cityscapes | 1024 | mIoU | 81.1 | 80.1 | 79.4 | 76.6 | 73.8 | 71.8 | 72.8 | 71.2 | 71.0 | 68.8 |
| VOC | 512 | mIoU | 86.6 | 85.9 | 86.3 | 86.0 | 84.6 | 83.3 | 81.7 | 81.2 | 80.8 | 79.3 |
| *Linear Depth Estimation (Sec. 6.1.2)* | | | | | | | | | | | | |
| NYUv2 | Orig | RMSE ↓ | 0.309 | 0.354 | 0.352 | 0.373 | 0.399 | 0.403 | 0.403 | 0.419 | 0.432 | 0.448 |
| KITTI | Orig | RMSE ↓ | 2.346 | 2.471 | 2.504 | 2.680 | 2.751 | 2.750 | 2.859 | 2.960 | 2.917 | 2.949 |
| *3D Correspondence Estimation (Sec. 6.1.3)* | | | | | | | | | | | | |
| NAVI | 512 | Recall | 64.6 | 63.3 | 62.3 | 58.8 | 57.1 | 56.3 | 57.9 | 57.1 | 56.6 | 55.8 |
| NAVI | 1024 | Recall | 66.8 | 64.6 | 62.0 | 55.5 | 54.6 | 53.4 | 60.6 | 58.1 | 55.9 | 54.7 |
| SPair | 512 | Recall | 49.6 | 48.2 | 53.4 | 50.5 | 49.6 | 46.6 | 29.1 | 28.8 | 29.0 | 29.0 |
| SPair | 1024 | Recall | 58.7 | 56.3 | 61.2 | 57.2 | 55.2 | 50.4 | 45.2 | 43.6 | 43.4 | 42.7 |
| *Video Segmentation Tracking (Sec. 6.1.5)* | | | | | | | | | | | | |
| DAVIS | 480 | J&F | 71.1 | 72.0 | 72.6 | 71.3 | 69.6 | 68.5 | 44.2 | 43.9 | 44.0 | 42.6 |
| DAVIS | 960 | J&F | 79.7 | 79.3 | 79.9 | 77.2 | 75.5 | 72.7 | 65.3 | 65.2 | 65.9 | 65.1 |
| DAVIS | 1440 | J&F | 83.3 | 82.9 | 83.3 | 75.0 | 73.0 | 66.5 | 73.2 | 72.7 | 72.0 | 70.7 |
| YouTube-VOS | 480 | J&F | 74.1 | 75.0 | 75.2 | 74.6 | 73.6 | 73.3 | 49.2 | 49.1 | 49.3 | 49.6 |
| YouTube-VOS | 960 | J&F | 80.2 | 80.1 | 80.3 | 77.9 | 76.8 | 75.2 | 68.0 | 66.5 | 67.5 | 67.9 |
| YouTube-VOS | 1440 | J&F | 80.7 | 80.2 | 80.3 | 73.9 | 73.4 | 69.4 | 72.7 | 71.0 | 71.3 | 71.3 |
| MOSE | 480 | J&F | 46.0 | 45.6 | 45.7 | 45.2 | 44.0 | 43.7 | 28.0 | 27.1 | 27.2 | 27.5 |
| MOSE | 960 | J&F | 53.9 | 51.5 | 51.7 | 49.5 | 47.2 | 45.5 | 38.1 | 37.4 | 37.7 | 38.0 |
| MOSE | 1440 | J&F | 55.6 | 51.3 | 53.0 | 45.8 | 45.0 | 41.6 | 42.0 | 41.5 | 41.5 | 41.6 |
| *Video Classification (Sec. 6.1.6)* | | | | | | | | | | | | |
| UCF101 | 256 | T1 | 93.5 | 93.3 | 93.3 | 91.7 | 88.5 | 86.7 | 91.3 | 90.4 | 90.1 | 88.8 |
| UCF101+TTA | 256 | T1 | 93.5 | 93.7 | 93.3 | 91.5 | 89.0 | 88.1 | 91.9 | 90.2 | 90.1 | 89.1 |
| SSv2 | 256 | T1 | 70.1 | 69.8 | 68.9 | 65.2 | 63.3 | 63.2 | 61.8 | 62.6 | 61.9 | 60.7 |
| SSv2+TTA | 256 | T1 | 70.8 | 70.7 | 69.8 | 67.4 | 66.3 | 64.7 | 64.4 | 63.9 | 63.2 | 63.3 |
| K400 | 256 | T1 | 87.8 | 86.7 | 84.9 | 82.2 | 79.6 | 78.5 | 82.1 | 80.7 | 79.2 | 76.6 |
| K400+TTA | 256 | T1 | 88.2 | 87.2 | 86.0 | 83.4 | 80.8 | 80.0 | 83.4 | 81.6 | 80.3 | 78.3 |

Table 38: Memory use (MB) for inference in float32 precision, square images at various resolutions. For batch inference, model memory remains constant, while the per-sample value should be multiplied by the batch size.

| | ViT-7B | ViT-H+ | ViT-L | ViT-B | ViT-S+ | ViT-S | CNX-L | CNX-B | CNX-S | CNX-T |
|---|---|---|---|---|---|---|---|---|---|---|
| Model memory | 25,620 | 3,207 | 1,156 | 327 | 109 | 82 | 749 | 334 | 189 | 106 |
| Per sample @ 256 | 49 | 30 | 13 | 9 | 8 | 5 | 30 | 20 | 16 | 15 |
| Per sample @ 384 | 110 | 60 | 28 | 22 | 17 | 10 | 69 | 45 | 34 | 34 |
| Per sample @ 512 | 194 | 104 | 49 | 38 | 31 | 19 | 120 | 80 | 61 | 60 |
| Per sample @ 768 | 437 | 227 | 109 | 84 | 69 | 42 | 271 | 180 | 135 | 135 |
| Per sample @ 1024 | 773 | 402 | 194 | 146 | 122 | 74 | 480 | 320 | 240 | 240 |