# OpenReview forum: "DINOv3"
_TMLR — Accepted by TMLR_

### Review · Reviewer_k7MQ · 2026-02-24

**Summary Of Contributions:**

The paper builds on the popular DINO line of self-supervised image models, introducing DINOv3 which makes a number of improvements upon DINOv2. These include 1) a new data curation and sampling strategy, 2) changes to the architecture and optimizer, and 3) changes to the objective, most significantly the includion of the Gram anchoring objective that encourages patch-level consistency. The model is then evaluated in a number of ways including on the quality of dense and global features, and as a pretrained model for finetuning on complex downstream tasks. Some particularly interesting results include the strong performance on global classification comparable with supervised models, and applicability to satellite data.

**Audience:**

Yes

**Audience Explanation:**

Given the popularity of the DINO line of models, and the strong results that DINOv3 achieves, this paper would clearly be of interest to a wide audience.

**Broader Impact Concerns:**

There are two primary ethical concerns with this work. The first is privacy: the authors mention that the training data is taken from public posts on Instagram. This raises questions of transparency: to what extent do users know that their public pictures are being used to train a large vision model (users may not be aware of the platform's legal fine print)? This question warrants some discussion in the paper.

Second, the carbon footprint analysis could benefit from two additions. The first is the actual carbon intensity factor of the power used to train the model; in other words, is the power taken from the grid (in which case the assumed 0.385 kg/KWh may be reasonable), or is the power taken from a private power source with a different intensity factor (e.g. a carbon-neutral source). Also, if training DINOv3 was not carbon-neutral, then were carbon offsets purchased (and which organization were these offsets purchased from)? These questions help established whether this research was conducted in an ethical way.

I do commend the authors for including a fairness analysis in Appendix C.5.

**Claims And Evidence:**

Yes

**Claims Explanation:**

The design choices made in this paper are well-explained with ablations where appropriate (e.g. Table 1 and Figure 9). The experimental analysis is comprehensive with extensive evaluations on several datasets. To train models of the size and complexity of DINOv3, many design choices have to be made. Thus, my main technical concern is whether all these design choices have been properly evaluated with ablations. Please see requested changes section for a list of these.

**Requested Changes:**

**Would strengthen**
It would be great to have ablations for the following (I couldn't find them, but it's possible I missed them):
- Data sampling: homogenous batches vs heterogenous batches vs both (what is actually done)
- Main learning objective: remove iBOT loss, remove Koleo loss
- RoPE box jittering vs standard RoPE
- Optimizer choices: learning rate, weight decay, teacher EMA momentum

I understand it is not be feasible to train a version of DINOv3 with each of these changes, but for a paper of this caliber, these design choices need to be quantitatively justified.

---

> ### Author Response · Authors · 2026-04-16
> **Response**
>
> ### 1. Data sampling
>
> We thank the reviewer for the suggestion. We provide an explicit ablation, in Tab. 24 of the revised appendix (Sec. C.2), where we train for 200k iterations with fully heterogeneous batches, i.e., each batch contains samples across all datasets, and fully homogeneous batches, i.e, each batch contains samples from only one of the data parts (ImageNet1k, ImageNet22k, Mapillary, retrieval-based curated part, clustering-based curated part). We make sure that these mixtures have the same data proportions used in our main setup. We observe that overall best performance is obtained with our mixed setup.
>
> ### 2. Main learning objective (iBOT / KoLeo ablations)
>
> We thank the reviewer for the suggestion. Our goal in this work is to scale the DINOv2 training recipe, rather than to re-ablate components that were already carefully studied in prior work. In particular, both KoLeo and the iBOT loss have been ablated in the original papers: we refer the reviewer to Table 1 in the DINOv2 paper for the impact of KoLeo, and to Table 9 in the iBOT paper for the ablation of the iBOT loss. Given the substantial computational cost of our scaled setting, we did not repeat these ablations here, and instead built on these established results.
>
> ### 3. RoPE box jittering vs standard RoPE
>
> We thank the reviewer for the suggestion and provide an ablation in Tab.25, in Sec. C.3 of the revised appendix, when using RoPE with jittering (official) and the classic version, when training the model for 200k iterations. The largest performance differences appear on dense tasks, with jittering positively impacting results, while global tasks show a small difference.
>
> ### 4. Optimizer choices
>
> We thank the reviewer for the suggestion. We did not run a broad grid search over learning rate, weight decay, or teacher EMA momentum due to resource constraints at our training scale. That said, we performed a small learning-rate sweep, and reported results after 500k pre-training iterations in Tab.26 of the revised appendix; we have added a small hyper-parameter discussion in Sec.C.4. Overall, performance is fairly stable across the tested learning rates, with 0.0004 providing the best aggregate trade-off.
>
> We also tried to schedule up the weight decay by factors of 4–5x during training but that led to degraded results. Another sensitive hyperparameter is stochastic depth: following DINOv2 we use a value of 0.4; in one experiment, we decreased the value to 0.35, leading to improved linear classification results but worse results on dense and non-parametric tasks (like k-nn classification).
>
> ### 5. Privacy concern
>
> Instagram data used in the pretraining data is based on public posts of 18+ users. We guarantee the reviewers that the data was acquired rightfully and refer them to both Meta’s Privacy Policy and the AI section of the privacy center on Facebook services. These pages document relevant terms that Instagram users agree with and consent when using the services while being free to adjust their preferences around data collection and use.
>
> We have added a discussion in Sec. 11 Broader Impact.
>
> ### 6. Carbon footprint
>
> As mentioned in the paper, the reported numbers are provided for illustrative purposes only: this is a theoretical estimate intended to give an order-of-magnitude indication, rather than a precise accounting. As a simple rule of thumb, we used the average U.S. grid carbon-intensity factor to convert energy into emissions.

---

### Review · Reviewer_fbzL · 2026-03-20

**Summary Of Contributions:**

**Summary**

This paper presents DINOv3, a large-scale self-supervised visual foundation model. The work combines several components: large-scale data curation and training, a refinement stage based on Gram anchoring to address the degradation of dense feature maps during long training, and post-hoc adaptation steps for resolution scaling, distillation, and text alignment. The paper argues that long training improves global performance but harms dense representations, and proposes Gram anchoring as the key remedy for this issue. Empirically, the paper reports very strong performance, especially on dense prediction benchmarks.

**Strengths**

1. The paper tackles an important and practically relevant problem in large-scale SSL. The observation that dense feature quality degrades during long training, despite continued gains on global metrics, is interesting and well motivated. The paper clearly documents this phenomenon through both quantitative curves and qualitative patch-similarity visualizations, which makes the motivation for a dense-feature convincing

2. The proposed Gram anchoring refinement is simple and empirically effective. The paper makes a clear distinction between preserving patch-level similarity structure and directly constraining features themselves, and the reported results show that introducing the refinement step yields gains on dense tasks while only mildly affecting global performance. The high-resolution variant further strengthens the method.

3. The experimental scope is broad and the dense-task results are particularly strong.

**Weaknesses**

1. One area that could benefit from a bit more clarification is the choice of the Gram teacher. The paper shows that earlier checkpoints work better than later ones, which is helpful, but the selection criterion still feels somewhat empirical. A brief discussion of what makes a checkpoint have better “dense properties” would make this part easier to follow.

2. The motivation for the Gram objective could be elaborated further, especially regarding why this formulation is particularly suitable.

**Audience:**

Yes

**Audience Explanation:**

This work addresses an important and timely problem in large-scale self-supervised learning. The findings on dense feature degradation and the proposed refinement approach are likely to be of broad interest to the vision and representation learning community, especially in the context of foundation models.

**Claims And Evidence:**

Yes

**Claims Explanation:**

The core empirical claims, especially regarding improved dense representations and the effectiveness of the refinement stage, are generally well supported by the experiments and ablations. The comparison across teacher checkpoints also supports the claim about dense feature degradation during long training.

**Requested Changes:**

I would encourage the authors to further strengthen the analysis of Gram anchoring. In particular, it would be helpful to provide a clearer discussion on how the Gram teacher is selected, and whether there exists a more systematic or measurable criterion for identifying checkpoints with better dense properties.

In addition, I suggest expanding the discussion on the motivation behind the Gram objective. While the current explanation is reasonable, a bit more elaboration on why this formulation is particularly suitable, and how it compares conceptually to simpler alternatives, would further improve the clarity of the method.

---

> ### Author Response · Authors · 2026-04-16
> **Response**
>
> ### 1. Gram section
>
> We thank the reviewer for this suggestion. To further strengthen this section, we have extended our explanation in section 4.2, produced a new figure (new Fig. 7) and a new table (new Tab. 3) which explain the impact of the gram anchoring both qualitatively and quantitatively.
>
> The anchor model was selected based on qualitative inspection and a small number of segmentation results. We agree that identifying a more principled selection criterion is an interesting direction for future work.
>
> That said, our ablation study in Fig. 9(b) (now Fig. 10(b) in the revised manuscript) demonstrates that performance is largely robust to the choice of Gram anchor. On ImageNet-1K, linear evaluation accuracy remains essentially unchanged across all anchor choices. On ADE20K, the least favorable anchor (1M-iteration teacher) incurs only a modest drop of 0.8 points, while still yielding a +4.6-point improvement over the baseline. This confirms that the benefits of our method are not contingent on a particular anchor selection, preserving our main conclusions.
>
> We hypothesize that this reduced sensitivity stems from two design choices: (i) the ×2 resolution setting, which smooths noisy patch-level features, and (ii) updating the anchor during training, which reduces reliance on any single teacher checkpoint.
>
>
> ### 2. Simpler alternatives
>
> We are curious of what simpler alternatives the reviewer is thinking of, and would be happy to expand on it given further information.

---

### Review · Reviewer_8fNt · 2026-03-22

**Summary Of Contributions:**

This paper presents DINOv3, a large-scale self-supervised vision foundation model built on a teacher-student ViT training pipeline, followed by a refinement stage based on the proposed Gram anchoring objective, a high-resolution adaptation stage, and distillation into a family of smaller models. The central technical claim is that long-horizon SSL training leads to a degradation of dense features even as global performance continues to improve, and that preserving pairwise patch-similarity structure through Gram anchoring mitigates this effect. The paper then supports this claim with a very broad empirical evaluation across dense prediction, global recognition, retrieval, video, 3D, text alignment, and geospatial transfer.

**Additional Comments:**

1. The paper clearly demonstrates that dense features degrade during long training and that Gram anchoring helps, but the deeper reason why global objectives come to dominate local structure is not fully unpacked. The phenomenon looks related to a broader representation homogenization / over-smoothing effect in deep attention-based models, but the paper stops short of giving a more principled analysis along those lines. This does not weaken the empirical contribution, but a stronger explanation would improve the scientific value.

2. The final system combines many ingredients: very large scale, careful private-data curation, constant-hyperparameter long training, Gram refinement, high-resolution adaptation, and distillation. The paper includes useful ablations, but it is still somewhat hard to separate how much of the final gains come specifically from Gram anchoring versus scaling, curation, and later post-training stages. This is especially relevant because the paper’s broad downstream gains are one of its headline messages.

3. Reproducibility is an understandable but real limitation. The training setup relies on a private raw image pool of roughly 17B Instagram images, extensive compute, and a large 7B teacher model. The paper is transparent about these ingredients and even reports carbon estimates, which I appreciate, but it still means that some of the headline results are difficult for the broader community to reproduce directly.

**Audience:**

Yes

**Audience Explanation:**

The paper provides broadly useful insights into large-scale self-supervised vision learning and dense representation quality that would interest researchers working on foundation models and representation learning.

**Claims And Evidence:**

No

**Claims Explanation:**

This is a very long submission, with about 17 pages for experiments. Total GPU hours is about 60K.

**Requested Changes:**

First of all,
1. This paper does not contain much insights or noveltis in algorithm designs (which is not a main concern for TMLR).
2. I am not an expert for vision foundation model.

The paper is almost clear. My main suggestions is that keep main text short into 20 pages.

---

> ### Author Response · Authors · 2026-04-17
> **Response**
>
> We thank the reviewer for the feedback. While we would have liked to be able to reduce the paper to 20 pages, we believe that the current length is necessary: each section and experiment is included to substantiate the claims and to give readers a clear, complete picture of the model’s capabilities.
>
>
> Regarding the additional comments:
>
> ### 1. Deeper reasons for local feature degradation
>
> * We observed in Sec. 4.1 that extended training causes the student model to align its patch tokens too closely with the global CLS token, leading to a loss of spatial locality.
> * As noted in the Introduction, this phenomenon is most pronounced in models exceeding 300M parameters, where the global objective eventually dominates and collapses the local feature structure.
> * Sec. 4.2 explains how Gram anchoring resolves this by enforcing the similarity structure of an earlier teacher, allowing features to refine semantically while remaining anchored to their local geometric properties.
>
> Unfortunately we do not have stronger insights to share than what we describe in the paper, but we agree with the reviewer that this would provide a lot of scientific value; we have been investigating this problem but haven’t yet been able to add further conclusions decisively, due to the difficulty of the problem. The paper reflects the current form of our knowledge.
>
> ### 2. Separating the impact of system ingredients
>
> * Tab. 1 in Sec. 3.1 confirms that while scaling is necessary, no single curation technique is sufficient; only our specific data mixture achieves state-of-the-art results.
> * We highlighted in Sec. 3.2 that scaling to 7B without Gram anchoring—as seen in concurrent endeavors like Web-DINO—results in disappointing dense prediction performance.
> * Finally, Sec. 5.1 demonstrates that the high-resolution adaptation phase is a critical component for capturing the intricate spatial information required for tracking and segmentation tasks.
>
> While we agree that further ablations between all components would definitely be interesting, unfortunately we are limited by the computational resources at the scale DINOv3 operates at. Thus we picked a set of ablations that provide insight into the most critical components. That said, we added a few more ablations to the appendix of the revised manuscript (data mix, rope jittering, HP study, all in App. C) that we hope can provide further insight.
>
> ### 3. Reproducibility and resource transparency
>
> * To address reproduction challenges, we are releasing the full DINOv3 model suite, allowing researchers to leverage our flagship 7B features on diverse compute budgets.
> * For reproducibility, we describe the dataset curation in Sec. 3.1 of the main paper and Sec. D.2 of the supplementary material. The clustering-based component follows Vo et al. (2024), and we provide parameters details in Sec. 3.1. Moreover, Tab. 28 (new Tab. 32 in the revised manuscript) lists the seed datasets used in the retrieval-based stage; these sources are public. Overall, we provide explicit details on data processing and evaluation to enable third parties to replicate our curation on their own large-scale data pools.
> * Furthermore, Sec. 5.2 details our novel multi-student distillation procedure, which proves that the frontier-level power of a 7B teacher can be efficiently transferred into smaller, accessible architectures.
>
> We acknowledge that this setup remains difficult to reproduce and hope that the model weights and code release are alleviating the underlying concern.

---

### Review · Reviewer_fMJT · 2026-03-24

**Summary Of Contributions:**

This paper presents DINOv3, the latest in the DINO family of image models. DINOv3 differs from DINOv2 technically via a new semi-supervised learning approach (a pipeline largely created by taking lessons from existing research) on a massive dataset of instagram images, a new loss, and a new refinement stage. The authors also contribute an incredible number of evaluations of their model, showing that it meets or exceeds the performance of specialized models on many image processing tasks. DINOv3 clearly stands out as a new image foundation model.

### Strengths
1. Very well written paper
2. Strong overview of existing literature
3. Pipeline informed by existing literature
4. Very large number of experiments across different image processing domains
5. Very large number of ablations and other experiments to dig into model performance and impact

### Weaknesses
1. Some training pipeline design decisions not justified
2. Discussion of relative performance across results often minimal
3. No ethical discussion
4. Lack of clarity around selection of images in the paper
5. Unclear general value of Gram anchoring

**Audience:**

Yes

**Audience Explanation:**

Definitely yes. Anyone working with images and machine learning would want to at least scan this paper.

**Broader Impact Concerns:**

The only broader impact concern is with the lack of ethical acknowledgement, discussed above.

**Claims And Evidence:**

Yes

**Claims Explanation:**

The authors provide a wide swathe of experiments over existing benchmarks with existing baselines. These seem very convincing in terms of the overall performance.

**Requested Changes:**

My requested changes largely come down to the above identified weaknesses.

1. Many aspects of the training pipeline are justified with prior work. However, some are not. I would really like to at least see a short justification anytime we're seeing a specific number or hyperparameter, even if it's just in the appendix. I'm not asking for an ablation for everything, that would be ridiculous given the scale of the training process. But at least some prior work citation or experience-based argument would help strengthen this paper and inform prior work. If future researchers don't know if a specific hyperparamter was crucial or arbitrary they may waste a lot of time.
2. Discussion of relative performance of the model often takes the form of very high-level statements, like "V-JEPA v2 shines on this dataset". These statements lack depth or insight. Replacing them with specific reasons why the baselines perform well or do not would again strengthen the paper and help inform future research.
3. One of the major problems with the current paper is that it mentions using essentially all images from instagram without any deeper discussion than noting that these images have already undergone a moderation pass. If the folks developing DinoV3 are the same as those developing DinoV2, I don't doubt the legal right to use these images. But it would be good to acknowledge, particularly in an age of increasing anti-AI sentiment, particularly from the demographics active on instagram, that this decision is unlikely to be universally approved. Some discussion justifying it would benefit the paper and authors.
4. The paper makes use of a number of excellent illustrations, to its benefit! However, the images all appear to be cherry-picked. This is understandable as a decision, but it could serve to give reader an imprecise understanding of the model's behaviour. Some acknowledgement of this if I'm right, and alternatively a discussion of how the images were picked if I was wrong, would benefit the paper.
5. Gram anchoring is perhaps the most clear novel technical contribution in the paper. However, besides a teacher ablation experiment, there's no clear notion of when it is or isn't helpful. I'm not suggesting retraining DINOv3's backbone without it, the economics don't make sense. However, it would be useful to consider training a smaller model with and without it to demonstrate its utility in a more general case outside of SSL at this scale.

---

> ### Author Response · Authors · 2026-04-16
> **Response**
>
> ### 1. Hyperparameters
>
> We thank the reviewer for the suggestion. We did not run a broad grid search over learning rate, weight decay, or teacher EMA momentum due to resource constraints at our training scale. That said, we performed a small learning-rate sweep, and report now in the new Tab.26 (Sec. C.4 of the revised appendix) results after 500k pre-training iterations. Overall, performance is fairly stable across the tested learning rates, with 0.0004 providing the best aggregate trade-off.
>
> We also tried to schedule up the weight decay by factors of 4–5x during training but that led to degraded results. Another sensitive hyperparameter is stochastic depth: following DINOv2 we use a value of 0.4; in one experiment, we decreased the value to 0.35, leading to improved linear classification results but worsened results on dense and non-parametric tasks (like k-nn classification).
>
> ### 2. Relative performance
>
> We thank the reviewer for the suggestion and agree that “V-JEPA v2 shines on this dataset” is not our most interesting formulation. We improved the text and now explain that SSV2 emphasizes fine-grained temporal dynamics, and V-JEPA v2 is a video-specific model pretrained to capture motion/temporal structure, which better matches the dataset and explains its stronger results than more image-centric baselines.
>
> ### 3. Data
>
> We indeed confirm the public Instagram images leveraged for this paper and model are rightfully used for both training and evaluation purposes. We add a comment about it in Sec. 11 of the revised manuscript.
>
> Our motivation for using images from public Instagram posts is that they constitute a large-scale pool of 17B images, which is diverse, real-world imagery that would be difficult to obtain otherwise. This diversity carries a major weight for learning robust visual representations that generalize across domains and downstream applications. We will emphasize this in the paper.
>
> ### 4. Visuals
>
> We did spend quite some time to produce good quality visuals as we aimed to illustrate the abilities of our models. The images were not selected based on whether the model performed best, but chosen because their characteristics help illustrate specific behaviors we want to highlight (e.g., high-resolution, cluttered scenes for the marketplace example). All examples come from the same pool of personal holiday photos, and we add a brief note acknowledging that these figures are illustrative rather than a systematic evaluation in the appendix.
>
> ### 5. Gram anchoring
>
> We are not entirely sure we understand the reviewer’s request. In our setup, the model is trained without Gram anchoring for over 1M iterations, and Gram anchoring is then applied for only 70k additional steps, starting from the 1M-iteration checkpoint. The goal of this final Gram-anchoring phase is to improve the quality of the patch-level features, which we observe to degrade substantially by 1M iterations (as discussed in the paper).
>
> To make the impact of Gram anchoring clearer, we have extended our explanation in section 4.2, produced a new figure (new Fig. 7) and a new table (new Tab. 3) which explain the impact of the gram anchoring both qualitatively and quantitatively.

---

### Review · Reviewer_XT5p · 2026-03-24

**Summary Of Contributions:**

This paper introduces DINOv3, a large-scale self-supervised vision foundation model that builds upon the DINOv2 framework. The authors scale both model size and training data while introducing several key innovations: (1) a carefully curated data mixture combining clustering-based and retrieval-based curation, (2) architectural improvements including axial RoPE and register tokens, (3) a novel Gram anchoring objective to mitigate dense feature degradation during long training, (4) a high-resolution adaptation phase, and (5) an efficient multi-student distillation pipeline. The resulting model family achieves state-of-the-art performance across a wide range of dense and global vision tasks.
Strengths
1. The paper presents extensive evaluations across a broad spectrum of tasks and shows consistent improvements over strong baselines. The results on dense tasks are particularly impressive.
2. The Gram anchoring objective is a novel and well-motivated solution to the previously underexplored issue of dense feature degradation during long training. The use of an earlier EMA teacher as a "Gram teacher" and the incorporation of high-resolution downsampling are clever and effective.
3. The paper demonstrates that SSL can scale to 7B parameters and still produce high-quality features.
4. Clear writing and thorough ablation. The paper is well-structured, with clear motivations, detailed ablations, and visualizations that support the claims.
Weaknesses
1. While the authors include a carbon footprint estimate, the practical inference cost of the 7B model (especially at high resolutions) is not discussed in detail. This is important for practitioners considering deployment.
2. Several components are adapted from prior work. While their combination is effective, the novelty is primarily in the Gram anchoring and scaling strategy.
3. The LVD-1689M dataset is built from Instagram images and is not publicly available, limiting reproducibility. The authors acknowledge this but do not provide a clear path for others to replicate the data curation.

**Additional Comments:**

I’m very curious why DINOv3 is reported to underperform DINOv2 on many segmentation tasks, even though the authors put significant work into improving its segmentation capabilities. This aligns perfectly with my own experimental experience, and I’d be grateful if the authors could explain this phenomenon here.

**Audience:**

Yes

**Audience Explanation:**

The paper addresses a core problem in self-supervised learning—scaling while preserving dense feature quality—and presents a practical solution with broad applicability. Researchers in computer vision, foundation models, SSL, and domain-specific applications (e.g., remote sensing, video understanding, 3D vision) will find this work highly relevant.

**Broader Impact Concerns:**

The paper includes a brief environmental impact statement but does not discuss ethical implications such as:
1. Use of Instagram data without explicit user consent for model training.
2. Potential misuse of high-quality dense features for surveillance or facial recognition.
3. Bias and fairness in web-scraped datasets.

**Claims And Evidence:**

Yes

**Claims Explanation:**

The paper provides extensive and well-designed experiments across a wide range of tasks, with clear comparisons to strong baselines. The ablations are thorough and the visualizations convincingly illustrate the improvements. The claims about Gram anchoring fixing dense feature collapse are supported by both quantitative metrics and qualitative analysis.

**Requested Changes:**

Critical for acceptance:
1. Provide more details on the data curation pipeline. If possible, release a sample of the curated dataset or a synthetic proxy to aid reproducibility.
2. Include a table or discussion of inference FLOPs and memory usage for the 7B model and distilled variants at different resolutions. This is essential for practitioners.
3. The choice of early iterations (100k–200k) as Gram teachers is critical. Provide more analysis on how this choice affects performance and whether it generalizes across model sizes.
Would strengthen the work:
1. Add some failure cases. While the results are strong, it would be helpful to include examples where DINOv3 struggles.
2. Expand the broader impact section to discuss potential misuse (e.g., surveillance, bias in web-scraped data) and mitigation strategies.

---

> ### Author Response · Authors · 2026-04-16
> **Response**
>
> ### 1. Data
> We thank the reviewer for this feedback and suggestions.
>
> For reproducibility, we describe the dataset curation in Section 3.1 of the main paper and Section D.2 of the supplementary material. The clustering-based component follows Vo et al. (2024), and we provide parameters details in Sec. 3.1. Moreover, Tab. 28 (new Tab. 32 in the revised manuscript) lists the seed datasets used in the retrieval-based stage; these sources are public. Overall, we provide explicit details on data processing and evaluation to enable third parties to replicate our curation on their own large-scale data pools.
>
> Regarding sharing data samples, we do not have the rights to distribute the data. The alternative of preparing a synthetic proxy for this data is an interesting direction, it however remains unclear how to produce data that would be faithful enough and effective for reproducibility.
>
>
>
> ### 2. FLOPs
> We acknowledge that the 7B model may limit the practical use of DINOv3. This is precisely why we distill it into a family of smaller models, as described in Section 5.2. Moreover, we refer the reviewer to the table in Fig. 15(a) in the main paper (new Fig.16 (a) in the revised manuscript), which reports the GFLOPs for each model in the DINOv3 family, along with the corresponding parameter counts. We also provide a table (Tab. 38, Appendix G) for the memory usage of models during inference at various resolutions in the revised manuscript.
>
>
>
> ### 3. Gram teacher choice
>
> We thank the reviewer for the question. While the choice of Gram teacher matters, our method is not critically sensitive to it. We include an ablation in Fig. 9(b) (new Fig. 10(b) in the revised manuscript) where we vary the Gram teacher checkpoint (100k, 200k, and 1M iterations). IN-1K linear evaluation remains essentially unchanged across these choices. On ADE20K, using the 1M-iteration teacher yields a modest drop of 0.8 points, but performance still improves by +4.6 points over the baseline, preserving the main conclusion.
>
> We believe this reduced sensitivity stems from (i) using the ×2 resolution setting (which helps in smoothing out noisy patch features) and (ii) updating the anchor during training, which together mitigate dependence on any single teacher checkpoint.
>
>
> ### 4. Broader Impact Concerns: Consent
>
> Instagram data used in the pretraining data is based on public posts of users aged 18+. We guarantee the reviewers that the data was acquired rightfully and refer them to both Meta’s Privacy Policy and the AI section of the privacy center on Facebook services. These pages document relevant terms that Instagram users agree with and consent when using the services while being free to adjust their preferences around data collection and use.
>
> We also have added a discussion on this point in Sec. 11 Broader Impact.
>
>
> ### 5. Broader Impact Concerns: Potential misuse
> We acknowledge the reviewer’s concern, which applies to any generic computer vision model. While our goal is to support broad research and applications, we do acknowledge the usual risks of computer vision models.
>
> We have added a discussion in Sec. 11 Broader Impact.
>
> ### 6. Broader Impact Concerns: Bias and fairness in web-scraped datasets
>
> We appreciate the reviewer’s point that datasets made mostly of social media posts may inherit from societal and geographic biases.
> To address such concerns, we chose to include a dedicated fairness analysis in the supplementary material (Sec. C.5 Fairness Analysis, now C.8 in the revised manuscript), where we evaluate the geographical diversity and fairness of DINOv3-7B across regions and income buckets. We now reference Sec. C.5 in the main paper (in Sec.10) to ensure readers can easily locate the information.
>
> Moreover, to mitigate unbalances in the dataset, we employ a hierarchical clustering–based data selection strategy, which is designed to promote diversity and coverage rather than over-sampling the most frequent web patterns.
>
> ### 7. Segmentation
>
> We are curious about the results mentioned by the reviewer and would appreciate references so that we can provide an informed response.

---

### Author Response · Authors · 2026-04-16
**General response from Authors**

We thank all reviewers for their thorough and constructive feedback. We are encouraged that reviewers unanimously appreciated the clarity of the paper, the novelty and effectiveness of the Gram anchoring objective for addressing dense feature degradation during long SSL training, the breadth and rigor of our experimental evaluation—particularly the strong dense prediction results—and the demonstration that self-supervised learning can scale to 7B parameters while producing high-quality features competitive with specialized and supervised models.

In response to the shared concerns, we have made several additions to the revised manuscript, highlighted in orange for better visibility. Specifically, we have strengthened the Gram anchoring section (Sec. 4) with improved exposition and new qualitative and quantitative analyses (new Fig. 7 and Tab. 3). We have also added new ablations in the appendix (new Tab. 24, 25, 26) covering data sampling strategies, RoPE jittering, and learning rate sensitivity. On compute and practicality, we have added a new inference memory and FLOPs table (new Tab. 38 in app.). Finally, we have created the Broader Impact section to discuss data privacy and ethics, fairness and bias mitigation, as well as the societal and scientific benefits of our work.

We address each reviewer's specific questions individually below.

---

### Decision · Action_Editor_UYpa · 2026-04-25

**Recommendation:** Accept as is

**Additional Comments:**

N/A

**Audience:**

Yes

**Audience Explanation:**

- DINOv3 provides strong results for large-scale self-supervised vision backbones, it would clearly be of interest to a wide audience.

- DINOv3 has been verified by the community.

**Claims And Evidence:**

Yes

**Claims Explanation:**

- This work offers a broad range of carefully designed experiments across diverse tasks, each with clear comparisons to competitive baselines. The ablation studies are comprehensive, and the visualizations effectively demonstrate the observed gains. Both the quantitative metrics and the qualitative analysis strongly support the claim that Gram anchoring addresses the issue of dense feature collapse.

- The authors have adequately addressed the concerns and questions from reviewers.